# Bias-correcting input variables enhances forecasting of reference crop evapotranspiration

Qichun Yang[1], Quan J Wang[1], Kirsti Hakala[1], Yating Tang[1]

[1]Department of Infrastructure Engineering, The University of Melbourne, Parkville 3010, Australia

*Correspondence to*: Qichun Yang (qichun.yang@unimelb.edu.au)

**Abstract.** Reference crop evapotranspiration (ETo) is calculated using a standard formula with temperature, vapor pressure, solar radiation, and wind speed as input variables. ETo forecasts can be produced when forecasts of these input variables from numerical weather prediction (NWP) models are available. As raw ETo forecasts are often subject to systematic errors, statistical calibration is needed for improving forecast quality. The most straightforward and widely used approach is to directly

calibrate raw ETo forecasts constructed with the raw forecasts of input variables. However, the predictable signal in ETo forecasts may not be fully implemented by this approach, which does not deal with error propagation from input variables to ETo forecasts. We hypothesize that correcting errors in input variables as a precursor to forecast calibration will lead to more skillful ETo forecasts. To test this hypothesis, we evaluate two calibration strategies that construct raw ETo forecasts with the raw (strategy i) or bias-corrected (strategy ii) input variables in ETo forecast calibration across Australia. Calibrated ETo

forecasts based on bias-corrected input variables (strategy ii) demonstrate lower biases, higher correlation coefficients, and higher skills than forecasts produced by the calibration using raw input variables (strategy i). This investigation indicates that improving raw forecasts of input variables could effectively reduce error propagation and enhance ETo forecast calibration. We anticipate that future NWP-based ETo forecasting will benefit from adopting the calibration strategy developed in this study to produce more skillful ETo forecasts.

## 1 Introduction

As a variable measuring the evaporative demand of the atmosphere, reference crop evapotranspiration (ETo) has been widely used to estimate potential water loss from the land surface to the atmosphere (Hopson and Webster, 2009; Liu et al., 2019; Renard et al., 2010). Quantification of ETo has been increasingly performed to support efficient water use and water management (Mushtaq et al., 2019; Perera et al., 2016). Forecasts of short-term ETo (days to weeks) are highly valuable for

real-time decision-making on farming activities and water allocation to competing users (Djaman et al., 2018; Kumar et al., 2012).

A plethora of methods with different statistical assumptions, dependence on observations, and requirements of weather forecasts have been developed to predict future ETo (Bachour et al., 2016; Ballesteros et al., 2016; Karbasi, 2018; Mariito et al., 1993). ETo is affected jointly by temperature, vapor pressure, solar radiation, and wind speed (Bachour et al., 2016; Luo

et al., 2014). Prediction models using these weather variables as inputs allow for representations of atmospheric dynamics and often produce reasonable ETo forecasts (Torres et al., 2011). The increasing availability of weather and climate forecasts based on numerical models has opened up new opportunities for ETo forecasting (Cai et al., 2007; Srivastava et al., 2013; Tian and Martinez, 2014; Zhao et al., 2019a). Forecasts of temperature, vapor pressure, solar radiation, and wind speed from numerical weather prediction (NWP) models/General Circulation Models (GCMs) could be translated into ETo forecasts using the Food

and Agriculture Organization (FAO) ETo equation (Allen et al., 1998; Cai et al., 2007).

Despite the advantages in modelling atmospheric dynamics, flexibility in temporal and spatial scales (Pelosi et al., 2016), and high data availability (Er-Raki et al., 2010), NWP/GCM-based raw ETo forecasts often demonstrate systematic errors (Turco et al., 2017). Limitations in model algorithms, parameterization, and data assimilation often lead to significant errors in raw forecasts of weather variables (Lim and Park, 2019; Vogel et al., 2018). As a result, raw ETo forecasts calculated directly with

the raw forecasts of input weather variables (e.g., temperature, vapor pressure, solar radiation, and wind speed) typically demonstrate substantial inconsistencies with observations (Medina and Tian, 2020; Zhao et al., 2019a), and need to be calibrated to improve forecast quality.

Effective calibration aims to correct errors in raw forecasts and provide unbiased, reliable, and skillful calibrated forecasts. Theoretically, two different strategies could be adopted to achieve this goal in the calibration of ETo forecasts. The first strategy

is to construct raw ETo forecasts directly with the raw forecasts of the input variables and then calibrate the derived ETo forecasts. This strategy lumps errors from the input variables together in the raw ETo forecasts and corrects the combined errors directly (Tian and Martinez, 2014; Zhao et al., 2019a). This strategy is straightforward and thus has been adopted by most existing calibrations of NWP/GCM-based ETo forecasts. For example, Medina et al. (2018) used a linear regression bias-correction method to calibrate ETo forecasts from three NWP models and achieved significant improvements in forecast

quality. Medina and Tian (2020) employed three probabilistic-based calibration methods to calibrate ETo forecasts from multiple NWP models, and generated more skillful and reliable forecasts than using a simple regression bias-correction model. Another probabilistic post-processing method, the Bayesian Joint Probability (BJP) model, was adopted to improve the accuracy and skills of GCM-based ETo forecasts across multiple sites in Australia (Zhao et al., 2019a, 2019b).

Alternatively, ETo forecast calibration could start with correcting errors in input variables. Raw forecasts of input variables

could be improved first, and raw ETo forecasts could then be constructed with the corrected input variables. After that, the derived raw ETo forecasts could be further improved through calibration. This strategy requires one more step than the one using the raw input variables. With the improved input variables, errors in the resultant raw ETo forecasts could be significantly reduced (Nouri and Homaee, 2018; Perera et al., 2014). However, there is no conclusion on whether improving raw forecasts of input variables will eventually add additional skills to calibrated ETo forecasts (Medina and Tian, 2020).

Which calibration strategy produces more skillful calibrated forecasts is a critical question in NWP-based ETo forecasting, but the answer remains unclear. Since NWP/GCM-based ETo forecasting is increasingly conducted to support water resource management, there is a need to investigate the necessity of correcting raw forecasts of the input variables in ETo forecast calibration.

We hypothesize that reducing errors in input variables as a precursor will enhance ETo forecast calibration and lead to more skillful calibrated forecasts. To test this hypothesis, we compare two calibration strategies that construct raw ETo forecasts based on the raw (strategy i) or bias-corrected (strategy ii) input variables in calibrating ETo forecasts across Australia. This study aims to fill a knowledge gap in NWP-based ETo forecasting and develop a calibration strategy to produce more skillful ETo forecasts.

## 2 Method

### 2.1 Reference data and forecasts

In this study, we use the ETo data derived from the Australian Water Availability Project (AWAP)'s gridded data of temperature, vapor pressure, and solar radiation (Jones et al., 2007, 2014), and wind speed data developed by Mcvicar et al. (2008), as observations for ETo forecast calibration. Weather forecasts from the Australian Community Climate and Earth System Simulator G2 version (ACCESS-G2) model are extracted as inputs for the calculation of raw ETo forecasts. We modify the spatial resolution of ACCESS-G2 forecasts using bilinear interpolation to match the AWAP data's grid spacing. The 3-hourly ACCESS-G2 forecasts during 4/2016-3/2019 are aggregated to the daily scale to match the timeframe of the original site observations used to generate the AWAP data. The ACCESS-G2 weather forecasts have a forecast horizon of 9 days. AWAP ETo during 4/1999-3/2019 is used for the training of the calibration model, and data during 4/2016-3/2019 is selected for forecast calibration and evaluation.

### 2.2 Calculation of ETo

We calculate ETo forecasts and AWAP ETo using the FAO56 equation (Allen, et al., 1998):

$$ET_O = \frac{0.408\Delta(R_n-G)+\gamma\frac{900}{T+273}u_2(e_s-e_a)}{\Delta+\gamma(1+0.34u_2)} \tag{1}$$

where $ET_O$ is the reference crop evapotranspiration ($mm\ day^{-1}$); $\Delta$ is the slope of the vapor pressure curve ($kPa\ °C^{-1}$); $R_n$ is net radiation at the crop surface ($MJ\ m^{-2}\ day^{-1}$); $G$ is soil heat flux density ($MJ\ m^{-2}\ day^{-1}$); $\gamma$ is the psychrometric constant ($kPa°C^{-1}$); $T$ is average air temperature (°C); $u_2$ is the wind speed at the height of 2 m ($m\ s^{-1}$); $e_s$ and $e_a$ are saturated and actual vapor pressure ($kPa$), respectively.

In constructing raw ETo forecasts, temperature and solar radiation are readily available from the ACCESS-G2 outputs. To obtain the wind speed forecasts, we first use the forecasts of zonal (*u*) and meridional (*v*) components to calculate wind speed forecasts at 10 m, and then estimate wind speed at 2 m using the equation recommended by FAO (Allen, et al., 1998). In addition, we use the ACCESS-G2 forecasts of air pressure and specific humidity to obtain the vapor pressure forecasts.

### 2.3 Calibration of ETo forecasts

The calibration model used in this study is the Seasonally Coherent Calibration (SCC) model, which is introduced in detail in section 2.3.2. For the calibration across Australia with a spatial resolution of 0.05 degree, we process 281,655 grid cells in total. We apply the SCC model for ETo forecast calibration to each grid cell and lead time separately.

95

**Table 1.** Four sets of ETo forecast calibrations

| Calibrations | Construction of raw ETo forecasts | Application of the SCC model |
|---|---|---|
| Calibration 1 | Raw forecasts of input variables | SCC calibration based on anomaly and climatological mean |
| Calibration 2 | Bias-corrected forecasts of input variables | SCC calibration based on anomaly and climatological mean |
| Calibration 3 | Raw forecasts of input variables | The SCC model applied directly to raw ETo forecasts |
| Calibration 4 | Bias-corrected forecasts of input variables | The SCC model applied directly to raw ETo forecasts |

We conduct four calibrations to evaluate how the two different strategies will affect the calibrated ETo forecasts (Table 1 and Figure S1). Our recent investigation suggests calibrating ETo anomalies, which are calculated as departures from the climatological mean, could produce more skillful calibrated forecasts than calibrating ETo forecasts directly (Yang et al., 2021b). As a result, in this study, we primarily focus on calibrations based on ETo anomalies (Calibrations 1 and 2). The comparison between Calibrations 1 and 2 is to investigate whether the bias-correction of input variables would further improve ETo forecasts when the calibration is conducted based on ETo anomalies and climatological mean. We also conduct additional calibrations which post-process ETo forecasts directly (Calibrations 3 and 4), to test whether the contribution of improving input variables to ETo forecast calibration, if there is any, will depend on how ETo forecasts are calibrated (based on anomalies vs. based on ETo). Calibrations 3 and 4 will help evaluate the general applicability of strategy ii to enhance NWP/GCM-based ETo forecasting. Key steps of the four calibrations could be found in the schematic diagram introducing how raw ETo forecasts are constructed and how calibrations are conducted (Figure S1). In the main text, we primarily analyze results from Calibrations 1 and 2. Improvements with the adoption of bias-correction to input variables in Calibrations 3 and 4 are very similar to Calibrations 1 and 2 (see the Supplementary Material). To avoid redundancy, we mainly present results from Calibrations 3 and 4 in the Supplementary Material.

### 2.3.1 Bias-correction of input variables

In ETo forecast calibration employing strategy ii (Calibrations 2 and 4), we use a non-parametric quantile mapping method (QUANT) to correct raw forecasts of the input variables. The QUANT method has been widely used in hydrological and climatological investigations to correct bias in raw forecasts (Boe et al., 2007). To use QUANT, we first build up the empirical

cumulative density function (CDF) of both raw forecasts and AWAP data for each variable. We then calculate the percentile of each record in raw forecasts in their CDF. Next, these percentiles are used to search values in the corresponding AWAP data, which is then treated as the bias-corrected forecasts.

### 2.3.2 Key steps of ETo forecast calibration using the SCC model

After we construct the raw ETo forecasts, based on either raw (Calibrations 1 and 3) or bias-corrected (Calibrations 2 and 4) forecasts of the input variables, we employ the SCC model to further calibrate the ETo forecasts. For the calibrations (Calibrations 1 and 2) applying SCC to ETo anomalies, the first step is to derive the climatological mean at the daily scale using the 20-year AWAP ETo. Calibrations 3 and 4 skip this step and apply the SCC model to ETo forecasts directly. We use the method developed by Narapusetty et al. (2009) and adopt trigonometric functions and harmonics to simulate the annual

cycle of AWAP ETo to derive the climatological mean:

$$y_{cm}(t) = a_0 + \sum_{j=1}^{H}[a_j \cos(w_j t) + b_j \sin(w_j t)] \tag{2}$$

where $y_{cm}(t)$ is the climatological mean of AWAP ETo at the daily scale; $H$ is the number of harmonics. We use $H=4$ following Narapusetty et al. (2009); $a_0$, $a_j$, and $b_j$ are coefficients, estimated through minimizing the mean squared differences between climatological mean and observations; $w_j=2\pi j/P$, and P is days in one year.

We then remove the climatological mean from both raw ETo forecasts and AWAP ETo to generate anomalies. We calibrate the derived anomalies of raw ETo forecasts against the anomalies of AWAP ETo using the SCC model. The SCC model is composed of four key components, including i) a joint probability model to characterize the connection between raw forecasts and observations, ii) reconstruction of seasonal patterns in raw forecasts based on the long-term observations, iii) reparameterization to obtain parameters for short-archived raw forecasts, and iv) generation of calibrated forecasts with the

parameters and the joint model. The SCC model has been introduced in detail in our site- and continental-scale calibrations of NWP precipitation forecasts (Wang et al., 2019; Yang et al., 2021a).

In this study, we use the Yeo-Johnson transformation method to transform the anomalies of forecasts and reference data to approach a normal distribution (Yeo and Johnson, 2000):

$$\hat{x} = \begin{cases} (\lambda x + 1)^{\frac{1}{\lambda}} - 1, & (x \geq 0, \lambda \neq 0) \\ exp(x) - 1, & (x \geq 0, \lambda = 0) \\ -(\lambda - 2)x + 1)^{\frac{1}{2-\lambda}} + 1, & (x < 0, \lambda \neq 2) \\ -exp(-x) + 1, & (x < 0, \lambda = 2) \end{cases} \tag{3}$$

where $\lambda$ is a transformation parameter; $x$ refers to anomalies of daily raw ETo forecasts or AWAP ETo (*mm day$^{-1}$*); $\hat{x}$ is the transformed $x$.

We assume that the transformed anomalies of ETo forecasts ($f(t)$) and AWAP ETo ($o(t)$) are drawn from a bivariate normal distribution:

$$[f(t), o(t)] \sim BN(f(t), o(t) | \mu_f(m(t)), \sigma_f^2(m(t)), \mu_o(m(t)), \sigma_o^2(m(t)), \rho(m(t))) \tag{4}$$

where $m(t)$ returns the month k (k=1 to 12) of daily forecasts or observations of day $t$; $\mu_f(m(t))$ and $\sigma_f(m(t))$ refer to the marginal distribution's mean and standard deviation of $f(t)$ in month $m(t)$, respectively; $\mu_o(m(t))$ and $\sigma_o(m(t))$ are the mean and standard deviation of the marginal distribution of $o(t)$ in month $m(t)$; $\rho(m(t)$ is the correlation between $f(t)$ and $o(t)$ of month $m(t)$.

With the long-term (20-year) AWAP ETo data, we can directly estimate $\mu_o(m(t))$ and $\sigma_o(m(t))$ based on a maximum likelihood optimization. Calculation of the mean ($\mu_f(m(t))$) and standard deviation ($\sigma_f(m(t))$) using the short-archived raw ETo forecasts (3-year) is subject to significant sampling errors. Instead, we indirectly estimate them using the following linearly regressions:

$$\mu_f(k) = a + b\mu_o(k) \tag{5}$$

$$\sigma_f(k) = c + d\sigma_o(k) \tag{6}$$

$$\rho(k) = r \tag{7}$$

where $k$ refers to month of the year (k= 1 to 12); $a$, $b$, $c$, and $d$ are parameters characterizing the linear relationships; $\rho(k)$ denotes the correlation coefficient between anomalies of raw forecast and AWAP ETo for each month; $r$ is the correlation coefficient between anomalies of raw forecasts and observations in the transformed space (Wang et al.,

160 2019).

With the optimized parameters (means, standard deviations, and correlations) for the BN distribution (equation 4), a conditional distribution for o(t) for a given raw forecast (f(t)) is derived. From this conditional distribution, we randomly draw 100 samples, which are treated as the calibrated ensemble forecasts for that raw forecast. Finally, the calibrated anomalies are back-transformed to their original space and added back to the climatological mean to produce calibrated ETo forecasts.

**2.4 Evaluation of calibrated forecasts**

We evaluate the performance of the calibrations using a strict leave-one-month-out cross-validation, in which each of the 36 months during 4/2016 - 3/2019 and the same month in the 20-year reference data (4/1999 to 3/2019) are left out in parameter inference. Optimized parameters are then used to calibrate raw forecasts of this specific month. This process is repeated until all 36 months are processed. The cross-validation is to make sure that raw forecasts used to generate calibrated forecasts are

not used in parameter optimization.

We also produce climatology forecasts based on the monthly mean and standard deviation parameters of AWAP ETo (equation 4). The randomly sampled climatology is used as the baseline to evaluate the calibrated ETo forecasts. We evaluate the calibrations by checking bias, temporal variability, skill score, and reliability of the calibrated forecasts. We conduct *t-tests* to

compare the performance of bias-correction to input variables and the calibrations of ETo forecasts (Tables S1 and S2). The
evaluation metrics are further introduced in detail as follows.

### 2.4.1 Bias

We evaluate bias of the raw and calibrated forecasts relative to AWAP ETo using the following equation:

$$Bias = \frac{1}{T}\sum_{t=1}^{T}(x(t) - y(t)) \tag{8}$$

where *Bias* refers to the average difference between ETo forecasts and AWAP ETo (*mm day$^{-1}$*); $T$ is total days during the 3-year validation period (4/2016-3/2019); $x(t)$ is raw or calibrated forecasts of ETo (*mm day$^{-1}$*), and $y(t)$ is the corresponding AWAP ETo of the same period. Since bias measures the average difference between forecasts and observations, and could be either possible and negative, comparing biases of forecasts from different Calibrations directly will not demonstrate which Calibration has better performance. To solve this problem, we compare the absolute bias of calibrated forecasts from different calibrations to evaluate how bias-correction of input variables affects the accuracy of calibrated ETo forecasts. Lower absolute bias indicates smaller differences between forecasts and AWAP ETo, and therefore suggests better performances.

### 2.4.2 Temporal variability

We use the Pearson correlation coefficient ($r$) between raw/bias-corrected forecasts of input variables and the corresponding AWAP data to evaluate how quantile mapping improves the temporal patterns of the input variables. We also compare the $r$ between raw ETo forecasts (Calibrations 1 or 3) constructed with raw inputs and AWAP ETo vs. the $r$ between raw ETo forecasts (Calibrations 2 or 4) constructed with bias-corrected inputs and AWAP ETo. In the evaluation of calibrated ensemble forecasts, we use the ensemble mean of the 100 ensemble members to calculate $r$:

$$r = \frac{\sum_{t=1}^{n}(x(t)-\bar{x})(y(t)-\bar{y})}{\sqrt{\sum_{t=1}^{n}(x(t)-\bar{x})^2}\sqrt{\sum_{t=1}^{n}(y(t)-\bar{y})^2}} \tag{9}$$

where $x(t)$ is raw or calibrated forecasts; $\bar{x}$ is the average of $x(t)$; $y(t)$ is the corresponding AWAP ETo data of the same period, and $\bar{y}$ is the average of $y(t)$.

### 2.4.3 Skills of the raw and calibrated forecasts

We use the continuous ranked probability score (CRPS) to measure skills in the raw and calibrated forecasts (Grimit et al., 2006):

$$CRPS(t) = \int\{F(t,x) - H(x - y(t))\}^2 dx \tag{10}$$

$$\overline{CRPS} = \frac{1}{T}\sum_{t=1}^{T}CRPS(t) \tag{11}$$

where $F(t,x)$ is the cumulative density function of an ensemble forecast, and $y(t)$ is the observation at time $t$; $H$ is the Heaviside step function ($H = 1$ if $x - y(t) \geq 0$ and $H = 0$ otherwise); the overbar represents averaging across the $T$ days. For deterministic forecasts, CRPS is reduced to absolute errors.

We further calculate the CRPS skill score ($CRPS_{SS}$) to measure the skills relative to climatology forecasts using the following equation:

$$CRPS_{SS} = \frac{CRPS_{reference} - CRPS_{forecasts}}{CRPS_{reference}} \times 100 \qquad (12)$$

where $CRPS_{reference}$ is the CRPS value of climatology forecasts (%); $CRPS_{forecasts}$ refers to CRPS value of raw or calibrated forecasts.

In the calculation of CRPS skill score, both climatology forecasts or the last observations (persistence) have been used as reference forecasts (Pappenberger et al., 2015; Thiemig et al., 2015). However, reference forecasts based on persistence are

more suitable for evaluating the performance of forecasts shorter than two days. As a result, we choose climatology forecasts as the reference, since errors in climate forecasts are similar among all lead times and thus could be used to demonstrate the increasing errors in raw and calibrated forecasts as lead time advances. For $CRPS_{SS}$ of Calibrations 1 and 2, climatology forecasts from Calibration 1 are used; For $CRPS_{SS}$ of Calibrations 3 and 4, climatology forecasts from Calibration 3 are used. Positive skill scores indicate better skills than the climatology forecasts and vice versa. We use percentage as the unit of CRPS

skill score so low skill scores at long lead times will be converted from small decimals to more readable percent.

### 2.4.4 Reliability

We evaluate the reliability of calibrated forecasts using the probability integral transform (PIT) value calculated with the following equation:

$$\pi(t) = F\big(t, x = y(t)\big) \qquad (13)$$

where $F(t, x)$ is the cumulative density function of the ensemble forecast, and $y(t)$ is the AWAP ETo. For reliable forecasts, $\pi(t)$ follows a uniform distribution. We use the alpha index ($\alpha$) to summarize the reliability in each grid cell with the following equation to check the spatial patterns of forecast reliability (Renard et al., 2010):

$$\alpha = 1 - \frac{2}{n} \sum_{t=1}^{n} \left| \pi^*(t) - \frac{t}{n+1} \right| \qquad (14)$$

where $\pi^*$ (t) is the sorted $\pi(t)$, t=1,2,…n in ascending order, and $n$ is the total number of days during 4/2016-3/2019. The $\alpha$

index measures the total deviation of calibrated forecasts from the corresponding uniform quantile. Perfectly reliable forecasts should have an α of 1, and forecasts with no reliability would have an α of 0.

We further evaluate the reliability of calibrated ETo forecasts from calibration 2 using the reliability diagram (Hartmann et al., 2002), which assesses how well the predicted probabilities of forecasts match observed frequencies. We convert the calibrated ensemble ETo forecasts to forecast probabilities exceeding three thresholds, including 3, 6, and 9 mm day[-1]. We pool forecasts

of different grid cells, days, and lead times together in the calculation of forecast probability. In the reliability diagram, perfectly reliable forecasts would demonstrate a curve along the diagonal. A plotted curve above the diagonal indicates underestimations and vice versa.

**3 Results**

**3.1 Quality of raw and bias-corrected input variables**

Raw forecasts of the five input variables demonstrate significant inconsistencies with the corresponding AWAP data (Figures S2-S6). In most parts of Australia, raw daily maximum temperature (Tmax) forecasts are lower than AWAP data by 1-2 °C. Overpredictions in Tmax are only found in coastal areas of northwestern Australia. The daily minimum temperature (Tmin) is underpredicted by more than 1.5 °C in western and central parts of Australia by the raw forecasts, but is overpredicted by ca. 1 °C in eastern and southern Australia. Vapor pressure is underpredicted in western and central regions by ca.14%, but is

overpredicted by ca. 6% in coastal areas of southeastern Australia by the raw forecasts. Raw solar radiation forecasts are about 5% higher than AWAP data across Australia. Forecasted wind speed is higher than the reference data by more than 1 m s$^{-1}$ (or by ca. 63%) in most parts of Australia. For each input variable, spatial patterns of biases in raw forecasts are consistent across the 9 lead times, demonstrating systematic errors in the raw NWP forecasts. According to our statistical test, overpredictions or underpredictions in raw forecasts of the input variables are statistically significant *(P<0.05)* for most lead times (Table S1).

Raw forecasts of the input variables generally agree with the AWAP data in temporal patterns during the study period, but the *r* varies with variables (Figures S7-S11). The two temperature variables (Tmax and Tmin) have higher *r* (>0.9) than the other three variables, and wind speed forecasts demonstrate the lowest correlations with AWAP data. For all variables, the *r* decreases with lead time, indicating higher uncertainties at long lead times in raw forecasts.

Quantile mapping effectively corrects biases in raw forecasts of the input variables. Through the bias correction, significant

overpredictions and underpredictions in raw forecasts of the five variables are significantly reduced, resulting in biases close to zero for all lead times across Australia (Figures S2-S6). In addition, quantile mapping also improves the correlation between forecasts of input variables and AWAP data (Figures S7-S11). The most significant improvements are found in wind speed forecasts, in which the *r* is improved by up to 0.2 in central and southern parts of Australia. Forecasts of Tmax and solar radiation also demonstrate higher *r* with the adoption of quantile mapping. Both increases and slight decreases were found for

vapor pressure and Tmin, indicating less significant improvements in temporal patterns than other variables.

**3.2 Quality of raw ETo forecasts constructed with raw and bias-corrected input variables**

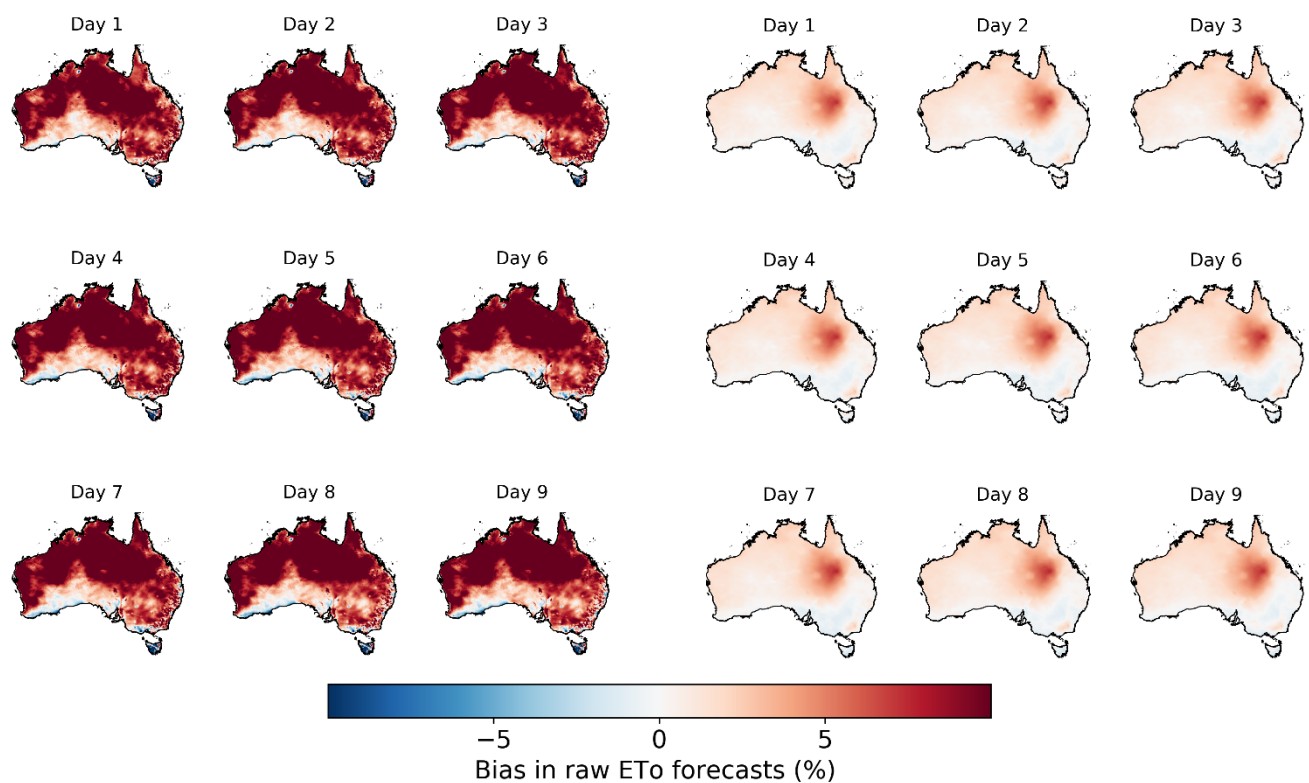

**Figure 1: Bias in (three panels on the left) raw ETo forecasts constructed with raw forecasts of input variables and (three panels on the right) raw ETo forecasts constructed with bias-corrected input variables.**

Raw ETo forecasts constructed with the bias-corrected input variables are much more accurate than those calculated with raw forecasts of the input variables (Figures 1 and S12). When raw ETo forecasts are constructed with raw input variables, biases in input variables are translated into errors in the raw ETo forecasts, which demonstrate substantial positive biases of 1 *mm day$^{-1}$* (ca.12% larger than AWAP ETo) in large areas of central and northern Australia. In the raw ETo forecasts constructed with bias-corrected input variables, biases approach zero in most parts of Australia, except for inland regions of Queensland, where biases are close to 0.3 *mm day$^{-1}$* (ca. 4% larger than AWAP ETo). The remaining biases are significantly lower than those in the raw ETo forecasts constructed with raw input variables (Table S2).

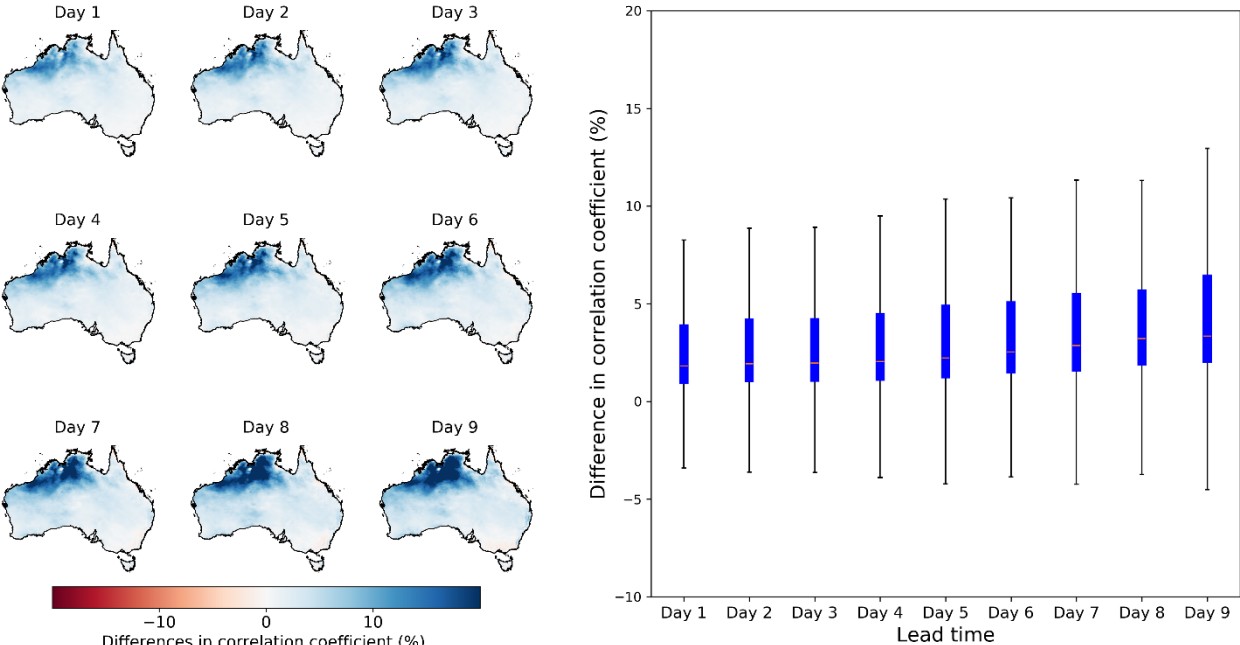

**Figure 2: The comparison between the correlation coefficient of raw ETo forecasts constructed with the bias-corrected inputs and AWAP ETo vs. the correlation coefficient of raw ETo forecasts constructed with the raw inputs and AWAP ETo. The boxplot on the right summarizes results across all grid cells.**

The adoption of quantile mapping to input variables also improves the temporal patterns of raw ETo forecasts (Figure 2). Compared with the raw ETo forecasts constructed with raw input variables, the raw ETo forecasts based on bias-corrected inputs generally shows higher correlations with AWAP ETo, particularly in northern Australia, where $r$ is improved by more than 10%. However, due to the nonlinearity in the calculation of ETo using the input variables, spatial patterns of improvements in $r$ (Figure 2) does not resemble improvements in any individual input variables (Figures S7 to S11). The improvements in $r$ of raw ETo forecasts seem to be contributed jointly by these input variables and their interactions.

### 3.3 Bias in calibrated ETo forecasts

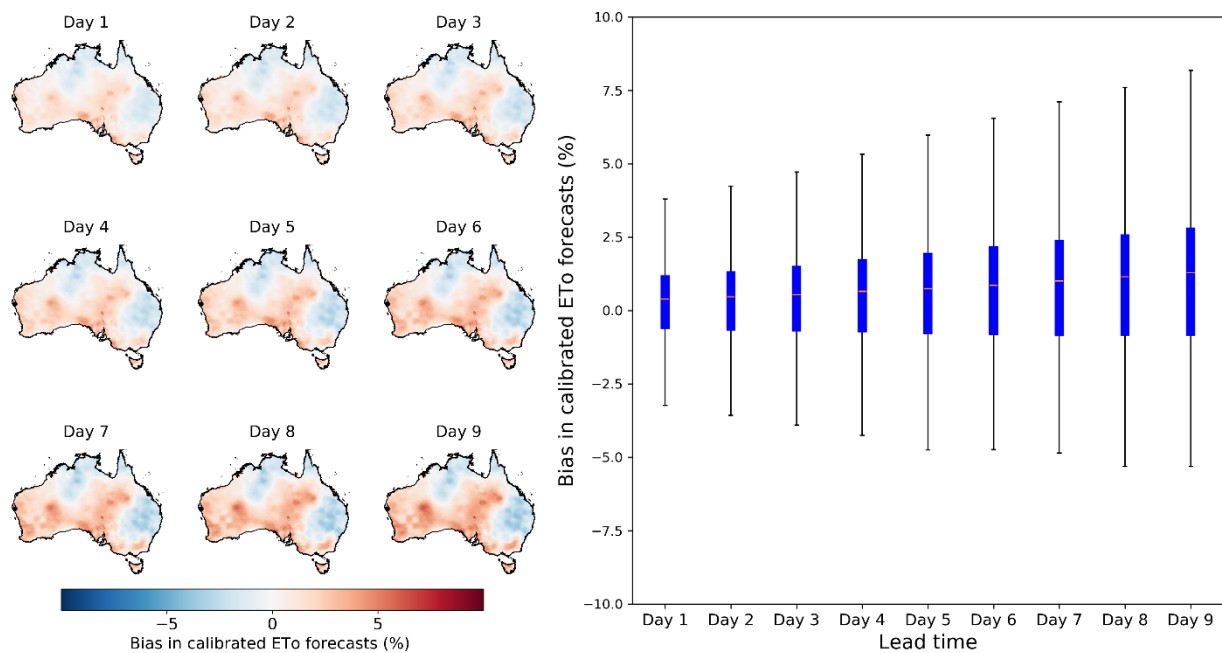

285

**Figure 3: Bias in calibrated ETo forecasts from Calibration 2, in which raw ETo forecasts are constructed with bias-corrected input variables. Maps on the left show the spatial patterns of bias, and the boxplot on the right summarizes biases across all grid cells.**

290     The calibration with the SCC model further reduces biases in ETo forecasts (Figure 3). The calibrated ETo forecasts from Calibration 2 show low biases close to zero across all grid cells and lead times. Overpredictions in Queensland in the raw ETo forecasts calculated with the bias-corrected input variables are effectively corrected (Figures 1, 3, and S12), leading to lower biases in the calibrated forecasts. According to the *t-test* (Table S2), biases of calibrated forecasts at the first two lead times are not significantly different from zero, indicating the effective bias reduction through Calibration 2. For the remaining 7 lead
295     times (Days 3 to 9), the overall biases are slightly higher than zero. The remaining biases in calibrated forecasts reflect deviations of ETo during the evaluation period (4/2016-3/2019) from the climatology during 4/1999-3/2019, since the SCC parameters are inferred with the 20-year AWAP ETo (equations 4 to 7).

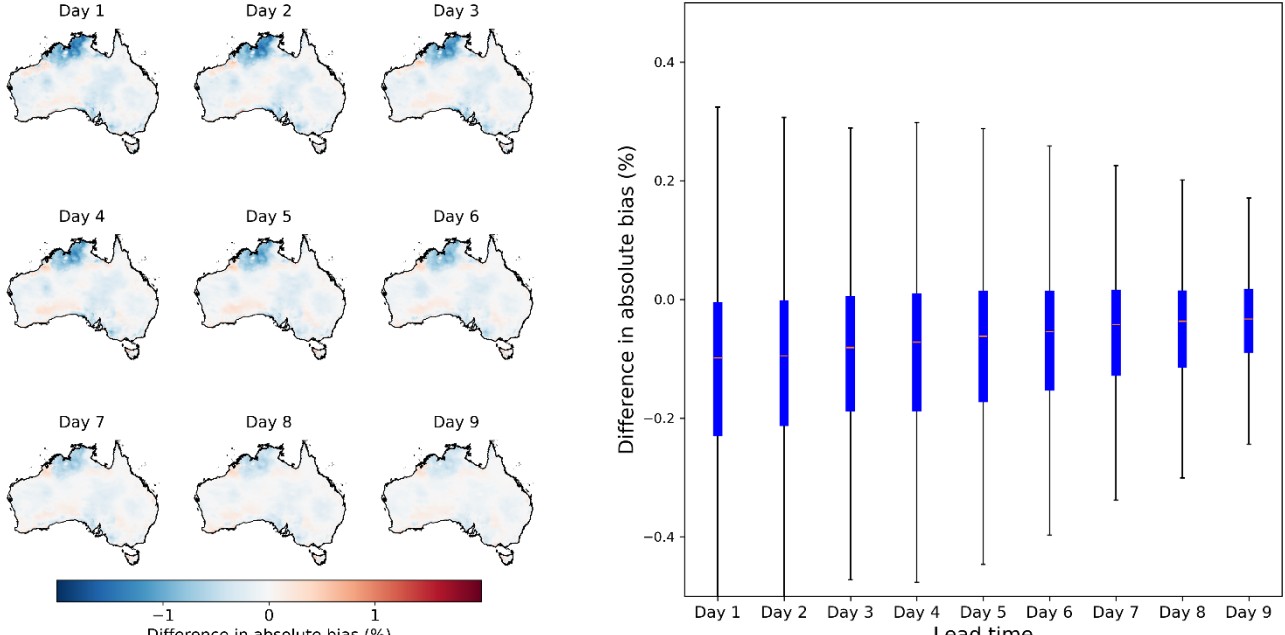

**Figure 4: Differences in absolute bias between calibrated ETo forecasts from Calibration 2 with Calibration 1. Maps on the left show the spatial patterns of difference in absolute bias, and the boxplot on the right summarizes results across all grid cells.**

Compared with the calibration constructing raw ETo forecasts with raw forecasts of input variables (Calibration 1), the post-processing based on bias-corrected input variables (Calibration 2) produces more accurate calibrated ETo forecasts (Figure 4). Specifically, calibrated ETo forecasts from Calibration 2 demonstrate significantly smaller ($P<0.05$) absolute biases than those of Calibration 1 across large areas of northern Australia, particularly in coastal regions of the Northern Territory. Larger reductions in absolute bias in northern Australia coincide with the improvements in the correlation between raw ETo forecasts and AWAP ETo (Figure 2). However, unlike the improvements in *r* for all lead times in raw ETo forecasts, the improvements in absolute bias are more pronounced at short lead times (Days 1-3) than long lead times (Days 7-9). The uneven improvements across different lead times may be caused by the significant intrinsic uncertainties in forecasts, which have hindered the manifestation of improvements to raw ETo forecasts at long lead times in calibrated forecasts.

**3.4 Correlation between calibrated ETo forecasts and AWAP ETo**

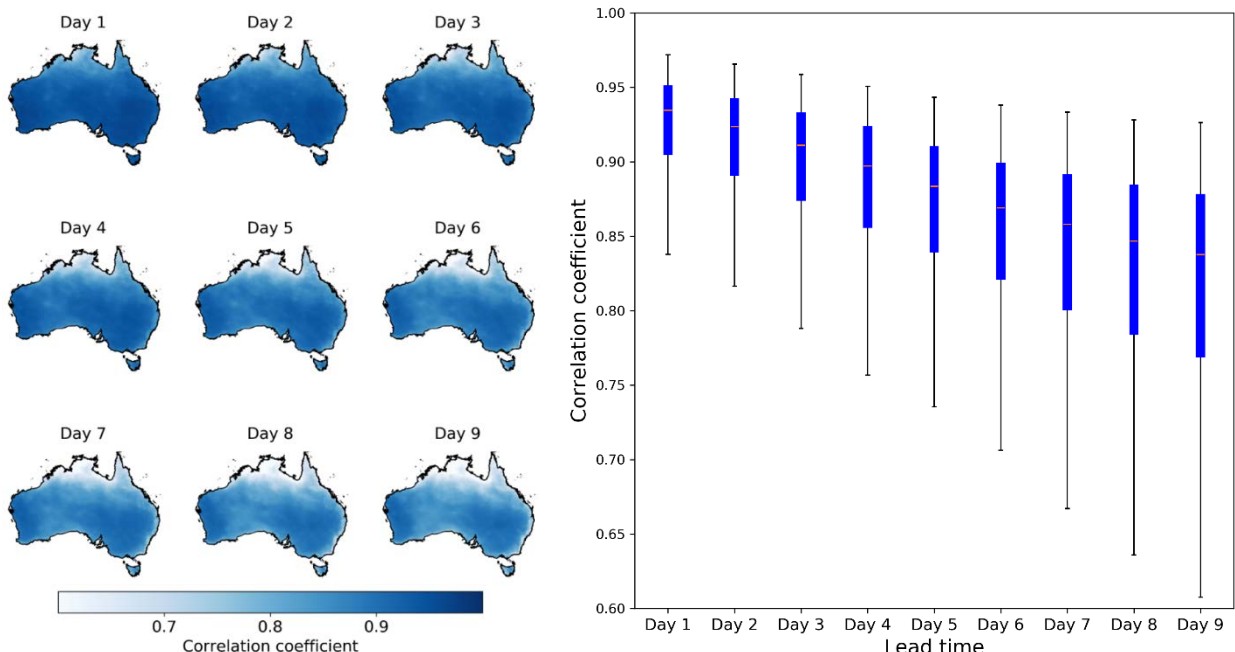

**Figure 5: The correlation coefficient between calibrated ETo forecasts from Calibration 2 and AWAP ETo. Maps on the left show the spatial patterns of *r*, and the boxplot on the right summarizes results across all grid cells.**

We further examine the representation of ETo temporal variability by calibrated forecasts. The *r* between calibrated ETo forecasts from Calibration 2 and AWAP ETo demonstrates high consistency in temporal variability (Figure 5). The correlation coefficient is mainly above 0.85 across Australia for the first three lead times. With increases in lead time, *r* decreases but remains above 0.75 in most grid cells, even at lead time 9. Coastal areas of northern Australia have lower *r* values than other regions of the country, demonstrating higher uncertainties in ETo forecasts this area. Deficiencies in ACCESS models in simulating dynamics of tropical climate systems may have resulted in the low r in northern Australia.

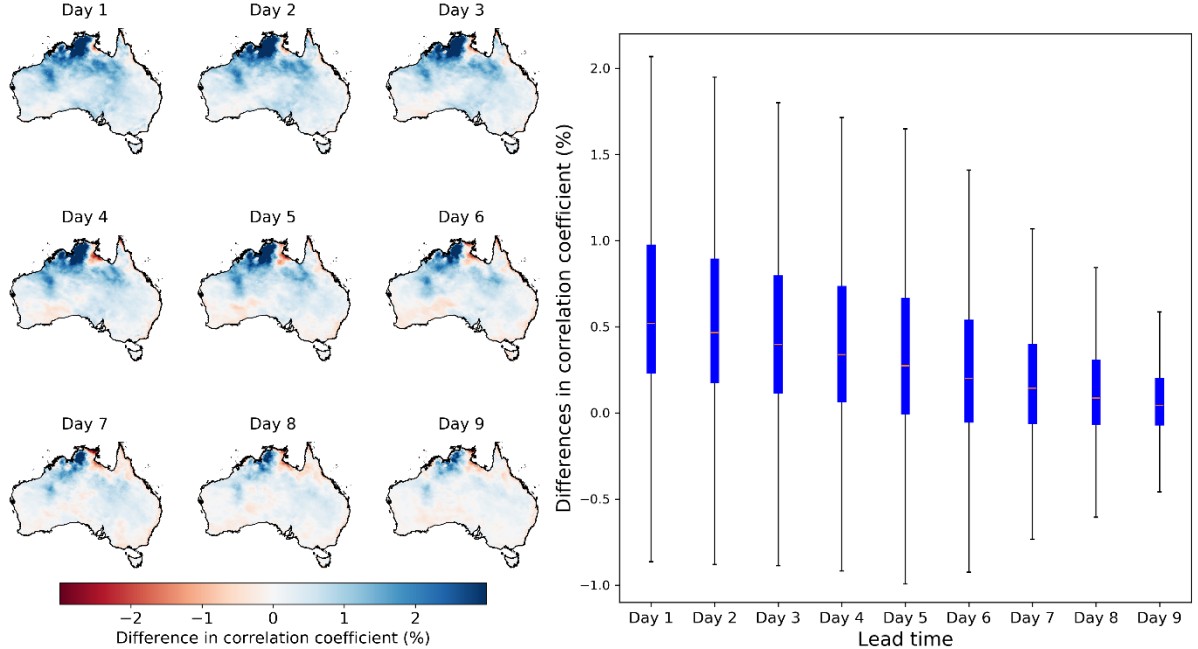

**Figure 6: Differences in the correlation coefficients between calibrated forecasts from Calibration 2 and AWAP ETo vs. Calibration 1. Maps on the left show the spatial patterns of differences in *r*, and the boxplot on the right summarizes results across all grid cells.**

The adoption of bias-correction to raw forecasts of input variables results in better representation of ETo variability in calibrated ETo forecasts (Figure 6 and Table S2). Increases in *r* are particularly significant for short lead times (Table S2). Specifically, for the first three lead times, increases in *r* are mainly around 2% in central and northern parts of Australia, with more pronounced (>4%) increases found in coastal regions of the Northern Territory. For lead times 4 to 6, increases in *r* values are mainly above 1%. For the remaining 3 lead times (7 to 9), increases in *r* are mainly located in Northern Territory. Spatial patterns of *r* increases from Calibration 1 to Calibration 2 are consistent across the 9 lead times.

Spatial patterns of improvements in *r* in calibrated ETo forecasts (Figure 6) are consistent with the improvements in *r* of raw ETo forecasts with the adoption of bias-correction (Figure 2), particularly for the short lead times. The improvements in *r* of calibrated ETo forecasts (Figure 6) may also lead to more reasonable conditional distributions for a given raw forecast (equation 4). As a result, regions showing improvements in *r* in calibrated ETo forecasts (Figure 6) often demonstrate reductions in absolute bias (Figure 4).

### 3.5 Improvements in forecast skills

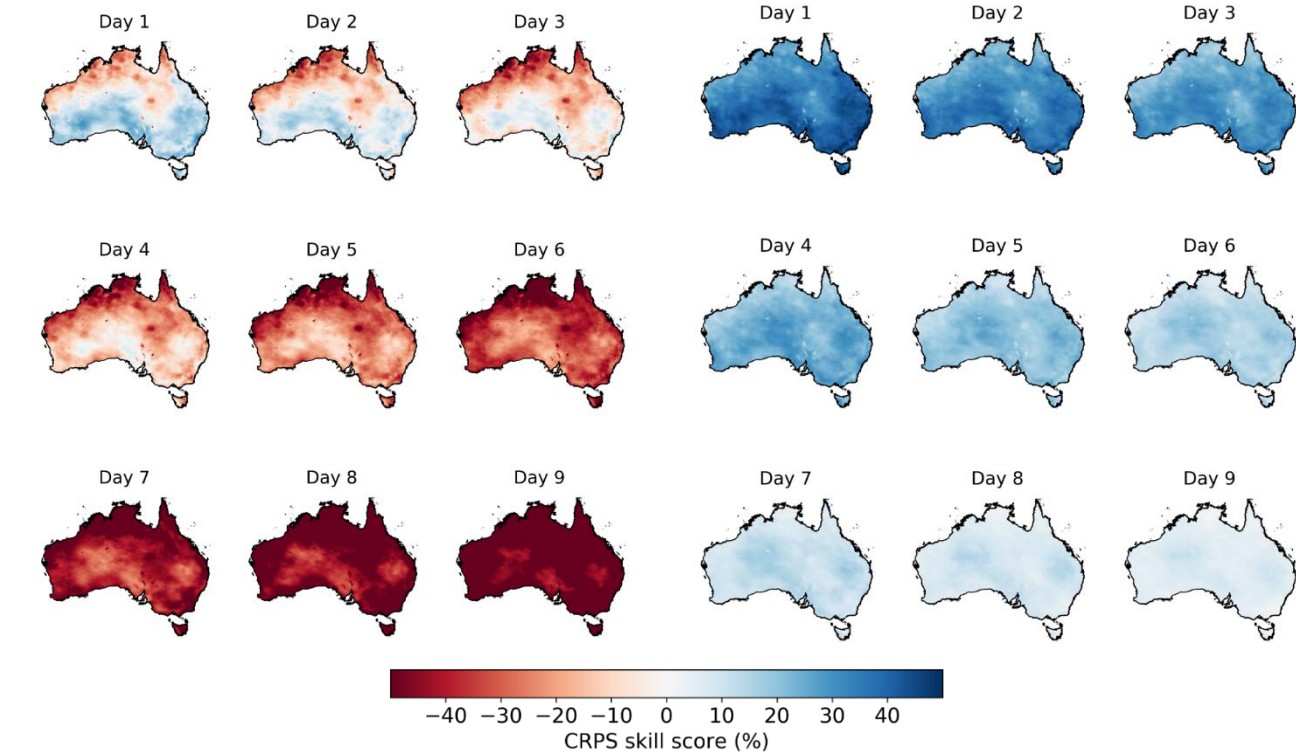

**Figure 7: CRPS skill score in the (three panels on the left) raw ETo forecasts calculated with bias-corrected input variables and (three panels on the right) calibrated forecasts from Calibration 2.**

345 The calibration of ETo forecasts with the SCC model significantly improves forecast skills. The raw ETo forecasts calculated with bias-corrected input variables demonstrate low skills, even at short lead times (Figures 7 and S13). Specifically, for the first two lead times, central and southern Australia show skills better than the climatology forecasts by 10% to 20%. However, in most parts of northern Australia, raw forecasts are worse than randomly sampled climatology. Skills in raw ETo forecasts decrease quickly with lead time. Regions with positive skills shrink substantially at lead times 3 and 4, and disappear at longer

350 lead times. At lead time 9, skills of raw forecasts are mainly below -40%.

The calibration significantly improves forecast skills across all lead times (Table S2). Calibrated ETo forecasts from Calibration 2 show CRPS skill scores above 35% at lead time 1 across Australia, and the skills are generally above 30% at lead times 2 and 3. Since ETo forecasts have been widely used to inform real-time decision-making for farming, high skills in calibrated ETo forecasts for the short lead times are expected to be highly valuable for activities such as irrigation scheduling.

355 Although skills of calibrated forecasts also decrease with lead time, they remain above zero at long lead times (Figures 7 and S13).

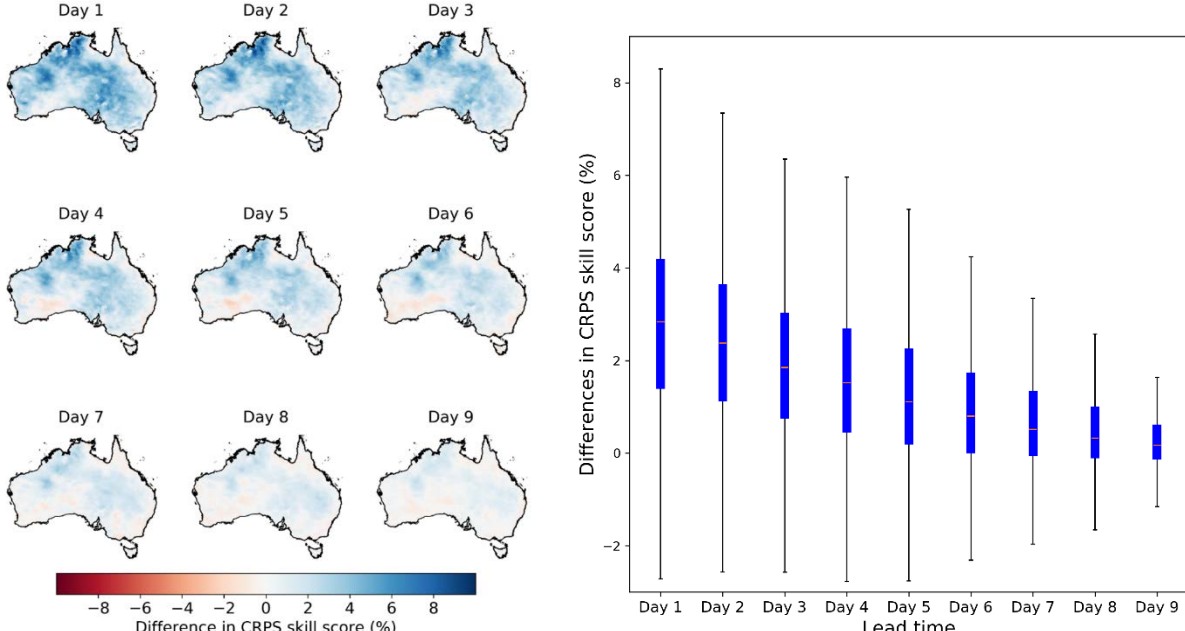

**Figure 8: Differences in CRPS skill score between the calibrated ETo forecasts from Calibration 2 with those from Calibration 1. Maps on the left show the spatial patterns of difference in CRPS skill score, and the boxplot on the right summarizes results across all grid cells.**

We further compare skills of calibrated ETo forecasts between Calibrations 2 and 1 (Figure 8). We achieve significant increases *(P<0.05)* in CRPS skill scores with the adoption of bias-correction to input variables, particularly for short lead times (Table S2). For the first three lead times, the CRPS skill scores are increased by more than 4% in northwestern and eastern Australia, when input variables are bias-corrected. For lead times 4 to 6, the CRPS skill scores are increased by 2% in these regions of Australia. Although the differences become less noticeable at lead times 7 to 9, they are generally above zero in most parts of Australia. Increases in CRPS skill score across the 9 lead times are in line with improvements in absolute bias (Figure 4) and correlation coefficient (Figure 6), which all show more significant improvements at short lead times than long lead times.

### 3.6 Reliability of calibrated ETo forecasts

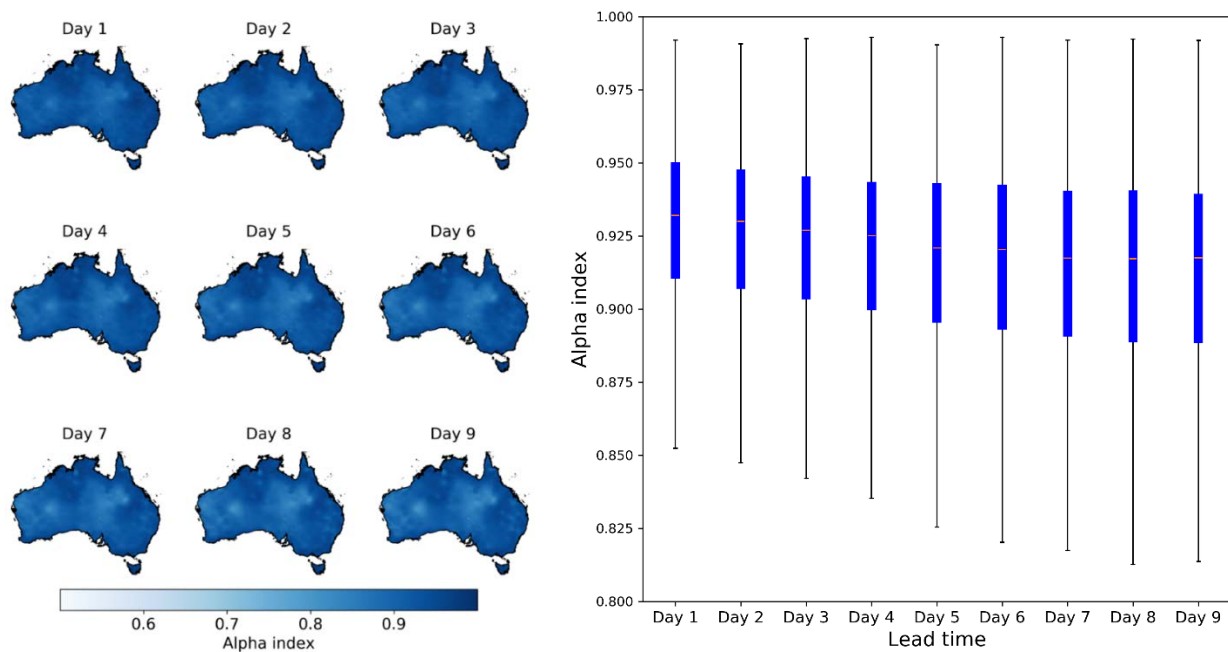

**Figure 9: Alpha index of calibrated ETo ensemble forecasts from Calibration 2. Maps on the left show the spatial patterns of the alpha index, and the boxplot on the right summarizes results across all grid cells.**

The calibrated ensemble ETo forecasts from Calibration 2 demonstrate high reliability (Figure 9). In addition to correcting bias, the SCC model converts deterministic raw forecasts to ensemble forecasts, which use 100 ensemble members to quantify forecast uncertainty. Figure 9 demonstrates highly reliable ensemble spreads in calibrated forecasts across all lead times. In most grid cells, the $\alpha$ index is over 0.9, indicating reasonable representations of ETo uncertainties by the ensemble spread, which is neither too narrow nor too wide (Figure 9). Calibrated forecasts from Calibration 1, which uses raw input variables, demonstrate similar reliability as those from the calibration with bias-corrected input variables (Calibration 2). Differences in $\alpha$ index of the calibrated ETo forecasts from Calibrations 1 and 2 are almost negligible (Figure S14), as shown by the *t-test* (Table S2), indicating that bias-correcting raw forecasts of input variables does not lead to significant changes in the reliability of calibrated ETo forecasts.

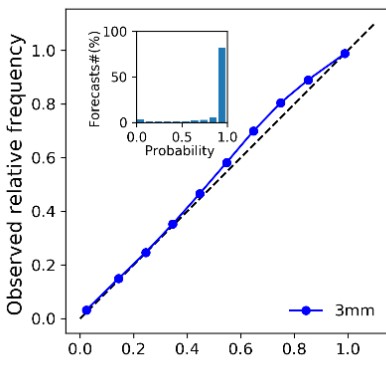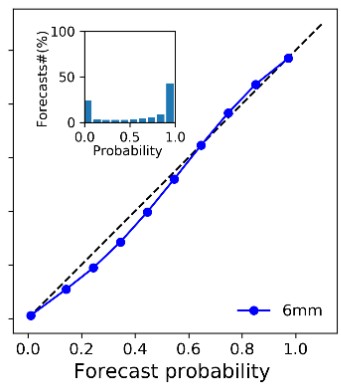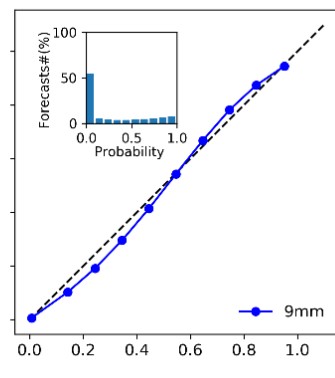

**Figure 10: Reliability diagrams of calibrated ETo forecasts during 4/2016-3/2019 with thresholds of 3, 6, and 9 mm day⁻¹**

The reliability diagram further confirms the consistency between forecast probabilities and observed frequencies (Figure 10). The plotted curves based on three thresholds (3, 6, and 9 mm day⁻¹) are mainly distributed along the 1:1 line, further indicating the high reliability of calibrated ETo forecasts.

### 3.7 Results from Calibrations 3 and 4

We also compare the bias, correlation coefficient, CRPS skill score, and reliability of calibrated forecasts from Calibrations 3 and 4, to evaluate whether we can obtain similar improvements through the bias-correction of input variables if we conduct the ETo forecast calibration in a different way (without using ETo climatological mean and anomalies). Results show that the adoption of bias-correction also leads to lower bias, higher correlation coefficient, and higher CRPS skill score in terms of magnitude, spatial patterns, and trend along the lead times, when ETo forecasts are calibrated directly (Figure S15-S17). In addition, the alpha index was only slightly different between Calibrations 3 and 4 (Figure S18). This additional comparison further confirms the general applicability of strategy ii for enhancing NWP-based ETo forecasting.

### 3.8 Summary of results

Although the selected metrics measure different aspects of forecast quality, they generally agree with each other in demonstrating improvements in calibrated ETo forecasts with the adoption of the Strategy ii. As introduced in the Method section, bias measures average differences; correlation coefficient shows consistency between observations and forecasts in temporal variability; the CRPS skill score measures the performance of the calibrated forecasts relative to climatology forecast; the α index is an indicator showing whether the distribution of calibrated forecasts is overconfident or underconfident. As a result, these metrics may differ from each other in magnitude when used to evaluate different calibrations (Figures 4, 6, 8, and S14). However, improvements in bias, correlation, and skills with the adoption of bias-correction to input variables are

generally consistent in spatial patterns. Compared with the other three metrics, α index demonstrates less significant changes when input variables are bias-corrected first (Table S2 and Figure S14), mainly because this index is less sensitive to changes in calibrated forecasts than other metrics.

## 4 Discussion

### 4.1 Importance of improving forecasts of input variables for NWP-based ETo forecasting

This investigation further highlights the importance of statistical calibration in NWP-based ETo forecasting (Medina and Tian, 2020). According to an investigation across 40 sites in Australia, raw ETo forecasts constructed with NWP outputs reasonably captured the magnitude and variability of ETo, but forecast skills better than climatology were only limited to the first 6 lead times (Perera et al., 2014). Our investigation suggests that statistical calibration could substantially improve forecast skills and successfully extend the skillful forecasts to lead time 9 across Australia. Findings of this investigation agree well with the site-scale short-term ETo forecasting based on GCM outputs (Zhao et al., 2019a) in the improvements of forecast skills through statistical calibration. Calibrated forecasts from Calibration 2 demonstrate similar skills as Zhao et al. (2019a) across three Australian sites. Thanks to the capability of SCC in calibrating short-archived forecasts (Wang et al., 2019), we achieve the improvements based on much shorter archived raw forecasts (3-year vs. 23-year) than Zhao et al. (2019a). Calibrated forecasts from Calibration 2 also demonstrate low biases (0.32-0.95%) comparable with calibrated ETo forecasts (0.49-0.63%) based on the Bayesian Model Averaging (BMA) model and weather forecasts from three NWP models in the U.S. during 2014-2016 (Medina and Tian, 2020).

This investigation also contributes to filling a knowledge gap in NWP-based ETo forecasting. Although previous calibrations using raw forecasts of input variables to construct the raw ETo forecasts (strategy i) for calibration often achieved significant improvements in skills, it is unclear whether improving forecasts of input variables could further enhance ETo forecast calibration (Medina and Tian, 2020). How the raw ETo forecasts should be constructed represents a critical knowledge gap in the area of NWP-based ETo forecasting (Medina and Tian, 2020). Results of this investigation provide strong evidence for the necessity of improving input variables prior to constructing raw ETo forecasts. The non-linear and non-stationary behaviors of the input variables used for ETo calculation have been reported (Paredes et al., 2018). This study suggests that when raw input variables are used to construct the raw ETo forecasts, complex interactions among these variables may lead to errors in raw ETo forecasts that could not be effectively corrected through statistical calibration. Bias-correction of input variables could help prohibit the propagation of errors from input variables to ETo forecasts (Zappa et al., 2010), as evidenced by the higher accuracy and higher skills in calibrated ETo forecasts when input variables are bias-corrected. In addition, a further evaluation based on a different way of implementing the SCC model demonstrates similar improvements in calibrated ETo forecasts with the adoption of bias-correction to input variables (Calibrations 3 and 4). Results from Calibrations 3 and 4 further confirms that additional skills have been added to raw ETo forecasts through the bias-correction of input variables, and the improvements

to calibrated ETo forecasts tend to be independent of calibration models. Consequently, we anticipate that future NWP-based ETo forecasting could benefit from adopting this calibration strategy to produce more skillful calibrated ETo forecasts.

## 4.2 Implications for forecasting of integrated variables and future work

This investigation also provides valuable implications for the forecasting of integrated variables, which are derived based on multiple NWP/GCM variables. Variables such as drought index (Zhang et al., 2017), bushfire danger index (Sharples et al., 2009), and severe weather index (Rabbani et al., 2020), are often derived by combining multiple weather variables produced by NWP models. Our investigation suggests that improving the input variables could effectively reduce error propagation from inputs to integrated variables. This extra step is proven to be particularly useful in reducing errors in the integrated variables that could not be corrected through calibration. We anticipate that this extra step could help improve the predictability of integrated variables.

Although we have conducted thorough analyses on the contribution of improving input variables to ETo forecast calibration, further investigations will be needed to validate the robustness of findings in this study. First, we anticipate that the ETo forecasts could be further improved if a more sophisticated calibration model is applied to raw forecasts of the input variables. In this study, we adopt a simple bias-correction method to improve the input variables. Limitations of quantile mapping have been reported in previous studies (Schepen et al., 2020; Zhao et al., 2017). Our analyses demonstrate that the raw ETo forecasts calculated with the bias-corrected input variables still show low forecast skills, particularly at long lead times (Figure 7). If a more sophisticated calibration method is employed to the input variables, error propagation from input variables to ETo forecasts will likely be further reduced. As a result, we anticipate that the calibrated ETo forecast will gain further improvements in forecast skills. Another advantage of correcting input variables with a sophisticated model is that it will produce a set of skillful calibrated weather forecasts. Well-calibrated forecasts of temperature, vapor pressure, solar radiation, and wind speed, could be useful for forecast users such as crop modelers and bushfire managers.

Second, the two calibration strategies should be tested using other NWP models. In this study, we use one NWP model to investigate a critical knowledge gap in NWP-based ETo forecasting. Additional investigations are needed to examine whether improvements achieved with the adoption of calibration strategy ii will hold for ETo forecasting based on other NWP models. Third, further investigations based on other calibration models are needed to validate findings of this investigation. Our analyses based on two different methods (based on ETo anomalies vs. based on original ETo) demonstrate similar improvements in calibrated ETo forecasts with the adoption of bias-correction to input variables. Additional evaluations will be needed to verify whether forecast skills will be improved using strategy ii but based on a different calibration model. In addition, we use bilinear interpolation to match the NWP forecasts and AWAP data. More sophisticated remapping methods should be evaluated to understand the impacts of forecast regridding on statistical calibration.

The applicability of the calibration strategy developed in this study to seasonal ETo forecasting should be further investigated. Seasonal ETo forecasting based on GCM climate forecast has been increasingly performed (Tian et al., 2014; Zhao et al.,

2019b). In these investigations, raw ETo forecasts were also constructed directly with raw GCM climate forecasts. As a result, it is expected that these investigations have suffered from error propagation from input variables to seasonal ETo forecasts. Whether the calibration strategy (strategy ii) developed in this study will be applicable to seasonal ETo forecasting warrants further investigations.

## 5 Conclusions

NWP outputs have been increasingly used for ETo forecasting to support water resource management. Statistical calibration plays an essential role in improving the quality of ETo forecasts. However, it is unclear whether improving raw forecasts of input variables is necessary for the calibration of ETo forecasts. We aim to fill this knowledge gap through a thorough comparison of two calibration strategies in the calibration of NWP-based ETo forecasts.

This investigation clearly suggests the necessity of improving input variables as part of ETo forecast calibration. With this

extra step, the bias, correlation coefficient, and skills of the calibrated ETo forecasts are all improved. Further investigation indicates that the improvements tend to be independent of the calibration method applied to ETo forecasts. Forecasting the highly variable ETo is often challenging. This investigation addresses a common challenge in NWP-based ETo forecasting and develops an effective calibration strategy for adding extra skills to ETo forecasts. We anticipate that future NWP-based ETo forecasting could benefit from adopting this strategy to produce more skillful calibrated ETo forecasts. This strategy is

also expected to be applicable to enhancing the forecasting of other integrated variables that are calculated using multiple NWP/GCM variables as inputs.

**Data availability:**

Data used in this study are available by contacting the corresponding author.

**Author contributions:**

Q. Yang and Q. J. Wang conceived this study. Q. J. Wang developed the calibration model. Q. Yang took the lead in writing and improving the manuscript. All authors, including K Hakala and Y. Tang, contributed to discussing the results and improving this study.

**Competing interests:**

The authors declare that there is no conflict of interest regarding the publication of this article.

**Acknowledgments:**

This study has been supported by a collaborative research project (TP707466) between the University of Melbourne and the
Australian Bureau of Meteorology and an ARC Linkage Project (LP170100922). Computations of this research were undertaken with the assistance of resources and services from the National Computational Infrastructure (NCI), which is supported by the Australian Government. This research was supported by the Sustaining and strengthening merit-based access to National Computational Infrastructure (NCI) LIEF Grant (LE190100021) and facilitated by The University of Melbourne.

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
