# Peer review of "Bias-correcting input variables enhances forecasting of reference crop evapotranspiration"

_Hydrology and Earth System Sciences, 2021_

## Referee Comment (RC1)

Review to Yang et ., 2021, Bias-correcting individual inputs prior to combined calibration leads to more skillful forecasts of reference crop evapotranspiration. HESSD.

In this study, the authors investigated a critical issue in the forecasting of short-term reference crop evapotranspiration (ETo) based on NWP outputs. It is getting popular that weather forecasts from NWP models are used to predict water loss through evapotranspiration. Such information is highly valuable for the effective management of water resources, particularly in arid/semi-arid regions. This investigation develops a new methodology that effectively corrects errors in ETo forecasts, and adds extra skills to statistical calibration. I believe this new post-processing strategy could benefit future NWP-based ETo forecasting. To improve this work, the authors should pay special attention to the following key issues:

1, Presentation of the results could be improved. Currently, the authors use maps to show/compare results from different model experiments. These figures could demonstrate the spatial patterns of modeling results. However, it might be more useful if the authors could summarize regional results in a different way, such as using box-plots. I believe that will better show readers the overall statistical information across the whole country than simply plotting the results as maps.

2, Implications for ETo forecasting at the monthly or seasonal scales should be further discussed. ETo forecasting based on monthly or seasonal climate forecasts from GCMs is also widely performed. This study develops the new strategy for short-term forecasts. The applicability of this method to ETo forecasting based on GCM forecasts should be briefly discussed, to benefit a broader range of readers.

Specific comments:

Line 20, rewrite this sentence. Not clear

Line 74 Calibrate->calibrate

Line 80 compiled as the inputs…..

Line 95 10m -> 10 m.

Line 107-108, need to clarify what the anomaly and climatological mean are referring to

Line 165 consider rewriting this sentence. Does not read well.

Line 172, what is specific month

Figures in Results: shouldn't the figures be centralized?

Line 360, not calibrate directly, should be without correcting forecasts of the inputs

Line 365, consider rewriting this sentence

Line 377-378, two 'calibration models' consider to rewrite

Line 385, in the calibrated forecasts

Line 386, consider making it shorter and clearer

---

## Author Comment (AC1)

**Responses to Reviewer #1**

**Point #1**

Review to Yang et ., 2021, Bias-correcting individual inputs prior to combined calibration leads to more skillful forecasts of reference crop evapotranspiration. HESSD.

In this study, the authors investigated a critical issue in the forecasting of short-term reference crop evapotranspiration (ETo) based on NWP outputs. It is getting popular that weather forecasts from NWP models are used to predict water loss through evapotranspiration. Such information is highly valuable for the effective management of water resources, particularly in arid/semi-arid regions. This investigation develops a new methodology that effectively corrects errors in ETo forecasts, and adds extra skills to statistical calibration. I believe this new post-processing strategy could benefit future NWP-based ETo forecasting. To improve this work, the authors should pay special attention to the following key issues:

**Response: We appreciate the reviewer's insightful comments. We also believe the findings of this work could contribute to improving future NWP-based ETo forecasting. We address your constructive comments thoroughly and carefully and believe this work has been improved significantly. Please find more details in our point-by-point response.**

**Point #2**

1, Presentation of the results could be improved. Currently, the authors use maps to show/compare results from different model experiments. These figures could demonstrate the spatial patterns of modeling results. However, it might be more useful if the authors could summarize regional results in a different way, such as using boxplots. I believe that will better show readers the overall statistical information across the whole country than simply plotting the results as maps.

**Response: Thank you for the valuable suggestions. We create boxplots for all the maps shown in the main text. Since we already have eight figures in the main text and seven figures in the supplementary material, we think it is better not to add too many new figures. We combine these new boxplots with maps for Figures 2-7, which have extra zoom for adding new subplots. For Figures 1 and 7, which already include many subplots, we present the corresponding boxplots in the Supplementary Material. Please find the boxplots as follows:**

[Figure]

**Figure 2 Boxplot summarizing bias in calibrated ETo forecasts**

[Figure]

**Figure 3 Boxplot summarizing differences in absolute bias between calibrated ETo forecasts from**
                    **Calibration 2 with Calibration 1**

[Figure]

**Figure 4 Boxplot summarizing the alpha index in the calibrated ETo forecasts**

[Figure]

**Figure 5 Boxplot summarizing correlation coefficient between calibrated ETo forecasts from Calibration 2 and AWAP ETo data**

[Figure]

**Figure 6 Boxplot summarizing differences in the correlation coefficient (calibrated forecasts vs.**
**AWAP ETo) between Calibrations 2 and 1**

[Figure]

**Figure 8 Boxplot summarizing differences in CRPS skill scores between the calibrated forecast**
**from Calibration 2 with those from Calibration 1**

[Figure]

*Figure S7. Boxplot of biases in raw ETo forecasts constructed without bias-corrected input*
*variables (pink) and correct inputs (blue)*

[Figure]

*Figure S9. Boxplot of CRPS skill score in raw (pink) and calibrated ETo forecasts (blue) from*
*Calibration 2*

**Point #3**

2, Implications for ETo forecasting at the monthly or seasonal scales should be further discussed. ETo forecasting based on monthly or seasonal climate forecasts from GCMs is also widely performed. This study develops the new strategy for short-term forecasts. The applicability of this method to ETo forecasting based on GCM forecasts should be briefly discussed, to benefit a broader range of readers.

**Response: We agree with the reviewer that ETo forecasting with longer forecast horizons (e.g., monthly and seasonal) based on GCM forecasts is increasingly performed, and it is necessary to evaluate whether the post-processing strategy developed in this investigation is applicable to the GCM-based seasonal ETo forecasting. As we have shown in this manuscript, the reduction of error propagation from the input variables to ETo is the key reason why the new strategy has better performance than the original strategy (no improvement to raw forecasts of input variables). We expect this will be the case for GCM-based seasonal forecasts. However, testing this idea will be beyond the scope of this current study. To highlight the necessity of adopting this strategy in seasonal ETo forecasting, we add the following paragraph to section 4.2 (Implications for forecasting of integrated variables and future work):**

" In addition, seasonal ETo forecasting based on GCM climate forecast has been increasingly performed (Tian et al., 2014; Zhao et al., 2019b). In these investigations, raw ETo forecasts were also constructed with raw GCM forecasts. As a result, it is unavoidable that these investigations have suffered from error propagation from input variables to seasonal ETo forecasts. We expect that the calibration strategy (strategy ii) tested in this study will be applicable to seasonal ETo forecasting, considering its capability in reducing errors that could not be corrected through statistical calibration. Further investigations are needed to examine how the bias-correction of raw forecasts of input variables will affect the calibration of GCM-based seasonal ETo forecasts."

**Point #4**

Specific comments:

Line 20, rewrite this sentence. Not clear

**Response: we replace the original sentence:**

"This calibration strategy is expected to enhance future NWP-based ETo forecasting."

**with**

" We anticipate that future NWP-based ETo forecasting will benefit from adopting the calibration strategy developed in this study to produce more skillful ETo forecasts. "

**Point #5**

Line 74 Calibrate->calibrate

**Response: We correct the word accordingly.**

## Point #6

Line 80 compiled as the inputs…..

**Response: We improve the sentence of:**

"Weather forecasts from the ACCESS-G2 model are compiled to generate ETo forecasts."

**with:**

"Weather forecasts from the ACCESS-G2 model are extracted as inputs for the calculation of ETo
forecasts."

## Point #7

Line 95 10m -> 10 m.

**Response: We add a space between the number and the unit. We also check the entire
manuscript to correct this issue.**

## Point #8

Line 107-108, need to clarify what the anomaly and climatological mean are referring to

**Response: To clarify how the anomaly and climatological mean are derived, we replace the
sentence:**

"Our recent investigation suggests that ETo forecast calibration based on anomaly and climatological
mean produces more skillful calibrated forecasts than calibrating ETo forecasts directly."

**with:**

"Our recent investigation suggests calibrating anomalies of raw forecasts, which are calculated as the
departure from the observed climatological mean, could produce more skillful calibrated forecasts than
calibrating ETo forecasts directly."

## Point #9

Line 165 consider rewriting this sentence. Does not read well.

**Response: We replace the original sentence of**

"Once we obtain all the parameters for the BN distribution (equation 4), a conditional distribution is
established for $o(t)$ when a raw forecast ($f(t)$) is provided."

**with:**

"With the optimized parameters (means, standard deviations, and correlations) for the BN distribution
(equation 4), a conditional distribution for o(t) for a given raw forecast (f(t)) is derived "

**Point #10**
Line 172, what is specific month

**Response: we replace "specific" with "unselected" to make the wording more specific.**

**Point #11**
Figures in Results: shouldn't the figures be centralized?

**Response: The original format following a template from HESS. After we add boxplots to**
**these maps, the empty space for each figure is significantly reduced. We keep them aligned**
**to the left to be consistent with the provided template.**

**Point #12**
Line 360, not calibrate directly, should be without correcting forecasts of the inputs

**Response: Thank you for the suggestion. The key message we want to present here is that**
**statistical models may not be able to correct all errors in integrated variables (such as ETo).**
**However, when the input variables are corrected first, error propagation from inputs to**
**integrated variables, particularly for the errors which could not be corrected by calibration**
**models, will be reduced. To make it clear, we improved the original sentence of:**

"Our investigation suggests that improving the input variables may help correct errors that could not be
fixed when calibrating the integrated variables directly."

**with:**

" Our investigation suggests that improving the input variables could reduce error propagation from
inputs to integrated variables, and particularly reduce errors that could not be corrected by the calibration
model. "

**Point #13**

Line 365, consider rewriting this sentence

**Response: Thank you for the suggestion. We replace the original sentence:**

"As a result, using a more sophisticated calibration method to correct errors in input variables, is expected to further improve forecasts of these input variables, resulting in more significant improvements in the final calibrated ETo forecasts"

**with:**

" If a more sophisticated calibration method is employed to correct errors in input variables, error propagation from input variables to ETo forecasts will likely be further reduced. As a result, we anticipate that the calibrated ETo forecast will gain further improvements in forecast skills."

**Point #14**

Line 377-378, two' calibration models' consider to rewrite

**Response: We improve the original sentence:**

"Additional investigations using other calibration models will help clarify whether the improvements will hold for other calibration models."

**With**

" Additional evaluations using other calibration models will be needed to ascertain whether the improvements will be achieved when the calibration is conducted with a different model. "

**Point #15**

Line 385, in the calibrated forecasts

**Response: We add the missing 'in' to this sentence.**

**Point #16**

Line 386, consider making it shorter and clearer

**Response: We improve the following sentence:**

"Further investigation indicates that the contribution of improving input variables to the ETo forecasting tends to be independent of the calibration method applied to raw ETo forecasts."

**With**

"Further investigation indicates that the improvements tend to be independent of the calibration method
applied to raw ETo forecasts."

---

## Author Comment (AC3)

**Responses to Reviewer #2**

**Point #1**

Comments on âBias-correcting individual inputs prior to combined calibration leads to more skillful forecasts of reference crop evapotranspirationâ by Yang et al. This study evaluated two calibration strategies for simulating reference crop evapotranspiration. The two strategies are (1) calibration directly applied to raw ETo forecast constructed with raw forecast of input variables; (2) bias-correcting input variables. The bias-correcting algorithm has been proved to be more feasible. Although this study is of significance, improvements and revision can make the study stronger and more compelling.

**Response: We appreciate the reviewer's insightful suggestions and comments on the manuscript. We address concerns from the reviewer carefully and improve the manuscript accordingly.**

**Point #2**

Core of my concerns is the results presentation and discussion, many sections are superficial; the results are simply described, more insightful explanation and discussion are needed. See below for my suggestion. A moderate revision can easily address these comments. So I suggest a moderate revision.

**Response: We appreciate the reviewer's constructive comments. We improved the analysis and presentations by (1) creating boxplots to summarize results plotted as maps to better demonstrate results quantitatively, (2) performing statistical analyses (t-test) when comparing results from different model runs, (3) providing more statistical information in the Results section, and (4) Comparing findings of this work with published investigations. We further explain these improvements in detail as follows:**

**(1) Adding boxplots to Results**
**We created boxplots for results shown as maps (Figures 1 to 8 in the main text). We combine these boxplots with maps for Figures 2-7, which have extra zoom for adding new subplots. For Figures 1 and 7, which already include many subplots, we present the corresponding boxplots in the Supplementary Material. We also update the main text accordingly. Please find the boxplots as follows:**

[Figure]

**Figure 2 Boxplot summarizing bias in calibrated ETo forecasts**

[Figure]

**Figure 3 Boxplot summarizing differences in absolute bias between calibrated ETo forecasts from Calibration 2 with Calibration 1**

[Figure]

  **Figure 4 Boxplot summarizing the alpha index in the calibrated ETo forecasts**

[Figure]

**Figure 5 Boxplot summarizing correlation coefficient between calibrated ETo forecasts from**
                **Calibration 2 and AWAP ETo data**

[Figure]

**Figure 6 Boxplot summarizing differences in the correlation coefficient (calibrated forecasts vs. AWAP ETo) between Calibrations 2 and 1**

[Figure]

**Figure 8 Boxplot summarizing differences in CRPS skill scores between the calibrated forecast from Calibration 2 with those from Calibration 1**

[Figure]

*Figure S12. Boxplot of biases in raw ETo forecasts constructed without bias-corrected input*
*variables (pink) and correct inputs (blue)*

[Figure]

*Figure S14. Boxplot of CRPS skill score in raw (pink) and calibrated ETo forecasts (blue) from*
*Calibration 2*

**(2) Conducting t-test to compare results from different Calibrations.**

**We conduct t-tests to further evaluate the performance of the two calibration strategies.**
**Specifically, T-tests were conducted in the evaluation of bias, correlation coefficient, and**
**CRPS skill score (figures 1, 2, 3, 6, 7, 8) of the raw or calibrated forecasts (Table S2). In**
**addition, we also conducted t-tests (Table S1) to evaluate raw forecasts of the five input**
**variables (Figures S2 to S6).**

**In the calculation of $t$ statistics, we used the Spatial Degrees of Freedom (SDOF), rather than**
**using the total grid cells in the study area, to account for the spatial correlation in the t-test.**
**The SDOF is substantially smaller than total grid cells** (Toth, 1995)**.** Wang and Shen (1999)
**investigated SDOF of GCM outputs and reported a range of 90-120, out of 738 grid cells for**
**the southern hemisphere. In this study, we use 50 as the SDOF for our t-tests. Considering the**
**large amount of total grid cells (281,622) in this study, we believe that 50 is a conservative**
**estimate of SDOF for this investigation. We calculated the *t-statistics* and evaluate whether**
**they are statistically significant using the SDOF of 50. Results of the t-tests (Tables S1 and S2)**
**are added to the supplementary material.**

**Reference:**

Toth, Z.: Degrees of freedom in Northern Hemisphere circulation data, Tellus, Ser. A, 47 A(4),
457–472, doi:10.3402/tellusa.v47i4.11531, 1995.

Wang, X. and Shen, S. S.: Estimation of spatial degrees of freedom of a climate field, J. Clim.,
12(5 I), 1280–1291, doi:10.1175/1520-0442(1999)012<1280:EOSDOF>2.0.CO;2, 1999.

**Table S1 Results of t-tests** (*t-statistic*) **for raw forecasts of input variables**

| Tests

Lead times | Test if bias in raw Tmax forecasts is different from zero (Figure S2) | Test if bias in raw Tmin forecasts is different from zero (Figure S3) | Test if bias in raw vapor pressure forecasts is different from zero (Figure S4) | Test if bias in raw solar radiation forecasts is different from zero (Figure S5) | Test if bias in raw wind speed forecasts is different from zero (Figure S6) |
|---|---|---|---|---|---|
| Day 1 | -8.96** | 1.66 | -3.18** | 11.83** | 16.04** |
| Day 2 | -8.16** | 2.65** | -3.43** | 11.39** | 16.50** |
| Day 3 | -8.19** | 2.68** | -3.77** | 11.81** | 16.57** |
| Day 4 | -8.12** | 2.56** | -4.05** | 12.17** | 16.56** |
| Day 5 | -7.87** | 2.41** | -4.09** | 12.45** | 16.45** |
| Day 6 | -7.70** | 2.27** | -4.21** | 11.88** | 16.45** |
| Day 7 | -7.73** | 2.22** | -4.33** | 10.81** | 16.29** |
| Day 8 | -7.70** | 2.17** | -4.30** | 11.41** | 16.56** |
| Day 9 | -7.44** | 2.20** | -4.18** | 11.95** | 16.82** |

**The Spatial Degrees of Freedom (SDOF) is 50 in the tests; ** indicates statistically significant differences at the 95%**
**confidence interval.**

**Table S2 Results of t-tests (*t-statistic*) for performance evaluation**

| Tests / Lead times | Comparison of bias in raw ETo forecasts constructed with vs. without bias correction (Figure 1) | Test if bias in calibrated ETo forecasts from Calibration 2 (Figure 2) is different from zero | Test differences in absolute bias between calibrated ETo forecasts from Calibrations 2 and 1 (Figure 3) | Test difference in $r$ between observations and calibrated ETo forecasts from Calibrations 2 and 1 (Figure 6) | Comparison of CRPS skill score between raw and calibrated ETo forecasts (Figure 7) | Test difference in CRPS skill score of calibrated ETo forecasts from Calibrations 2 and 1 (Figure 8) | Test difference in α-index between Calibrations 2 and 1 (Figure S8) | Test difference in CRPS skill scores between Calibrations 3 and 4 (Figure S10) |
|---|---|---|---|---|---|---|---|---|
| Day 1 | -9.76** | 1.80 | -4.08** | 5.73** | 27.59** | 11.53** | -0.54 | 11.53** |
| Day 2 | -9.86** | 1.91 | -3.93** | 4.93** | 29.03** | 10.86** | -1.47 | 10.86** |
| Day 3 | -9.86** | 2.07** | -3.68** | 4.43** | 31.14** | 9.77** | -1.81 | 9.77** |
| Day 4 | -9.81** | 2.27** | -3.54** | 4.01** | 33.77** | 8.58** | -1.17 | 8.58** |
| Day 5 | -9.71** | 2.40** | -3.36** | 3.75** | 38.11** | 7.16** | -2.09** | 7.16** |
| Day 6 | -9.54** | 2.60** | -3.37** | 3.17** | 42.59** | 6.44** | -1.28 | 6.44** |
| Day 7 | -9.34** | 2.76** | -3.26** | 2.69** | 44.38** | 6.15** | -1.99 | 6.15** |
| Day 8 | -9.04** | 2.98** | -3.13** | 2.32** | 45.57** | 5.85** | -1.57 | 5.85** |
| Day 9 | -9.21** | 3.13** | -2.91** | 1.85 | 51.91** | 5.05** | -1.70 | 5.05** |

**The Spatial Degrees of Freedom (SDOF) is 50 in the tests; ** indicates statistically significant differences at the 95% confidence interval.**

**(3) Improving the Results section**

**We add more specific information in describing the key findings of this study and introduce the results of the statistical analyses (Tables S1 and S2). Since we modified many sentences, we decide not to list them here. Please see details in the revised manuscript.**

**(4) Improving the Discussion section**

**We further compare the findings of this investigation with related studies in discussion:**

"In the ETo forecasting across 40 Australia, although raw ETo forecasts constructed with NWP outputs reasonably captured the magnitude and variability of ETo, forecast skills better than climatology were only found for the first 6 lead times (Perera et al., 2014). Our investigation suggests that statistical calibration could substantially improve forecast skills and outperform the climatology forecasts for all 9 lead times across Australia. The findings of this investigation agree well with the site scale short-term ETo forecasting based on GCM outputs (Zhao et al., 2019a) in improving forecast skills. Calibrated forecasts from Calibration 2 demonstrate similar skills as those of Zhao et al. (2019a). However, our calibration achieves the improvements using much shorter archived raw forecasts (3-year vs. 23-year) than Zhao et al. (2019a), thanks to the capability of SCC in calibrating short-archived forecasts (Wang et al., 2019). Calibrated forecasts from Calibration 2 also demonstrate comparable biases (0.32-0.95%) with calibrated ETo forecasts (0.49-0.63%) in the U.S. based on the Bayesian model averaging (BMA) model and weather forecasts from three NWP models during 2014-2016."

**Point #3**

Lines 11, fully implemented.

**Response: we change it to 'fully implemented '.**

**Point #4**

Line 27, â□□□divergentâ□□□ emphasizes completely different assumption, you can just use replace it different to ensure a general term.

**Response: We replace the word 'divergent' with 'different'.**

**Point #5**

Line 38, physical processes of the atmosphere, it is unclear, atmospheric circulation or atmospheric wind formation, or physical processes in the atmosphere

**Response: Thank you for the suggestion. We change the sentence as follows:**

"ETo is affected jointly by temperature, vapor pressure, solar radiation, and wind speed (Bachour et al., 2016; Luo et al., 2014). Prediction models using these weather variables as inputs allow for representations of atmospheric dynamics and often produce reasonable ETo predictions (Torres et al., 2011)."

**Point #6**

Section 3.1, 3.2, the authors described the results in the figures. However, most of those text are vague, please provide more specific (quantitative) information to support your statement. When you compare different results or method, it is better to report some statistic results (p value, r2, etc).

**Response: We conduct statistical analysis to quantify the difference between different model runs, and update the Results sections accordingly. Details of the t-tests could be found in our response to your comments point #1.**

**Point #7**

for example, line line 223-225, you report the overprediction in Tmax, and underpredict in Tmin in different regions. If it is underprediction, what is the range of that underprediction, same for overprediction, are these different statistically significant? There are many similar issues in other sections.

**Response: We appreciate the reviewer's valuable suggestions. The reason we did not introduce errors in raw forecasts of the input variables in detail is that systematic errors in raw NWP forecasts have been well documented. Evaluation of the raw forecasts of the inputs is not the key information we want to deliver in this study. However, we agree with the reviewer that more statistical information is needed. We conduct statistical analysis to quantify errors in raw forecasts (Table S1), and update contents in Results accordingly. Statistical analyses could be found in our response to your comment #1. Here is the updated description of errors in raw forecasts:**

"Raw forecasts of the five input variables demonstrate significant inconsistencies with the corresponding AWAP data (Figures S2-S6). In most parts of Australia, daily maximum temperature (Tmax) forecasts are lower than AWAP data by 1-2 °C. Overpredictions in Tmax are only found in coastal areas of northwestern Australia. The daily minimum temperature (Tmin) is underpredicted by more than 1.5 °C in western and central parts of Australia by the raw forecasts, but is overpredicted by ca. 1 °C in eastern and southern Australia. Forecasted wind speed is higher than the reference data by more than 1m/s (or by 50%) in most parts of Australia. Similarly, raw solar radiation forecasts are about 5% higher than AWAP data across Australia. Vapor pressure is underpredicted in western and central regions by ca.14%, but is overpredicted by ca. 6% in coastal areas of south-eastern Australia by the raw forecasts. For each of the five variables, spatial patterns of biases in raw forecasts are consistent across the 9 lead times, demonstrating systematic errors in the raw NWP forecasts. According to our statistical test, overpredictions or unerpredictions in raw forecasts of the input variables are statistically significant $(P<0.05)$ for most lead times (Table S1). "

Point #8

In the discussion section, I would be willing to see a comparison with other studies with different algorithms for the ETo simulation. Some quantitative comparison to elucidate the better performance of the new bias-correction algorithm needs to be done. I believe it will prove the reliability of the new algorithm.

**Response: We appreciate the constructive comments. This is the first continental-scale ETo forecasting in Australia. Previous NWP/GCM-based ETo forecasting in Australia is conducted at the site scale. As a result, in the original manuscript, our evaluation was primarily focused on the comparison against observations. In this area of weather/climate forecasting, different calibration models, based on different statistical theories, have been developed and implemented. Previous comparisons suggest that the performance of these models varied with study areas, NWP models, and choice of evaluation metrics (Wilks, 2018), and there is no conclusion regarding which group of post-processing models has the best performance.**

**More importantly, rather than developing a new calibration model, this investigation is to evaluate the necessity of including an extra step before forecasts are calibrated. As we introduced in the maint ext, the objective of our investigations is to address a common challenge faced by NWP-based ETo forecasting, and we expect the calibration strategy developed in this study will benefit ETo forecast calibrations, no matter which statistical model is employed.**

**However, we agree with the reviewer that comparison of model performance with other models will help readers better understand the reliability of this work. We review previous studies and add the following content to Discussion (section 4.1):**

"In the ETo forecasting across 40 Australia, although raw ETo forecasts constructed with NWP outputs reasonably captured the magnitude and variability of ETo, forecast skills better than climatology were only found for the first 6 lead times (Perera et al., 2014). Our investigation suggests that statistical calibration could substantially improve forecast skills and outperform the climatology forecasts for all 9 lead times across Australia. The findings of this investigation agree well with the site scale short-term ETo forecasting based on GCM outputs (Zhao et al., 2019a) in improving forecast skills. Calibrated forecasts from Calibration 2 demonstrate similar skills as those of Zhao et al. (2019a). However, our calibration achieves the improvements using much shorter archived raw forecasts (3-year vs. 23-year) than Zhao et al. (2019a), thanks to the capability of SCC in calibrating short-archived forecasts (Wang et al., 2019). Calibrated forecasts from Calibration 2 also demonstrate comparable biases (0.32-0.95%) with calibrated ETo forecasts (0.49-0.63%) in the U.S. based on the Bayesian model averaging (BMA) model and weather forecasts from three NWP models during 2014-2016."

**In addition, we also highlight the importance of testing the proposed calibration strategy (strategy ii) based on other calibration models in the future in section 4.2:**

"Second, further investigations based on other calibration models are needed to validate the conclusions of this investigation. Our analyses based on two different methods (based on ETo anomalies vs. based on original ETo) find similar improvements in calibrated ETo forecasts following bias-correction of input variables. Additional tests using other calibration models will be needed to evaluate whether the improvements will be achieved when the calibration is conducted with a different model."

**Reference:**

Wilks, D.S., 2018. Chapter 3. Univariate Ensemble Forecasting, in: Vannitsem, S., Wilks, D.S., Messner, J.W. (Eds.), Statistical Postprocessing of Ensemble Forecasts. pp. 49–89. https://doi.org/https://doi.org/10.1016/C2016-0-03244-8

**Point #9**

Line 388, feasible or reliable ETo forecasting.

**Response: This paragraph has been rewritten. Please see the revised contents in our response to your comment #10.**

**Point #10**

 Line 390, short-term ETo forecasting provides highly valuable information for real-time decision making on water resource management and planning farming practices. This study proved the bias-correction approach is a feasible method for a more robust calibration of the NWP-based ETo forecasting.

**Response: We appreciate the reviewer's valuable suggestions. We remove redundant sentences and combine the last two paragraphs in the Conclusion section:**

"This investigation clearly suggests the necessity of improving input variables as part of the NWP-based ETo forecasting. With this extra step, the bias, correlation coefficient, and skills in the calibrated ETo forecasts are all improved, particularly for the short-lead-time forecasts. Further investigation indicates that the improvements tend to be independent of the calibration method applied to raw ETo forecasts. Forecasting the highly variable ETo is often challenging. Our investigation provides an effective calibration strategy for improving NWP-based ETo forecasting. As a result, we anticipate that future calibration of NWP-based ETo forecasts could benefit from adopting this strategy to produce skillful calibrated ETo forecasts. This strategy is also expected to be applicable to enhancing the forecasting of other integrated variables that are calculated using multiple NWP/GCM variables."

---

## Author Comment (AC4)

**Responses to Reviewer #3**

**Point #1**

*Author(s): Qichun Yang et al.*

*MS No.: hess-2021-69*

*This paper focuses on the comparison of two calibration strategies to provide short-term reference crop evapotranspiration (ETo). ETo forecasting is still a relatively new area of research, in Australia and elsewhere, and has received more attention in the past few years. Skilful ETo forecasts in Australia would help support efficient water use and water management. Two strategies to calibrate ETo forecasts have emerged: i) the calibration of raw ETo forecasts and ii) bias-correcting input variables first before calibrating ETo forecasts. Little work to date compares the two approaches, it is unclear which method might be more advantageous or skilful. This paper therefore addresses a topical subject with a large audience interest.*

*I have some reservations regarding some methodological choices and justifications (purpose and inclusion of experiment 3 and 4), as well as a lack of interpretations of the results overall. I recommend revision to strengthen this paper.*

**Response: Thank you for the valuable suggestions and careful review. We revise this work carefully based on your constructive suggestions.**

**Point #2**

*The authors re-grid the weather forecast variables of ACCESS-G2 to match the timeframe and resolution of the gridded data AWAP. They perform four experiments: experiments 1) and 2) are based on the ETo anomaly and climatological mean, whereas experiment 3 and 4) use the ETo values directly. Furthermore, experiment 1) and 3) use raw inputs to calculate and calibrate ETo forecasts whereas experiments 2) and 4) first bias-correct inputs before ETo calibration. The SCC calibration method is used for ETo forecast while a quantile mapping method is used to bias-correct input forecasts. The authors evaluate the forecasts using three metrics for the theoretical assessment of bias, reliability and accuracy. Overall results suggest that the second strategy (bias-correction of inputs before ETo calibration) provides more skilful forecasts.*

**Response: We appreciate the reviewer's thorough review. The work has been substantially improved based on the valuable comments.**

Point #3

*Major comments:*

*Methodology:*

*P4 section 2.3: Why not compare the calibration method used SCC to other methods tested in the literature which would enable to place this work in context to other studies on ETo forecasting?*

**Response: We appreciate the constructive comments. We understand that comparing the performance of SCC with existing methods will help readers better understand the strengths of SCC. We did not compare the SCC model directly with other models in the original submission for a couple of reasons:**

**First, this investigation addresses a common challenge faced by NWP-based ETo forecasting, rather than developing a new calibration model for ETo forecasting. The primary objective of this investigation is to evaluate the necessity of correcting forecasts of input variables prior to calibrating ETo forecasts. As we introduced in the main text, the calibration strategy developed in this study is expected to benefit ETo forecast calibrations broadly, rather than improving an individual model. As suggested by the model experiments in our investigation (Calibrations 1-4), the developed strategy could potentially be applied to other calibration models.**

**Second, we feel it is not necessary to compare the performance of SCC against calibration models with widely used but less sophisticated algorithms. Simple calibration models, such as quantile mapping (QM), have been widely used in calibrating hydroclimate forecasts. These models are often readily available, or could be easily coded and implemented. However, the limitations of these models have been reported (Zhao et al., 2017). When we started this investigation, we used quantile mapping to calibrate ETo forecasts (raw ETo forecasts constructed with raw forecasts of input variables). As demonstrated in the following figure, the CRPS skill score of quantile mapped ETo forecasts is not only lower than the SCC-calibrated forecasts for each corresponding lead time, but also becomes negative (worse than climatological forecasts) in parts of Australia starting from lead time 4. As a result, calibration of ETo forecasts with quantile mapping further confirms the limitations of this model. Using such models as a reference to evaluate the performance of SCC is not fair, since their limitations have been reported. As a result, we decide not to include a comparison with quantile mapping in this manuscript.**

[Figure]

*CRPS skill score of calibrated ETo forecasts using Quantile Mapping*

Third, we have limited access to sophisticated calibration models. There is no global post-processing software library archiving these models. As a result, we found it was hard to access the source code of these models and to directly compare SCC with them. In addition, previous comparisons suggest that the performance of these models varied with study areas, NWP models, and choice of evaluation metrics (Wilks, 2018), and there is no conclusion regarding which group of post-processing models has the best performance. Our indirect comparison with other models confirms the above study. Details will be presented in the following paragraphs.

**Fourth, the short-achieved NWP forecasts (3-year) used in this study represent a challenge for conducting the calibration using other models. Many calibration models, particularly those based on models of the joint probability of forecasts and observations** (Krzysztofowicz and Herr, 2001; Wang and Robertson, 2011)**, require long hindcasts (20-30 years) to establish a joint distribution to link observations and forecasts. Applying such models to short-archived forecasts such as those used in this study will substantially undermine the statistical assumption of these models. The advantages of SCC in calibrating short-archived forecast has been explained in our recent publications** (Wang et al., 2019; Yang et al., 2021)**.**

**As a result, we did not compare SCC directly with other models. However, we totally agree with the reviewer that comparison of model performance with other models will help readers better understand the reliability of this work. For example, we extract our results at three sites in Australia where ETo forecasts were also calibrated based on the Bayesian joint probability (BJP) model** (Zhao et al., 2019)**, and compare the results of the two investigations. In addition, we also compare our results with investigations in other regions of Australia and the U.S. We add the following paragraph to discuss findings of our work relative to existing investigations to the Discussion section (section 4.1):**

**"**This investigation further highlights the importance of statistical calibration in improving the quality of raw ETo forecasts (Medina and Tian, 2020). In the ETo forecasting across 40 sites in Australia, although raw ETo forecasts constructed with NWP outputs reasonably captured the magnitude and variability of ETo, forecast skills better than climatology were only found for the first 6 lead times (Perera et al., 2014). Our investigation suggests that statistical calibration could substantially improve forecast skills and outperform the climatology forecasts for all 9 lead times across Australia. The findings of this investigation agree well with the site scale short-term ETo forecasting based on GCM outputs (Zhao et al., 2019a) in terms of improvements in forecast skills. Calibrated forecasts from Calibration 2 demonstrate similar skills as those of Zhao et al. (2019a). However, our calibration achieves the improvements using much shorter archived raw forecasts (3-year vs. 23-year) than Zhao et al. (2019a), thanks to the capability of SCC in calibrating short-archived forecasts (Wang et al., 2019). Calibrated forecasts from Calibration 2 also demonstrate comparable biases (0.32-0.95%) with calibrated ETo forecasts (0.49-0.63%) in the U.S. based on the Bayesian model averaging (BMA) model and weather forecasts from three NWP models during 2014-2016 (Medina and Tian, 2020).**"**

**In addition, we also highlight the importance of further testing the proposed calibration strategy (strategy ii) based other calibration models. We add the following content to section 4.2:**

**"**Second, further investigations based on other calibration models are needed to validate the conclusions of this investigation. Our analyses based on two different methods (based on ETo anomalies vs. based on original ETo) find similar improvements in calibrated ETo forecasts with the adoption of bias-correction of input variables. Additional evaluations using other calibration models will be needed to ascertain whether the improvements will be achieved when the calibration is conducted with a different model.**"**

**Reference:**

Medina, H. and Tian, D.: Comparison of probabilistic post-processing approaches for improving numerical weather prediction-based daily and weekly reference evapotranspiration forecasts, Hydrol. Earth Syst. Sci., 24, 1011–1030, 2020.

Perera, K. C., Western, A. W., Nawarathna, B. and George, B.: Forecasting daily reference evapotranspiration for Australia using numerical weather prediction outputs, Agric. For. Meteorol., 194, 50–63, doi:10.1016/j.agrformet.2014.03.014, 2014.

Wilks, D.S., 2018. Chapter 3. Univariate Ensemble Forecasting, in: Vannitsem, S., Wilks, D.S., Messner, J.W. (Eds.), Statistical Postprocessing of Ensemble Forecasts. pp. 49–89. https://doi.org/https://doi.org/10.1016/C2016-0-03244-8

Krzysztofowicz, R., Herr, H.D., 2001. Hydrologic uncertainty processor for probabilistic river stage forecasting: precipitation-dependent model. J. Hydrol. 249, 46–68.

Wang, Q.J., Robertson, D.E., 2011. Multisite probabilistic forecasting of seasonal flows for streams with zero value occurrences. Water Resour. Res. 47, 1–19. https://doi.org/10.1029/2010WR009333

Wang, Q.J., Zhao, T., Yang, Q., Robertson, D., 2019. A Seasonally Coherent Calibration ( SCC ) Model for Postprocessing Numerical Weather Predictions. Mon. Weather Rev. 147, 3633–3647. https://doi.org/10.1175/MWR-D-19-0108.1

Yang, Q., Wang, Q.J., Hakala, K., 2021. Achieving effective calibration of precipitatioAn forecasts over a continental scale. J. Hydrol. Reg. Stud. 35, 100818. https://doi.org/10.1016/j.ejrh.2021.100818

Zhao, T., Wang, Q.J., Schepen, A., 2019. A Bayesian modelling approach to forecasting short-term reference crop evapotranspiration from GCM outputs. Agric. For. Meteorol. 269–270, 88–101. https://doi.org/10.1016/j.agrformet.2019.02.003

**Point #4**

*Presentation of summary statistics. Why not use boxplots to present overall statistics and across lead times (for example next to figure 4 and so on)? Reliability diagrams for particular ETo thresholds would be helpful to communicate when the forecasts are reliable.*

**Response: Thank you for the constructive suggestions. We created boxplots for results shown as maps (Figures 1 to 9 in the main text). For Figures 1 and 7, which already include many subplots, we present the corresponding boxplots in the Supplementary Material. For other map figures (Figures 2-6, and 8-9), which have extra zoom for adding new subplots, we combine these boxplots with maps. We also update the main text accordingly. Please find the boxplots as follows:**

[Figure]

**Figure 2 The boxplot summarizing improvements in correlation coefficient between raw ETo forecasts and AWAP ETo with the adoption of bias-correction to input variables**

[Figure]

**Figure 3 The boxplot summarizing bias in calibrated ETo forecasts from Calibration 2**

[Figure]

**Figure 4 The boxplot summarizing differences in absolute bias between calibrated ETo forecasts from Calibration 2 with Calibration 1**

[Figure]

**Figure 5 The boxplot summarizing correlation coefficient between calibrated ETo forecasts from Calibration 2 and AWAP ETo data**

[Figure]

**Figure 6 The boxplot summarizing differences in the correlation coefficient (calibrated forecasts vs. AWAP ETo) between Calibrations 2 and 1**

[Figure]

**Figure 8 The boxplot summarizing differences in CRPS skill scores between the calibrated forecast from Calibration 2 with those from Calibration 1**

[Figure]

**Figure 9 The boxplot summarizing the alpha index in the calibrated ETo forecasts**

[Figure]

*Figure S12. The boxplot of biases in raw ETo forecasts constructed without bias-corrected input variables (pink) and correct inputs (blue)*

[Figure]

*Figure S14. The boxplot of CRPS skill score in raw (pink) and calibrated ETo forecasts (blue)*

 **We also created reliability diagrams to summarize to evaluate the calibrated ensemble forecasts from Calibration 2. The three thresholds used to generate the reliability diagram are 3 mm/day, 6mm/day, and 9 mm/day. This diagram (Figure 10) is added to the main text to further evaluate the reliability of calibrated ETo forecasts**

[Figure]

**Figure 10: Reliability diagrams of calibrated ETo forecasts during 4/2016-3/2019 with thresholds of 3, 6, and 9 mm day$^{-1}$.**

**We updated the Method section to introduce how the reliability diagram is created and how to understand the diagram:**

"We evaluate the reliability of calibrated ETo forecasts from calibration 2 using the reliability diagram (Hartmann et al., 2002), which assesses how well the predicted probabilities of an event corresponding to their observed frequencies. We convert the calibrated ensemble ETo forecasts to forecast probabilities exceeding three thresholds, including 3, 6, and 9 mm day$^{-1}$. We pool forecasts of different grid cells, days, and lead times together in the calculation of forecast probability. In the reliability diagram, perfectly reliable forecasts will demonstrate a curve along the diagonal. A plotted curve above the diagonal indicates underestimations and vice versa."

**We add the following sentence to section 3.5 (Reliability of calibrated ETo forecasts) to introduce the reliability diagram.**

"The reliability diagram further confirms the consistency between forecast probabilities and observed frequencies (Figure 10). The plotted curves based on three thresholds (3, 6, and 9 mm day$^{-1}$) are mainly distributed along the 1:1 line, indicating high reliability of calibrated ETo forecasts."

**Point #5**

*Authors present experiments 1-4 in the method but then only present some results one experiment 3) and 4) in the last section of results (CRPSS in 3.5). No explanation are provided of why calibration 3) and 4) are only briefly introduced. Why is there a big gap with no results on calibration 3) and 4) on the bias and reliability results? Could the authors please expand on the purpose of including these at all in? At p17 l350-354, 'a further evaluation based on a different way of implementing the calibration demonstrate similar improvements in calibrated ETo forecasts with the adoption of bias-correction to input variables'. Is the purpose of including experiment 3) and 4) to test the generalisation of the method? If so, it needs to be clearly stated and justified earlier.*

**Response: Thank you for the valuable comments. The reviewer is correct that adding calibrations 3 and 4 is to further evaluate that whether our strategy could be generally applied to future NWP-based ETo forecasting, and will the strategy be independent of calibration models. We further explain the reason by adding the following sentences to clarify why Calibrations 3 and 4 are included in this study in Method (section 2.3):**

"The comparison between Calibrations 1 and 2 is to investigate whether the bias-correction of input variables would further improve ETo forecasts when the calibration is conducted based on ETo anomalies and climatological mean. We also conduct additional calibrations which post-process ETo forecasts directly (Calibrations 3 and 4), to test whether the contribution of improving the input variables to ETo forecast calibration, if there is any, will depend on how ETo forecasts are calibrated (based on anomalies vs. based on original ETo forecasts). Calibrations 3 and 4 will help evaluate the feasibility of strategy ii for the general application in NWP/GCM-based ETo forecasting. Key steps of the four calibrations could

be found in the schematic diagram (Figure S1). In the main text, we primarily analyze results from Calibrations 1 and 2. Improvements with the adoption of bias-correction to input variables in Calibrations 3 and 4 are very similar to those of Calibrations 1 and 2 (see the Supplementary Material). To avoid redundancy, we present results from Calibrations 3 and 4 in the Supplementary Material.**"**

**In the original submission, we did not present all results from Calibrations 3 and 4 because these two calibrations were complementary for supporting findings from Calibrations 1 and 2. This is an extra step to further evaluate the robustness of the calibration strategy developed in this study. In addition, differences in bias, reliability, and correlation coefficient between Calibrations 3 and 4 are very similar to those between Calibrations 1 and 2. We thought it might be a bit redundant and may confuse readers if we present all results from Calibrations 3 and 4 in the main text. However, we also agree with the reviewer that it is necessary to present results from Calibrations 3 and 4 if readers may be interested in them. In the revised manuscript, we present them in the supplementary material (See the figures below), in order not to distract readers from understanding key objectives (e.g., the necessity of bias-correcting input variables prior to ETo calibration) of this investigation. Specifically, in addition to the figure showing improvements in CRPS skill score, we also add figures demonstrating differences in absolute bias (Figure S15), correlation coefficients (Figure S16), and alpha index (Figure S18) between Calibrations 3 and 4 in the Supplementary Material:**

[Figure]

*Figure S15.  Differences in absolute bias between Calibrations 3 and 4*

[Figure]

*Figure S16. Differences in correlation coefficient between Calibrations 3 and 4*

[Figure]

*Figure S18. Differences in alpha index between Calibrations 3 and 4*

**We add one new section in Results to introduce results from Calibrations 3 and 4**

**3.6 Results from Calibrations 3 and 4**

"We also compare the bias, reliability, correlation coefficient, and CRPS skill score of calibrated forecasts from Calibrations 3 and 4, to evaluate whether we can obtain similar improvements through the bias-correction of input variables if we conduct the ETo forecast calibration in a different way (without using climatological mean and anomalies). Results show that the adoption of bias-correction also leads to lower bias, higher correlation coefficient, and higher CRPS skill score in terms of magnitude, spatial patterns,

and trend along the lead times, when ETo forecasts are calibrated directly (Figure S10, and S12-S13). In addition, the alpha index was only slightly different between Calibrations 3 and 4 (Figure S11). This additional comparison further confirms the general applicability of strategy ii for enhancing NWP-based ETo forecasting."

**Point #6**

*Methodological choices for evaluation:*

*P7 l 180-185 : why choosing the absolute bias and over a relative measure e.g. percentage bias? This choice makes it difficult to compare the magnitude of the errors in the results across different variables and studies. For example, figure 1 shows a bias between -2 to 2mm/day which does not seem like much compared to other input variables such as precipitation. Figure 3 with a range of -0.1 to 0.1 seems very small. Conversely, percentages are used for the correlation coefficient in Figure 6 so why not use it for the bias?*

**Response: We appreciate the reviewer's valuable comments. Bias shows differences with the observed mean, and could be either positive or negative. Larger departures from mean, no matter the bias is positive or negative, suggest larger inconsistencies with observations. Using absolute bias will help measure the departure, rather than indicating overestimations or underestimations. As a result, using absolute bias, we can compare results from two different calibrations, with smaller absolute bias indicating closer to the mean, and thus suggesting better performance.**

**We agree with the reviewer that using percentages will make the results more comparable with other variables, or with other studies. As a result, we change the unit of bias in figures 1, S12, 3, 4 to percentage:**

[Figure]

**Figure 1: Bias in (three panels on the left) raw ETo forecasts constructed with raw forecasts of input variables and (three panels on the right) raw ETo forecasts constructed with bias-corrected input variables.**

[Figure]

*Figure S12. Boxplot of biases in raw ETo forecasts constructed without bias-corrected input variables (pink) and correct inputs (blue)*

[Figure]

**Figure 3: Bias in calibrated ETo forecasts of 9 lead times from Calibration 2, in which raw ETo forecasts are constructed with bias-corrected input variables. Maps on the left show the spatial patterns of bias, and the boxplot on the right summarizes results for all grid cells.**

[Figure]

**Figure 4: Differences in absolute bias between calibrated ETo forecasts from Calibration 2 with Calibration 1. Maps on the left show the spatial patterns of difference in absolute bias, and the boxplot on the right summarizes results for all grid cells.**

Point #7

*P8 l205-2015: why is climatology used as reference forecast for the skill score? In hydrological forecasting persistence is typically used for short lead times, whereas climatology would be used for longer lead times, see fore example (Pappenberger, Ramos et al. 2015). Could you please expand and justify the choice of reference forecast used and implication of interpretation of results?*

**Response: We really appreciate the reviewer's valuable suggestion and the introduction of this classic paper. We choose the climatology forecasts as the reference rather than using persistency for several reasons:**

**1, Climatology forecasts have been widely used as the reference in the calculation of CRPS skill score for short-term hydroclimate forecasts. One advantage of climatology forecasts is that it often has similar error across all lead times (Bennett et al., 2014), and will be useful to evaluate forecasts skills among different lead times. Therefore, climatology forecasts could be used to show to decreasing skills of the calibrated forecasts as lead time advances (Academies, 2014; Zhao et al., 2019).**

**2, Persistence is also a good reference, but it's been mainly used for the first two lead times. As demonstrated in figure 5 of Bennett et al. (2014), errors in persistency could increase quickly with lead time. As a result, multiple studies suggested that persistence could be good for skill discrimination for the short lead times (Pappenberger et al., 2015; Thiemig et al., 2015).**

**Since we investigate 9 lead times in this study, errors in persistency are expected to be large at long lead times. As a result, we think the use of climatology forecasts as the reference for the calculation of the CRPS skill score is acceptable.**

**We add the following sentence to section 2.4.4 (Skills of the raw and calibrated forecasts) to explain the use of climatology forecasts as the reference for the calculation of CRPS skill score**

"In the calculation of CRPS skill score, both climatology forecasts or the last observations (persistence) have been used as reference forecasts (Pappenberger et al., 2015; Thiemig et al., 2015). However, reference forecasts based on persistence are more suitable for evaluating the performance of forecasts shorter than two days. As a result, we choose climatology forecasts as the reference since errors in climate forecasts are similar among all lead times and thus could be used to evaluate the increasing errors in raw and calibrated forecasts as lead time advances."

**Reference:**

Academies, N.: The science of NOAA'S Operational Hydrologic Ensemble Forecast Service, Bull. Am. Meteorol. Soc., (January), 79–98, doi:10.1175/BAMS-D-12-00081.1, 2014.

Bennett, J. C., Robertson, D. E., Lal, D., Wang, Q. J., Enever, D., Hapuarachchi, P. and Tuteja, N. K.: A System for Continuous Hydrological Ensemble Forecasting (SCHEF) to lead times of 9 days, J. Hydrol., 519, 2832–2846, doi:10.1016/j.jhydrol.2014.08.010, 2014.

Pappenberger, F., Ramos, M. H., Cloke, H. L., Wetterhall, F., Alfieri, L., Bogner, K., Mueller, A. and Salamon, P.: How do I know if my forecasts are better? Using benchmarks in hydrological ensemble prediction, J. Hydrol., 522, 697–713, doi:10.1016/j.jhydrol.2015.01.024, 2015.

Thiemig, V., Bisselink, B., Pappenberger, F. and Thielen, J.: A pan-African medium-range ensemble flood forecast system, Hydrol. Earth Syst. Sci., 19, 3365–3385, doi:10.5194/hess-19-3365-2015, 2015.

Zhao, T., Wang, Q. J. and Schepen, A.: A Bayesian modelling approach to forecasting short-term reference crop evapotranspiration from GCM outputs, Agric. For. Meteorol., 269–270(January), 88–101, doi:10.1016/j.agrformet.2019.02.003, 2019.

**Point #8**

*P8 l214. Why is the definition of CRPSS using percentage? As far as I am aware, most studies do not present the CRPSS in terms of percentage, could you please comment on the reason of this choice with references that also use percentages and if there is any advantages?*

**Response: Thank you for the comments. We agree with the reviewer that many studies use ratios when presenting the CRPS skill score. Meanwhile, we also notice that some studies (see the reference list at the bottom of our response to this comment) use percentage as the unit of CRPS skill score. No matter which unit is used, CRPS skill score could effectively demonstrate higher skills in calibrated forecasts relative to the raw forecasts (Figure 7), and quantify improvements in forecast skill (Figures 8, S9, S12) with the adoption of the calibration strategy.**

**As shown in Figure 7, skills of calibrated forecasts decreased quickly with lead time. As a result, the CRPS skill score decreases to small numbers and approaches zero at lead time 9. One advantage of using percentages as the unit for CRPS skill score is that these small numbers will be expressed as integers rather than small decimals.**

**We add the following sentence to explain why the percentage is used for CRPS skill score:**

**"We use percentage as the unit of CRPS skill score so low skill scores at long lead times will be expressed as integers. "**

**Here are some investigations using % as the unit of CRPS skill score**

Brown, J. D. and Seo, D. J.: A nonparametric postprocessor for bias correction of hydrometeorological and hydrologic ensemble forecasts, J. Hydrometeorol., 11(3), 642–665, doi:10.1175/2009JHM1188.1, 2010.

Kumar, L. G. A., Smith, A. S. D., Gonzalez, G. B. P., Merryfield, V. K. W. and Newman, A. S. Á. M.: A verification framework for interannual-to-decadal predictions experiments, Clim. Dyn., 40, 245–272, doi:10.1007/s00382-012-1481-2, 2013.

Munkhammar, J., van der Meer, D. and Widén, J.: Probabilistic forecasting of high-resolution clear-sky index time-series using a Markov-chain mixture distribution model, Sol. Energy, 184(January), 688–695, doi:10.1016/j.solener.2019.04.014, 2019.

Robertson, D. E. and Wang, Q. J.: Seasonal Forecasts of Unregulated Inflows into the Murray River , Australia, Water Resour. Manag., 27, 2747–2769, doi:10.1007/s11269-013-0313-4, 2013.

Schepen, A., Wang, Q. J. and Robertson, D. E.: Seasonal Forecasts of Australian Rainfall through Calibration and Bridging of Coupled GCM Outputs, Mon. Weather Rev., 142, 1758–1770, doi:10.1175/MWR-D-13-00248.1, 2014.

**Point #9**

Analysis and interpretation of results:

*P11 l259-261: why the higher difference in bias in approaches for the Nothern Territory? How does this relate to the biases, errors and assumptions of the NWP? Is it correlated to the biases of specific input variables? How is it correlated to the nonlinear relationship in calculatint ETo? Why are the biases most pronounced for shorter lead times? Please comment.*

**Response: Thank you for the valuable comments. To answer these questions, we present more results to explain how quantile mapping to input variables contributes to improving calibrated ETo forecasts. Specifically, we (1) calculate the correlation coefficients ($r$) between raw/bias-corrected forecasts of the five input variables and AWAP data to further analyze how quantile mapping has improved input variables, in addition to correcting bias (shown in figure 1); (2) investigate the improvements in correlation coefficients between raw ETo forecasts following the bias-correction to input variables and AWAP ETo, to examine how improvements in each variable are translated into the resultant raw ETo forecasts; (3) explain how improvements in raw ETo forecasts through bias-correcting input variables lead to improvements in calibrated ETo forecasts. Please find more details as follows:**

**1, In addition to correcting bias (Figures S2 to S6), quantile mapping also generally improves the temporal patterns of raw forecasts of the input variables. Following figures shows $r$ between raw forecasts of the input variables and their corresponding AWAP data (three columns on the left), and improvements in $r$ by quantile mapping (three columns on the right):**

[Figure]

*Figure S7. Correlation coefficients (r) between raw Tmax forecasts and AWAP data (three panels on the left), and improvements in r (three panels on the right) through quantile mapping*

[Figure]

*Figure S8. Correlation coefficients (r) between raw Tmin forecasts and AWAP data (three panels on the left), and improvements in r (three panels on the right) through quantile mapping*

[Figure]

*Figure S9. Correlation coefficients (r) between raw vapor pressure forecasts and AWAP data (three panels on the left), and improvements in r (three panels on the right) through quantile mapping*

[Figure]

*Figure S10. Correlation coefficients (r) between raw solar radiation forecasts and AWAP data (three panels on the left), and improvements in r (three panels on the right) through quantile mapping*

[Figure]

*Figure S11. Correlation coefficients (r) between raw wind speed forecasts and AWAP data (three panels on the left), and improvements in r (three panels on the right) through quantile mapping*

As shown in the above figures, *r* between raw forecasts of the input variables and AWAP data varies with the input variables. The two temperature variables have higher r values than the other three variables, and wind speed forecasts demonstrate the lowest correlation with AWAP data. For all variables, the *r* values decrease with lead time, indicating higher uncertainties in raw forecasts at longer lead times.

Quantile mapping generally improves the correlation between forecasts and AWAP data. The above figures show that bias-corrected forecasts demonstrate higher *r* for Tmax, solar radiation, and wind speed across most parts of Australia; for Tmin and vapor pressure, changes in *r* are less significant and both improvements and slight decreases in *r* are observed.

We add the above figures to the supplementary. We also add following descriptions to section 3.1:

"Raw forecasts of the input variables generally agree with the AWAP data in temporal patterns during the study period, but the *r* varies with variables (Fig. S7-S11). The two temperature variables (Tmax and Tmin) have higher *r* values (>0.9) than the other three variables, and wind speed forecasts demonstrate the lowest correlations with AWAP data. For all variables, the *r* decreases with lead time, indicating higher uncertainties in raw forecasts at longer lead times."

"In addition, quantile mapping also improves the correlation between forecasts and AWAP data (Fig. S7-S11). The most significant improvements are found in wind speed forecasts, showing increases in *r* by up to 0.2 in central and southern parts of Australia. Forecasts of Tmax and solar radiation also demonstrate higher *r* with the adoption of quantile mapping. Both increases and slight decreases were found for vapor pressure and Tmin, showing that temporal patterns of forecasts of these two variables are not changed much through the bias-correction."

2, With the adoption of quantile mapping to raw forecasts of individual variables, raw ETo forecasts (Calibrations 2 or 4) also show higher *r* with observations, than the raw ETo forecasts constructed with the original raw forecasts of input variables (Calibrations 1 or 3):

[Figure]

**Figure 2: The comparison between the correlation coefficient of AWAP ETo and raw ETo forecasts constructed with the bias-corrected inputs vs. the correlation coefficient of AWAP ETo and raw ETo forecasts constructed with the uncorrected inputs. The boxplot on the right summarizes results for all grid cells.**

As is shown in the above figure, the quantile mapping also improves the temporal patterns of raw ETo forecasts, for all the lead times. More significant improvements are found in northern Australia. However, due to the nonlinearity in the calculation of ETo using the input variables, spatial patterns of improvements in *r* (Figure 2) does not resemble that of any individual input variables. Although both Tmax and wind speed show more significant improvements in northern Australia, where the *r* improvements are greater than other regions (Figure 2), the spatial patterns of *r* improvements in ETo forecasts are different from these two variables in other parts of the country. As a result, we believe that improvements in *r* of raw ETo forecasts are contributed jointly by these input variables and their interactions.

**We add the above figure (Figure 2) to the manuscript and add the following contents to the manuscript:**

**"**The adoption of quantile mapping to improve input variables also improves the temporal patterns of raw ETo forecasts (Figure 2). Compared with the raw ETo forecasts constructed with uncorrected input variables, the raw ETo forecasts based on bias-corrected inputs generally shows higher correlation coefficients with AWAP ETo, particularly in northern Australia. However, due to the nonlinearity in the calculation of ETo using the input variables, spatial patterns of improvements in *r* (Figure 2) does not resemble improvements in any individual input variables (Figures S7 to S11). The improvements in *r* of raw ETo forecasts seem to be contributed jointly by these input variables and their interactions.**"**

**3, We add the following contents to section 3.2 to explain the spatial patterns of changes in *r* and absolute bias:**

"Larger reductions in absolute bias in northern Australia coincide with the improvements in the correlation between raw ETo forecasts and AWAP ETo (Figure 2). However, unlike the improvements in *r* for all lead times in raw ETo forecasts, the improvements in absolute bias are more pronounced for short lead times (Days 1-3) than long lead times (Days 7-9). The uneven improvements may reflect that intrinsic uncertainties at long lead times have hindered the manifestation of improvements to the raw ETo forecasts in calibrated ETo forecasts."

**Based on the above analyses, we can then answer the questions the reviewer raised regarding the figure of absolute bias in this comment.**

**More significant reductions in absolute bias in northern Australia show similar spatial patterns with that of the improvements in correlation coefficient between raw ETo forecasts and AWAP ETo. As we further explained in our response to your next comment (#10), deficiencies in NWP models in simulating weather dynamics in tropical regions have been reported. However, improvements to raw ETo forecasts in r with the application of quantile mapping could not be explained by any individual variable. The nonlinearity in calculating ETo based on the individual variables may have combined improvements in each variable and lead to more significant improvements in northern Australia. Less significant improvements in ETo forecasts at longer lead times may be caused by the more significant intrinsic uncertainties than short lead times. These uncertainties have inhibited the translation of improvements in raw ETo forecasts to calibrated forecasts.**

**Point #10**

*P13 l282-285: Why lowest score of correlation coefficient in northern Territory? Is it linked to the NWP (and if so how?) or is it linked to observations? E.g. differneces in observations compared to rest of country?*

**Response:  Thank you for the comments. We believe the correlation results from the NWP forecasts rather than from observations for several reasons:**

**1, Evaluation of the observations (AWAP data) did not show larger errors in this region, than other areas of Australia (Jones et al., 2009). As a result, we do not have evidence that the quality of observations in this region is lower than in other regions**

**2, Deficiencies of NWP forecasts in tropical regions in Australia have been well documented. Due to its highly dynamic nature, tropical regions often demonstrate larger errors than other climate zones. In the evaluation of NWP forecasts in Australia, tropical zones often show lower skills than other regions (Ebert and Mcbride, 2000; Mcbride and Ebert, 2000; Roux et al., 2010). According to Huang et a. (2018), ACCESS models have been suffering from low skills in simulating the convective processes in tropical zones of Australia.**

**3, Raw ETo constructed with the ACCESS outputs showed higher RMSE in Northern Territory than other regions (Perera et al., 2014), further confirms that lower correlation coefficient is mainly caused by the NWP forecasts.**

**We add the following sentences to the section 3.3:**

"Deficiencies in ACCESS models in simulating dynamics of tropical climate systems may have resulted in low correlation coefficients in Northern Territory."

**Reference:**

Ebert, E. E. and Mcbride, J. L.: Verification of precipitation in weather systems : determination of systematic errors, J. Hydrol., 239, 179–202, 2000.

Huang, J., Rikus, L. J., Qin, Y. and Katzfey, J.: Assessing model performance of daily solar irradiance forecasts over Australia, Sol. Energy, 176(November), 615–626, doi:10.1016/j.solener.2018.10.080, 2018.

Jones, D. A., Wang, W. and Fawcett, R.: High-quality spatial climate data-sets for Australia, Aust. Meteorol. Oceanogr. J., 58, 233–248, 2009.

Mcbride, J. L. and Ebert, E. E.: Verification of quantitative precipitation forecasts from operational numerical weather prediction models over Australia, Weather Forecast., 15(1), 103–121, doi:10.1175/1520-0434(2000)015<0103:VOQPFF>2.0.CO;2, 2000.

Perera, K. C., Western, A. W., Nawarathna, B. and George, B.: Forecasting daily reference evapotranspiration for Australia using numerical weather prediction outputs, Agric. For. Meteorol., 194, 50–63, doi:10.1016/j.agrformet.2014.03.014, 2014.

Roux, B., Seed, A., Pagano, T. and Roux, B.: Improved use of precipitation forecasts in short-term water forecasting – progress report, The Centre for Australian Weather and Climate Research A partnership between CSIRO and the Bureau of Meteorology Improved., 2010.

**Point #11**

*P14 l294-297: The geographical patterns of the correlation performance is very similar to the patterns of the bias performance. Could you please comment why and if the reasons are the same? Are these related to either the NWP or observations?*

**Response: Thank you for the valuable comments. We add the following figure to the manuscript to demonstrate how bias-correction of input variables improves correlations between raw ETo forecasts and AWAP ETo:**

[Figure]

**Figure 2: The comparison between the correlation coefficient of AWAP ETo and raw ETo forecasts constructed with the bias-corrected inputs vs. the correlation coefficient of AWAP ETo and raw ETo forecasts constructed with the uncorrected inputs. The boxplot on the right summarizes results for all grid cells.**

**The above figure shows that when input variables are bias-corrected, the resultant raw ETo forecasts show higher correlation coefficients, than raw ETo forecasts constructed with uncorrected inputs. Spatial patterns of the improvements in *r* in raw forecasts for short lead times are consistent with the improvements in *r* in calibrated forecasts (Figure 6). As a result, we believe this is how the new calibration strategy improves the calibration of ETo forecasts. Less significant improvements in ETo forecasts at longer lead times may be caused by the more significant intrinsic uncertainties in raw forecasts than short lead times. These uncertainties have inhibited the translation of improvements in raw ETo forecasts to calibrated forecasts. We have explained the connections between improvements in raw forecasts and calibrated forecasts in response to your comment #9.**

**As we introduced in the manuscript, when we calibrate the raw ETo forecasts (f(t)), we built a conditional distribution ($\tilde{o}(\mathrm{m}(t))$) for observations ($o(t)$), and 100 values will be drawn from this conditional distribution to generate the calibrated ensemble forecasts:**

$$\tilde{o}(\mathrm{m}(t)) \sim N\left(\mu_o(\mathrm{m}(t)) + r\frac{\sigma_o(\mathrm{m}(t))}{\sigma_f(\mathrm{m}(t))}(f(t) - \mu_f(\mathrm{m}(t))), (1 - r^2)\sigma_o^2\right)$$

**in which where $\mathrm{m}(t)$ returns the month k (k=1 to 12) of daily forecasts or observations of day $t$; $\mu_f(\mathrm{m}(t))$ and $\sigma_f(\mathrm{m}(t))$ refer to the marginal distribution's mean and standard deviation of $f(t)$ in month m(t), respectively; $\mu_o(\mathrm{m}(t))$ and $\sigma_o(m(t))$ are the mean and standard deviation of the**

marginal distribution of $o(t)$ in month $\mathrm{m}(t)$; $r$ is the correlation between $f(t)$ and $o(t)$ in the transformed space.

As a result, when the correlation is improved, it will help improve the estimation of the mean and standard deviation of the above conditional distributions. As a result, bias in calibrated forecasts will be further reduced. That is why improvements in bias demonstrate a similar spatial pattern as those of the correlation coefficient.

To explain improvements in r in calibrated forecasts, we add the following sentence to the section 3.3:

"Spatial patterns of improvements in r of calibrated ETo forecasts (Figure 6) are similar to the improvements in $r$ of raw ETo forecasts (Figure 2), particularly for the short lead times. The improvements in $r$ of calibrated ETo forecasts (Figure 6) may also lead to more reliable conditional distributions for a given raw forecast (equation 4). As a result, regions showing improvements in $r$ in calibrated ETo forecasts (Figure 6) often demonstrate reductions in absolute bias (Figure 3)."

**Point #12**

*P16 l320-328. Please comment on why the accuracy has larger differences in terms of geographical patterns than for the bias and PIT performance which had very strong localised performance.*

Response: Thank you for the comments. We believe there are four reasons for the differences in spatial patterns of CRPS skill score (Figure 8) with changes in bias (Figure 4), correlation coefficient (Figure 6), and alpha index (Figure S13):

1, The metrics measure different features of the quality of forecasts, and may have different sensitivities to changes in calibrated forecasts. As a result, it is not unexpected that their spatial patterns show differences. The CRPS skill score measures the performance of calibrated forecasts relative to the climatology forecast; correlation coefficient shows consistency between observations and forecasts; bias measures average differences; the alpha index is an indicator showing whether the distribution of calibrated forecasts is overconfident or underconfident. As a result, improvements indicated by these metrics do not necessarily show exactly the same spatial patterns.

2, The alpha index is less sensitive to changes in forecasts than other metrics. It is well known that the quality of forecasts often declines with lead time, even for calibrated forecasts. This tendency can be seen from the correlation coefficient (Figure 5) and CRPS skill score (Figure 7). However, the same trend is not shown by the alpha index. As demonstrated by figure 9, the alpha index demonstrates similar magnitudes and spatial patterns among the 9 lead times. As was introduced in equations 13 and 14, PIT value and alpha index are mainly used to measure the consistency between distributions of forecasts and observations. Cmprovements achieved through the adoption of calibration strategy ii (e.g., Calibrations 2

and 4) may not significantly change the statistical distributions of the calibrated forecasts. As a result, differences in alpha index (Figure 13) between Calibrations 2 and 1 do not show spatial patterns resembling absolute bias (Figure 4), correlation coefficient (Figure 6), and CRPS skill score (Figure 8). In addition, the t-test suggested that differences in alpha index between Calibrations 2 and 1 are not statistically significant for most lead times (Table S2).

3, Although Improvements in absolute bias, correlation coefficient, and CRPS skill score measures different features of the improvements (explain in our point 1 of the our response to this current comment), their spatial patterns are generally consistent. We calculate the spatial correlation for changes in CRPS skill score vs. changes in absolute bias (figure 8 vs. figure 4), and the spatial correlation for changes in CRPS skill score vs. changes in correlation coefficients (figure 8 vs. figure 6). As is shown in the following figure, the spatial patterns of CRPS skill score improvements are generally consistent with the reduction in absolute bias (negative r values), and increases in *r* (positive r values).

[Figure]

4, The upper and lower limits used for the maps may have affected our understanding of the spatial patterns of the evaluation metrics. Following comparison shows that when using narrower limits (-3% to 3%, rather than -5% to 5%) for the color bar of the maps showing improvements in correlation coefficients (figure on the right), the spatial pattern is more consistent with the maps showing increases in CRPS skill score (Figure 8).  In the revised manuscript, we use the plot with narrower color bar limits in the revised manuscript.

[Figure]

Differences in correlation coefficient (%)

Difference in correlation coefficient (%)

**To explain spatial patterns of the evaluation metrics, we add a new subsection to the Results section (3.7 Summary of results):**

"Although the selected metrics measure different aspects of forecast quality, they generally agree with each other in demonstrating improvements in calibrated ETo forecasts with the adoption of the Strategy ii. As introduced in the Method section, the CRPS skill score measures the performance of the calibrated forecasts relative to climatology forecast; correlation coefficient shows consistency between observations and forecasts in temporal variability; bias measures average differences; the α-index is an indicator showing whether the distribution of calibrated forecasts is overconfident or underconfident. As a result, these metrics differ from each other when used to measure differences between different calibrations (Figures 4, 6, and 8). However, these three metrics are generally consistent in the spatial patterns of improvements. As demonstrated in Figure 4, the alpha index showed fewer decreases at longer lead times than other metrics, indicating that α-index is less sensitive to changes in the quality of calibrated forecasts. That is why the adoption of calibration Strategy ii did not lead to significant changes in the α-index."

**Point #13**

*P16 l329: Results on calibration 2 and 4: what is the comparison between 2 and 4? Why are these only addressed in the evaluation of forecast accuracy section? Why is there no mention of these for the bias and reliability evaluation? I suggest changing the section order and moving this section first. Then, add a sentence in the bias and reliability section to explicitly communicate what results of experiment 3) and 4) are not presented and why.*

**Response: Thank you for the valuable suggestions. We check the original submission and believe your comments refer to Calibrations 3 and 4 here.**

**As we explain in our response to your comment #5, calibrations 3 and 4 are to further confirm that whether our strategy is suitable for general application. We further explain the reason of by adding the following sentences to clarify why Calibrations 3 and 4 are included in this study in Method:**

"The comparison between Calibrations 1 and 2 is to investigate whether the bias-correction of input variables would further improve ETo forecasts when the calibration is conducted based on ETo anomalies and climatological mean. We also conduct additional calibrations which post-process ETo forecasts directly (Calibrations 3 and 4), to test whether the contribution of improving the input variables to ETo forecast calibration, if there is any, will depend on how ETo forecasts are calibrated (based on anomalies vs. based on original ETo forecasts). Calibrations 3 and 4 will help evaluate the feasibility of strategy ii for the general application in NWP/GCM-based ETo forecasting. Key steps of the four calibrations could be found in the schematic diagram (Figure S1). In the main text, we primarily analyze results from Calibrations 1 and 2. Improvements with the adoption of bias-correction to input variables in Calibrations 3 and 4 are very similar to those of Calibrations 1 and 2 (see the Supplementary Material). To avoid redundancy, we present results from Calibrations 3 and 4 in the Supplementary Material."

**As we introduced in our response to your comment point #5, we add more results (bias, correlation, and alpha-index) from Calibrations 3 and 4 to the Supplementary material and one new subsection (3.6) to briefly introduce these figures (Figures S15-S18).**

**Point #14**

*Discussion:*

*There are little to no direct comparison of results and calibration work presented here to any previous methods or studies (which were mentioned in the introduction). To address a research closure, please put the work presented in this paper in context with other studies applying strategy 1 and strategy 2.*

**Response: We appreciate the reviewer's valuable suggestion. We explain in detail why we do not compare our calibration directly with calibrations using other models in our response to your comment #3. However, we totally agree with the reviewer that it is necessary to compare our results with previous investigations in ETo forecasting to help the audience better understand the performance of our model. Therefore, we add the following contents to the Discussion:**

"This investigation further highlights the importance of statistical calibration in improving the quality of raw ETo forecasts (Medina and Tian, 2020). In the ETo forecasting across 40 sites in Australia, although raw ETo forecasts constructed with NWP outputs reasonably captured the magnitude and variability of ETo, forecast skills better than climatology were only found for the first 6 lead times (Perera et al., 2014). Our investigation suggests that statistical calibration could substantially improve forecast skills and outperform the climatology forecasts for all 9 lead times

across Australia. The findings of this investigation agree well with the site scale short-term ETo forecasting based on GCM outputs (Zhao et al., 2019a) in terms of improvements in forecast skills. Calibrated forecasts from Calibration 2 demonstrate similar skills as those of Zhao et al. (2019a). However, our calibration achieves the improvements using much shorter archived raw forecasts (3-year vs. 23-year) than Zhao et al. (2019a), thanks to the capability of SCC in calibrating short-archived forecasts (Wang et al., 2019). Calibrated forecasts from Calibration 2 also demonstrate comparable biases (0.32-0.95%) with calibrated ETo forecasts (0.49-0.63%) in the U.S. based on the Bayesian model averaging (BMA) model and weather forecasts from three NWP models during 2014-2016 (Medina and Tian, 2020)."

**In addition, we also highlight the importance of testing the proposed calibration strategy (strategy ii) in the future in section 4.2, in the hope that this strategy will be tested b based on other calibration models:**

"Second, further investigations based on other calibration models are needed to validate the conclusions of this investigation. Our analyses based on two different methods (based on ETo anomalies vs. based on original ETo) find similar improvements in calibrated ETo forecasts with the adoption of bias-correction of input variables. Additional evaluations using other calibration models will be needed to ascertain whether the improvements will be achieved when the calibration is conducted with a different model."

**Point #15**

*It is unclear whether authors recommend the use of experiment 2) or 4), when and why. In that sense, I question again the inclusion of these experiments without further elaborating and discussing these results.*

**Response: Thank you for the valuable suggestion. As we explain in our response to your comments #5 and #13, the objective of this study is to evaluate the necessity of correcting the input variables prior to ETo calibration. We also further explain that including Calibrations 3 and 4 was to further evaluate whether the strategy could be generally applied to other calibration models in the revised manuscript. In addition, we add results from Calibrations 3 and 4 and discussed implications from these two calibrations:**

"We also compare the bias, correlation coefficient, CRPS skill score, and reliability of calibrated forecasts from Calibrations 3 and 4, to evaluate whether we can obtain similar improvements through the bias-correction of input variables if we conduct the ETo forecast calibration in a different way (without using climatological mean and anomalies). Results show that the adoption of bias-correction also leads to lower bias, higher correlation coefficient, and higher CRPS skill score in terms of magnitude, spatial patterns, and trend along the lead times, when ETo forecasts are calibrated directly (Figure S15-S17). In addition, the alpha index was only slightly different between Calibrations 3 and 4 (Figure S18). This additional comparison further confirms the general applicability of strategy ii for enhancing NWP-based ETo forecasting."

**Point #16**

Structure:

*The introduction is well structured and appropriately present previous work studies and existing strategies.*

**Response:  We appreciate your constructive comments.**

**Point #17**

*The title is a bit lengthy, authors could consider shortening it.*

**Response: We change the title from:**

**"**Bias-correcting individual inputs prior to combined calibration leads to more skillful forecasts of reference crop evapotranspiration**"**

**to:**

"Bias-correcting individual inputs enhances forecasting of reference crop evapotranspiration."

**Point #18**

*As noted above, I suggest authors consider the order of results presented in the context of results from experiment 3) and 4).*

 **Response: As we explained in our response to your comments #5 and #13, we add a new subsection (3.6) to present results from calibrations 3 and 4 and discuss implications of these two Calibrations.**

**Point #19**

*Minor comments:*

 *P4 l106: I suggest adding a diagram clearly explaining steps and differences of procedure between the calibration experiments.*

**Response: We appreciate the valuable suggestions and create a diagram to show the key steps of the four calibrations**

[Figure]

*Figure S1. Schematic of the four calibrations*

**Point #20**

*P3 l68: '...pressing need to investigate.' Please expand why it is pressing?*

**Response: Thank you for the comments. ETo forecasts have been increasingly used in planning of farming activities (e.g., amount and timing of irrigation) in Australia. We improve this sentence as follows:**

"Since NWP/GCM-based ETo forecasting is increasingly conducted to support water resource management, there is a need to investigate the necessity of correcting raw forecasts of the input variables as part of ETo forecast calibration, to provide high-quality ETo forecasts."

**Point #21**

*P3 l74: Calibrate should be calibrate with small cap letter.*

**Response: Thank you for the careful review. We correct this typo.**

*P3 l80-84: There are many efforts to develop downscaling methods, please comment on what was been done here to downscale ACCESS-G2 to the AWAP grid. Why not scaling AWAP to the match the forecast grid?*

**Response: Thank you for the valuable suggestions. In the revised manuscript, we further introduce that we used bilinear interpolation to remap ACCESS-G2 forecasts. Meanwhile, we agree with the reviewer that sophisticated methods have been developed to downscale coarse resolution forecasts to match observations.**

**In this study, the purpose of the regridding is to connect forecasts with the corresponding observations so we can calibrate the forecasts, rather than trying to reconstruct the spatial patterns of forecasts at a finer scale.**

**We conducted a literature review on the remapping methods used in forecasts post-processing. It is common that raw forecasts and references data have different spatial resolutions. We found that bilinear interpolation of forecasts from a coarser resolution to a finer resolution has been widely used in forecast post-processing and verification. For example, Hamill et al. (2015) used bilinear interpolation to downscale the resolution of Global Ensemble Forecast System (GEFS) forecasts from 1° to 1/8° to match observations before post-processing with an analogy-based model. Yuan et al. (2014) used bilinear interpolation to remap the Global Ensemble Forecast System (GEFS, with resolutions of ~0.469° and ~0.625°) to match the North-American Land Data Assimilation System (NLDAS, with the resolution of 1/8°), before the forecasts were post-processed with a quantile mapping method. Zeng and Yuan (2018) used bilinear interpolation to remap sub-seasonal to seasonal forecasts from ECMWF (0.25°X0.25°to 0.5°X0.5° for different lead times), NCEP (1°X1°), China Meteorological Administration (CMA, 1°X1°), Hydrometeorological Centre of Russia (HMCR, 1.1°X1.4°), and Australian Bureau of Meteorology (BoM, 2°X2°) to a common resolution of 0.7°, in order to match the reanalysis data. James et al. (2017) regridded the wind forecasts with bilinear interpolation from the 3-km High-Resolution Rapid Refresh (HRRR) NWP model to an observation tower in Colorado to evaluate forecast quality. Bowler et al. (2008) interpolated the ECMWF forecasts with a grid spacing of 1.5° bilinearly to the site scale for forecast verification. Yuan and Wood (2012) used bilinear interpolation to match forecasts from the Euro- Mediterranean Centre for Climate Change (CMCC-INGV), the European Centre for Medium-Range Weather Forecasts (ECMWF), the Leibniz Institute of Marine Sciences at Kiel University (IFM-GEOMAR), Météo France, and UK Met Office (UKMO), which have a spatial resolution of 2.5° to match the observation of 1°.**

**As a result, previous investigations suggested that downscaling with a sophisticated method could potentially be useful, but that is not necessarily essential in forecast post-processing, and bilinear interpolation is acceptable.**

**However, we agree with the reviewer that it is necessary To acknowledge this need, we add the following sentence to section 4.2 Implications for forecasting of integrated variables and future work:**

"In the future, more sophisticated remapping method should be adopted to investigate the impacts of grid cell regridding on forecast calibration."

**Reference:**

Bowler, N.E., Arribas, A., Mylne, K.R., Robertson, K.B., Beare, S.E., 2008. The MOGREPS short-range ensemble prediction system. Q. J. R. Meteorol. Soc. 722, 703–722. https://doi.org/10.1002/qj

Hamill, T., Scheuerer, M., Bates, G., 2015. Analog Probabilistic Precipitation Forecasts Using GEFS Reforecasts and Climatology-Calibrated Precipitation Analyses. Mon. Weather Rev. 143, 3300–3309. https://doi.org/10.1175/MWR-D-15-0004.1

James, E.P., Benjamin, S.G., Marquis, M., 2017. A unfied high-resolution wind and solar dataset from a rapidly updating numerical weather prediction model. Renew. Energy 102, 390–405. https://doi.org/10.1016/j.renene.2016.10.059

Monteiro, J.A.F., Strauch, M., Srinivasan, R., Abbaspour, K., Gucker, B., 2016. Accuracy of grid precipitation data for Brazil : application in river discharge modelling of the Tocantins catchment. Hydrol. Process. 30, 1419–1430. https://doi.org/10.1002/hyp.10708

Yuan, X., Wood, E.F., 2012. On the clustering of climate models in ensemble seasonal forecasting. Geophys. Res. Lett. 39, 1–7. https://doi.org/10.1029/2012GL052735

Yuan, X., Wood, E.F., Liang, M., 2014. Integrating weather and climate prediction: Toward seamless hydrologic forecasting. Geophys. Res. Lett. 5891–5896. https://doi.org/10.1002/2014GL061076.Received

Zeng, D., Yuan, X., 2018. Multiscale Land – Atmosphere Coupling and Its Application in Assessing Subseasonal Forecasts over East Asia. J. Hydrometeology 19, 745–760. https://doi.org/10.1175/JHM-D-17-0215.1

**Point #23**

*P4 l100: please add a comment that SCC model will be described in section 2.3.2*

**Response: We added the following sentence to this section:**

"Details of the SCC model are presented in section 2.3.2"

**Point #24**

*P5 l134 climatological means or mean? Please rephrase and clarify this sentence.*

**Response: Thank you, and we change it to 'climatological mean'**

**Point #25**

*P6 l165: Why are only 100 members drawn, is there any difference with a varying number of ensemble members for forecast reliability?*

**Response: Thank you for the comments. We use 100 members because the computation cost is more affordable than using a larger ensemble size.**

**In order to evaluate how different ensemble sizes would affect the reliability and skills of forecasts, we choose 22 sites randomly across Australia and compare the alpha index and CRPS skill score across these sites using 100, 500, and 1000 ensemble sizes.  The following map shows the locations of the 22 sites.**

[Figure]

**The following figure shows the alpha index is almost identical across the selected sizes for the three ensemble sizes:**

[Figure]

**Comparison of CRPS skill score shows that different ensemble sizes have negligible impacts on the score:**

[Figure]

**As a result, we conclude that the ensemble size used in this study is reasonable.**

**Point #26**

*Is there a need or a reason to verify accumulated Eto forecast values across lead times (as is often the case for streamflow forecasting)? Please comment.*

**Response: Thank you for the comments. For short-term weather forecasts, which are issued on a daily basis, users are often interested in the short-lead-time forecasts (e.g., lead times 1 to 3). Accumulated forecasts across all lead times will not provide the information that users are particularly interested.**

**In addition, the evaluation by lead time shows that improvements with the adoption of the new calibration strategy (Calibrations 2 and 4) decrease with lead time, but still show better**

**performance than the calibrations (Calibrations 1 and 3) without correcting input variables, event at lead time 9. As a result, we are confident that evaluation based on accumulated ETo will not change the conclusion of this study.**

**Point #27**

*P8 l225: 'wind speed is higher than 1m/s than the reference in Australia'. Could you please translate that in terms of percentage so that this statement can be more easily compared to other locations.*

**Response: We add more quantitative information in the evaluation of raw forecasts of input variables:**

"The daily minimum temperature (Tmin) is underpredicted by more than 15 °C in western and central parts of Australia by the raw forecasts, but is overpredicted by ca. 1 °C in eastern and southern Australia. Forecasted wind speed is higher than the reference data by more than 1m/s (or by ca. 63%) in most parts of Australia. Similarly, raw solar radiation forecasts are about 5% higher than AWAP data across Australia. Vapor pressure is underpredicted in western and central regions by ca.14%, but is overpredicted by ca. 6% in coastal areas of south-eastern Australia by the raw forecasts."

**Point #28**

*P18 l380' NWP outputs have been increasingly used for ETo forecasting.' For which applications? Please finish the sentence.*

**Response: We modify this sentence as follows:**

" NWP outputs have been increasingly used for ETo forecasting to support water resource management."

**Point #29**

*P18 l385 Addition 'of' in … skill 'of' the calibrated ETo forecasts.*

**Response: We add the missing 'of' to this sentence:**

"With this extra step, the bias, correlation coefficient, and skills of the calibrated ETo forecasts are all improved, particularly for the short-lead-time forecasts."

**Point #30**

*References:*

*Pappenberger, F., M. H. Ramos, H. L. Cloke, F. Wetterhall, L. Alfieri, K. Bogner, A. Mueller and P. Salamon (2015). "How do I know if my forecasts are better? Using benchmarks in hydrological ensemble prediction." Journal of Hydrology 522: 697-713.*

**Response:  We cited this paper in the revised manuscript in introducing the CRPS skill score.**

---

## Author Response (AR1)

**Responses to Reviewer #1**

**Point #1**

*Review to Yang et ., 2021, Bias-correcting individual inputs prior to combined calibration leads to more skillful forecasts of reference crop evapotranspiration. HESSD.*

*In this study, the authors investigated a critical issue in the forecasting of short-term reference crop evapotranspiration (ETo) based on NWP outputs. It is getting popular that weather forecasts from NWP models are used to predict water loss through evapotranspiration. Such information is highly valuable for the effective management of water resources, particularly in arid/semi-arid regions. This investigation develops a new methodology that effectively corrects errors in ETo forecasts, and adds extra skills to statistical calibration. I believe this new post-processing strategy could benefit future NWP-based ETo forecasting. To improve this work, the authors should pay special attention to the following key issues:*

**Response: We appreciate the reviewer's insightful comments. We also believe the findings of this work could contribute to improving future NWP-based ETo forecasting. We address your constructive comments thoroughly and carefully and believe this work has been improved significantly. Please find more details in our point-by-point response.**

**Point #2**

*1, Presentation of the results could be improved. Currently, the authors use maps to show/compare results from different model experiments. These figures could demonstrate the spatial patterns of modeling results. However, it might be more useful if the authors could summarize regional results in a different way, such as using boxplots. I believe that will better show readers the overall statistical information across the whole country than simply plotting the results as maps.*

**Response: Thank you for the valuable suggestions. We create boxplots for all the maps shown in the main text. Since we already have 10 figures in the main text and 18 figures in the supplementary material, we think it is better not to add too many new figures. We combine these new boxplots with maps for Figures 2-6 and 8-9, which have extra zoom for adding new subplots. For Figures 1 and 7, which already include many subplots, we present the corresponding boxplots in the Supplementary Material. Please find the boxplots as follows:**

[Figure]

**Figure 2 Boxplot summarizing improvements in *r* in raw ETo forecasts following bias-correction to input variables**

[Figure]

**Figure 3 Boxplot summarizing bias in calibrated ETo forecasts**

[Figure]

**Figure 4 Boxplot summarizing differences in absolute bias between calibrated ETo forecasts from Calibration 2 with Calibration 1**

[Figure]

**Figure 5 Boxplot summarizing correlation coefficient between calibrated ETo forecasts from Calibration 2 and AWAP ETo data**

[Figure]

**Figure 6 Boxplot summarizing differences in the correlation coefficient (calibrated forecasts vs. AWAP ETo) between Calibrations 2 and 1**

[Figure]

**Figure 8 Boxplot summarizing differences in CRPS skill scores between the calibrated forecast from Calibration 2 with those from Calibration 1**

[Figure]

**Figure 9 Boxplot summarizing the alpha index in the calibrated ETo forecasts**

[Figure]

*Figure S12. Boxplot of biases in raw ETo forecasts constructed raw (blue) and bias-corrected*
*inputs (pink)*

[Figure]

*Figure S13. Boxplot of CRPS skill score in raw (pink) and calibrated ETo forecast (blue) from Calibration 2*

Point #3

*2, Implications for ETo forecasting at the monthly or seasonal scales should be further discussed. ETo forecasting based on monthly or seasonal climate forecasts from GCMs is also widely performed. This study develops the new strategy for short-term forecasts. The applicability of this method to ETo forecasting based on GCM forecasts should be briefly discussed, to benefit a broader range of readers.*

**Response: We agree with the reviewer that ETo forecasting with longer forecast horizons (e.g., monthly and seasonal) based on GCM forecasts is increasingly performed, and it is necessary to evaluate whether the calibration strategy developed in this investigation is applicable to the GCM-based seasonal ETo forecasting. As we have shown in this manuscript, the reduction of error propagation from the input variables to ETo is the key reason why the new strategy has better performance using raw input variables. We expect this will be the case for GCM-based seasonal forecasting. However, testing this idea will be beyond the scope**

**of this current study. To highlight the necessity of adopting this strategy in seasonal ETo**

**forecasting, we add the following paragraph to section 4.2 (Implications for forecasting of**

**integrated variables and future work):**

"The applicability of the calibration strategy developed in this study to seasonal ETo forecasting should be further investigated. Seasonal ETo forecasting based on GCM climate forecast has been increasingly performed (Tian et al., 2014; Zhao et al., 2019b). In these investigations, raw ETo forecasts were also constructed directly with raw GCM climate forecasts. As a result, it is expected that these investigations have suffered from error propagation from input variables to seasonal ETo forecasts. Whether the calibration strategy (strategy ii) developed in this study will be applicable to seasonal ETo forecasting warrants further investigations."

**92 Point #4**

*Specific comments:*

*Line 20, rewrite this sentence. Not clear*

**Response: we replace the original sentence:**

"This calibration strategy is expected to enhance future NWP-based ETo forecasting."

**with**

" We anticipate that future NWP-based ETo forecasting will benefit from adopting the calibration strategy developed in this study to produce more skillful ETo forecasts."

**101 Point #5**

*Line 74 Calibrate->calibrate*

**Response: We correct the typo accordingly.**

**105 Point #6**

*Line 80 compiled as the inputs…..*

**Response: We improve the sentence of:**

"Weather forecasts from the ACCESS-G2 model are compiled to generate ETo forecasts."

**with:**

"Weather forecasts from the Australian Community Climate and Earth System Simulator G2 version (ACCESS-G2) model are extracted as inputs for the calculation of raw ETo forecasts."

**Point #7**

*Line 95 10m -> 10 m.*

**Response: We add a space between the number and the unit. We also check the entire manuscript to correct the format of units.**

**Point #8**

*Line 107-108, need to clarify what the anomaly and climatological mean are referring to*

**Response: To clarify how the anomaly and climatological mean are derived, we replace the sentence:**

"Our recent investigation suggests that ETo forecast calibration based on anomaly and climatological mean produces more skillful calibrated forecasts than calibrating ETo forecasts directly."

**with:**

" Our recent investigation suggests calibrating ETo anomalies, which are calculated as departures from the climatological mean,  could produce more skillful calibrated forecasts than calibrating ETo forecasts directly."

**Point #9**

*Line 165 consider rewriting this sentence. Does not read well.*

**Response: We replace the original sentence of**

"Once we obtain all the parameters for the BN distribution (equation 4), a conditional distribution is established for $o(t)$ when a raw forecast ($f(t)$) is provided."

**with:**

" With the optimized parameters (means, standard deviations, and correlations) for the BN distribution (equation 4), a conditional distribution for o(t) for a given raw forecast (f(t)) is derived."

**Point #10**

*Line 172, what is specific month*

**Response: we replace "specific" with "unselected" to make the wording more specific.**

**Point #11**

Figures in Results: shouldn't the figures be centralized?

**Response: The original format following a template from HESS. After we add boxplots to these maps, the empty space for each figure is significantly reduced. We keep them aligned to the left to be consistent with the provided template.**

**Point #12**

*Line 360, not calibrate directly, should be without correcting forecasts of the inputs*

**Response: Thank you for the suggestion. The key message we want to present here is that statistical models may not be able to correct all errors in integrated variables (such as ETo). However, when the input variables are corrected first, error propagation from inputs to integrated variables, particularly for the errors which could not be corrected by calibration models, will be reduced. To make it clear, we improved the original sentence of:**

"Our investigation suggests that improving the input variables may help correct errors that could not be fixed when calibrating the integrated variables directly."

**with:**

"Our investigation suggests that improving the input variables could effectively reduce error propagation from inputs to integrated variables. This extra step is proven to be particularly useful in reducing errors in the integrated variables that could not be corrected through calibration."

**Point #13**

*Line 365, consider rewriting this sentence*

**Response: Thank you for the suggestion. We replace the original sentence:**

"As a result, using a more sophisticated calibration method to correct errors in input variables, is expected to further improve forecasts of these input variables, resulting in more significant improvements in the final calibrated ETo forecasts."

**with:**

" If a more sophisticated calibration method is employed to the input variables, error propagation from input variables to ETo forecasts will likely be further reduced. As a result, we anticipate that the calibrated ETo forecast will gain further improvements in forecast skills."

**Point #14**

*Line 377-378, two' calibration models' consider to rewrite*

**Response: We improve the original sentence:**

"Additional investigations using other calibration models will help clarify whether the improvements will hold for other calibration models."

**With**

" Additional evaluations will be needed to verify whether forecast skills will be improved using strategy ii but based on a different calibration model. "

**Point #15**
*Line 385, in the calibrated forecasts*

**Response: We add the missing 'in' to this sentence.**

**Point #16**
*Line 386, consider making it shorter and clearer*

**Response: We improve the following sentence:**

"Further investigation indicates that the contribution of improving input variables to the ETo forecasting tends to be independent of the calibration method applied to raw ETo forecasts."

**With**

" Further investigation indicates that the improvements tend to be independent of the calibration method applied to ETo forecasts."

**Responses to Reviewer #2**

**Point #1**

*Comments on âBias-correcting individual inputs prior to combined calibration leads to more skillful forecasts of reference crop evapotranspirationâ by Yang et al. This study evaluated two calibration strategies for simulating reference crop evapotranspiration. The two strategies are (1) calibration directly applied to raw ETo forecast constructed with raw forecast of input variables; (2) bias-correcting input variables. The bias-correcting algorithm has been proved to be more feasible. Although this study is of significance, improvements and revision can make the study stronger and more compelling.*

**Response: We appreciate the reviewer's insightful suggestions and comments on the manuscript. We address comments from the reviewer carefully and improve the manuscript accordingly. Please see details in our point-by-point response.**

**Point #2**

*Core of my concerns is the results presentation and discussion, many sections are superficial; the results are simply described, more insightful explanation and discussion are needed. See below for my suggestion. A moderate revision can easily address these comments. So I suggest a moderate revision.*

**Response: We appreciate the reviewer's constructive comments. We improve the analysis and presentations by (1) creating boxplots to summarize results plotted as maps to better demonstrate results quantitatively, (2) performing statistical analyses  (t-test) when comparing results from different Calibrations, (3) providing more statistical information in the Results section, and (4) Comparing findings of this work with published investigations. We further explain these improvements in detail as follows:**

**(1)  Adding boxplots to Results**

**We create boxplots for results shown as maps (Figures 1 to 9 in the main text). We combine these boxplots with maps for Figures 2-6, 8-9, which have extra zoom for adding new subplots. For Figures 1 and 7, which already include many subplots, we present the corresponding boxplots in the Supplementary Material. We also update the main text accordingly. Please find the boxplots as follows:**

[Figure]

**Figure 2 Boxplot summarizing improvements in *r* in raw ETo forecasts following bias-correction to
input variables**

[Figure]

**Figure 3 Boxplot summarizing bias in calibrated ETo forecasts**

[Figure]

**Figure 4 Boxplot summarizing differences in absolute bias between calibrated ETo forecasts from Calibration 2 with Calibration 1**

[Figure]

**Figure 5 Boxplot summarizing correlation coefficient between calibrated ETo forecasts from Calibration 2 and AWAP ETo data**

[Figure]

**Figure 6 Boxplot summarizing differences in the correlation coefficient (calibrated forecasts vs.**
**AWAP ETo) between Calibrations 2 and 1**

[Figure]

**Figure 8 Boxplot summarizing differences in CRPS skill scores between the calibrated forecast**
**from Calibration 2 with those from Calibration 1**

[Figure]

**Figure 9 Boxplot summarizing the alpha index in the calibrated ETo forecasts**

[Figure]

*Figure S12. Boxplot of biases in raw ETo forecasts constructed raw (blue) and bias-corrected*
*inputs (pink)*

[Figure]

*Figure S13. Boxplot of CRPS skill score in raw (pink) and calibrated ETo forecast (blue) from Calibration 2*

**(2) Conducting t-tests to compare results from different Calibrations.**

**We conduct t-tests (Table S1) to evaluate raw forecasts of the five input variables (Figures S2 to S6). T-tests were also conducted in the evaluation of bias, correlation coefficient, and CRPS skill score (Figures 1-3, 6-9) of forecasts produced in Calibrations 1 and 2(Table S2).**

**In the calculation of *t* statistics, we use the Spatial Degrees of Freedom (SDOF), rather than using the total grid cells in the study area, to account for the spatial correlation in the t-test. The SDOF is substantially smaller than total grid cells** (Toth, 1995)**. Wang and Shen (1999) investigated SDOF of GCM outputs and reported a range of 90-120, out of 738 grid cells for the southern hemisphere. In this study, we use 50 as the SDOF for our t-tests. Considering the large amount of total grid cells (281,622) in this study, we believe that 50 is a conservative estimate of SDOF for this investigation. We calculated the *t-statistics* and evaluate whether**

**they are statistically significant using the SDOF of 50. Results of the t-tests (Tables S1 and S2)**
**are added to the supplementary material.**

**Reference:**

Toth, Z.: Degrees of freedom in Northern Hemisphere circulation data, Tellus, Ser. A, 47 A(4),
457–472, doi:10.3402/tellusa.v47i4.11531, 1995.

Wang, X. and Shen, S. S.: Estimation of spatial degrees of freedom of a climate field, J. Clim.,
12(5 I), 1280–1291, doi:10.1175/1520-0442(1999)012<1280:EOSDOF>2.0.CO;2, 1999.

**Table S1 Results of t-tests** (*t-statistic*) **for raw forecasts of input variables**

| Tests

Lead
times | Test if bias in
raw Tmax
forecasts is
different from
zero (Figure
S2) | Test if bias in
raw Tmin
forecasts is
different from
zero (Figure
S3) | Test if bias in
raw vapor
pressure
forecasts is
different from
zero (Figure S4) | Test if bias in
raw solar
radiation
forecasts is
different from
zero (Figure S5) | Test if bias in
raw wind
speed forecasts
is different
from zero
(Figure S6) |
|---|---|---|---|---|---|
| Day 1 | -8.96** | 1.66 | -3.18** | 11.83** | 16.04** |
| Day 2 | -8.16** | 2.65** | -3.43** | 11.39** | 16.50** |
| Day 3 | -8.19** | 2.68** | -3.77** | 11.81** | 16.57** |
| Day 4 | -8.12** | 2.56** | -4.05** | 12.17** | 16.56** |
| Day 5 | -7.87** | 2.41** | -4.09** | 12.45** | 16.45** |
| Day 6 | -7.70** | 2.27** | -4.21** | 11.88** | 16.45** |
| Day 7 | -7.73** | 2.22** | -4.33** | 10.81** | 16.29** |
| Day 8 | -7.70** | 2.17** | -4.30** | 11.41** | 16.56** |
| Day 9 | -7.44** | 2.20** | -4.18** | 11.95** | 16.82** |

**The Spatial Degrees of Freedom (SDOF) is 50 in the tests; ** indicates statistically significant differences at the 95% confidence interval.**

**Table S2 Results of t-tests (*t-statistic*) for performance evaluation**

| Tests

Lead times | Comparison of bias in raw ETo forecasts constructed with vs. without bias correction (Figure 1) | Test if r in raw ETo forecasts constructed with raw and bias-corrected input variables are different (Figure 2) | Test if bias in calibrated ETo forecasts from Calibration 2 (Figure 3) is different from zero | Test differences in absolute bias between calibrated ETo forecasts from Calibrations 2 and 1 (Figure 4) | Test difference in r between observations and calibrated ETo forecasts from Calibrations 2 and 1 (Figure 6) | Comparison of CRPS skill score between raw and calibrated ETo forecasts (Figure 7) | Test difference in CRPS skill score of calibrated ETo forecasts from Calibrations 2 and 1 (Figure 8) | Test difference in α-index between Calibrations 2 and 1 (Figure S14) | Test if difference in CRPS skill scores between Calibrations 3 and 4 (Figure S17) |
|---|---|---|---|---|---|---|---|---|---|
| Day 1 | -9.76** | 7.26** | 1.80 | -4.08** | 5.73** | 27.59** | 11.53** | -0.54 | 11.81** |
| Day 2 | -9.86** | 7.13** | 1.91 | -3.93** | 4.93** | 29.03** | 10.86** | -1.47 | 10.26** |
| Day 3 | -9.86** | 7.01** | 2.07** | -3.68** | 4.43** | 31.14** | 9.77** | -1.81 | 9.16** |
| Day 4 | -9.81** | 7.04** | 2.27** | -3.54** | 4.01** | 33.77** | 8.58** | -1.17 | 8.33** |
| Day 5 | -9.71** | 7.09** | 2.40** | -3.36** | 3.75** | 38.11** | 7.16** | -2.09** | 7.25** |
| Day 6 | -9.54** | 7.33** | 2.60** | -3.37** | 3.17** | 42.59** | 6.44** | -1.28 | 6.66** |
| Day 7 | -9.34** | 7.40** | 2.76** | -3.26** | 2.69** | 44.38** | 6.15** | -1.99 | 6.25** |
| Day 8 | -9.04** | 7.54** | 2.98** | -3.13** | 2.32** | 45.57** | 5.85** | -1.57 | 5.67** |
| Day 9 | -9.21** | 7.50** | 3.13** | -2.91** | 1.85 | 51.91** | 5.05** | -1.70 | 4.95** |

**The Spatial Degrees of Freedom (SDOF) is 50 in the tests; \*\* indicates statistically significant differences at the 95%**
**confidence interval.**

**(3) Improving the Results section**

**We add more specific information in describing the key findings of this study and introduce the results of the statistical analyses (Tables S1 and S2). Since we modified many sentences, we decide not to list them here. Please see details in the revised manuscript.**

**(4) Improving the Discussion section**

**We further compare the findings of this investigation with existing studies in discussion:**

"This investigation further highlights the importance of statistical calibration in NWP-based ETo forecasting (Medina and Tian, 2020). According to an investigation across 40 sites in Australia, raw ETo forecasts constructed with NWP outputs reasonably captured the magnitude and variability of ETo, but forecast skills better than climatology were only limited to the first 6 lead times (Perera et al., 2014). Our investigation suggests that statistical calibration could substantially improve forecast skills and successfully extend the skillful forecasts to lead time 9 across Australia. Findings of this investigation agree well with the site-scale short-term ETo forecasting based on GCM outputs (Zhao et al., 2019a) in the improvements of forecast skills through statistical calibration. Calibrated forecasts from Calibration 2 demonstrate similar skills as Zhao et al. (2019a) across three Australian sites. Thanks to the capability of SCC in calibrating short-archived forecasts (Wang et al., 2019), we achieve the improvements based on much shorter archived raw forecasts (3-year vs. 23-year) than Zhao et al. (2019a). Calibrated forecasts from Calibration 2 also demonstrate low biases (0.32-0.95%) comparable with calibrated ETo forecasts (0.49-0.63%) based on the Bayesian Model Averaging (BMA) model and weather forecasts from three NWP models in the U.S. during 2014-2016 (Medina and Tian, 2020)."

**Point #3**

Lines 11, fully implemented.

**Response: we change it to 'fully implemented '.**

**Point #4**

*Line 27, âdivergentâ emphasizes completely different assumption, you can just use replace it different to ensure a general term.*

**Response: We replace the word 'divergent' with 'different'.**

**Point #5**

*Line 38, physical processes of the atmosphere, it is unclear, atmospheric circulation or atmospheric wind formation, or physical processes in the atmosphere*

**Response: Thank you for the suggestion. We change the sentence as follows:**

" ETo is affected jointly by temperature, vapor pressure, solar radiation, and wind speed (Bachour et al.,
2016; Luo et al., 2014). Prediction models using these weather variables as inputs allow for
representations of atmospheric dynamics and often produce reasonable ETo forecasts (Torres et al.,
2011)."

**Point #6**

*Section 3.1, 3.2, the authors described the results in the figures. However, most of those text are vague,*
*please provide more specific (quantitative) information to support your statement. When you compare*
*different results or method, it is better to report some statistic results (p value, r2, etc).*

**Response: We appreciate the constructive comments. We conduct statistical analysis to**
**quantify the difference between different model runs, and update the Results section**
**accordingly. Details of the t-tests could be found in our response to your comments point #2.**

**Point #7**

*for example, line line 223-225, you report the overprediction in Tmax, and underpredict in Tmin in*
*different regions. If it is underprediction, what is the range of that underprediction, same for*
*overprediction, are these different statistically significant? There are many similar issues in other*
*sections.*

**Response: We appreciate the reviewer's valuable suggestions. We agree with the reviewer**
**that more statistical information is needed. We conduct statistical analysis to quantify errors**
**in raw forecasts (Table S1), and update contents in Results accordingly. Statistical analyses**
**could be found in our response to your comment #2. Here is the updated description of errors**
**in raw forecasts of input variables:**

**"**Raw forecasts of the five input variables demonstrate significant inconsistencies with the
corresponding AWAP data (Figures S2-S6). In most parts of Australia, raw daily maximum
temperature (Tmax) forecasts are lower than AWAP data by 1-2 °C. Overpredictions in Tmax
are only found in coastal areas of northwestern Australia. The daily minimum temperature
(Tmin) is underpredicted by more than 1.5 °C in western and central parts of Australia by the
raw forecasts, but is overpredicted by ca. 1 °C in eastern and southern Australia. Vapor pressure
is underpredicted in western and central regions by ca.14%, but is overpredicted by ca. 6% in
coastal areas of southeastern Australia by the raw forecasts. Raw solar radiation forecasts are
about 5% higher than AWAP data across Australia. Forecasted wind speed is higher than the
reference data by more than 1 m s-1 (or by ca. 63%) in most parts of Australia. For each input
variable, spatial patterns of biases in raw forecasts are consistent across the 9 lead times,
demonstrating systematic errors in the raw NWP forecasts. According to our statistical test,
overpredictions or underpredictions in raw forecasts of the input variables are statistically
significant ($P<0.05$) for most lead times (Table S1).**"**

**Point #8**

*In the discussion section, I would be willing to see a comparison with other studies with different algorithms for the ETo simulation. Some quantitative comparison to elucidate the better performance of the new bias-correction algorithm needs to be done. I believe it will prove the reliability of the new algorithm.*

**Response: We appreciate the constructive comments. This is the first continental-scale ETo forecasting in Australia. Previous NWP/GCM-based ETo forecasting in Australia is conducted at the site scale. As a result, in the original manuscript, our evaluation was primarily focused on the comparison against observations. In this area of weather/climate forecasting, different calibration models, based on different statistical theories have been developed and implemented. Previous comparisons suggest that the performance of these models varied with study areas, NWP models, and choice of evaluation metrics (Wilks, 2018), and there is no conclusion regarding which group of post-processing models has the best performance.**

**More importantly, rather than developing a new calibration model, this investigation is to evaluate the necessity of including an extra step before ETo forecasts are calibrated. As we introduced in the main text, the objective of our investigations is to address a challenge commonly faced by NWP-based ETo forecasting. We expect the calibration strategy developed in this study will benefit ETo forecast calibrations broadly, no matter which statistical model is employed in ETo forecast calibration.**

**However, we agree with the reviewer that comparison of model performance with other models will help readers better understand the robustness of our calibration. We review previous studies and add the following content to the Discussion section (4.1):**

"According to an investigation across 40 sites in Australia, raw ETo forecasts constructed with NWP outputs reasonably captured the magnitude and variability of ETo, but forecast skills better than climatology were only limited to the first 6 lead times (Perera et al., 2014). Our investigation suggests that statistical calibration could substantially improve forecast skills and successfully extend the skillful forecasts to lead time 9 across Australia. Findings of this investigation agree well with the site-scale short-term ETo forecasting based on GCM outputs (Zhao et al., 2019a) in the improvements of forecast skills through statistical calibration. Calibrated forecasts from Calibration 2 demonstrate similar skills as Zhao et al. (2019a) across three Australian sites. Thanks to the capability of SCC in calibrating short-archived forecasts (Wang et al., 2019), we achieve the improvements based on much shorter archived raw forecasts (3-year vs. 23-year) than Zhao et al. (2019a). Calibrated forecasts from Calibration 2 also demonstrate low biases (0.32-0.95%) comparable with calibrated ETo forecasts (0.49-0.63%) based on the Bayesian Model Averaging (BMA) model and weather forecasts from three NWP models in the U.S. during 2014-2016 (Medina and Tian, 2020)."

**In addition, we also highlight the importance of testing the proposed calibration strategy (strategy ii) based on other calibration models in the future in section 4.2:**

"Third, further investigations based on other calibration models are needed to validate findings of this investigation. Our analyses based on two different methods (based on ETo anomalies vs. based on original ETo) demonstrate similar improvements in calibrated ETo forecasts with the adoption of bias- correction to input variables. Additional evaluations will be needed to verify whether forecast skills will
be improved using strategy ii but based on a different calibration model."

**Reference:**

Wilks, D.S., 2018. Chapter 3. Univariate Ensemble Forecasting, in: Vannitsem, S., Wilks, D.S., Messner,
J.W. (Eds.), Statistical Postprocessing of Ensemble Forecasts. pp. 49–89.
https://doi.org/https://doi.org/10.1016/C2016-0-03244-8

## Point #9

*Line 388, feasible or reliable ETo forecasting.*

**Response: This paragraph has been rewritten. Please see the revised contents in our response**
**to your comment #10.**

## Point #10

*Line 390, short-term ETo forecasting provides highly valuable information for real-time decision making*
*on water resource management and planning farming practices. This study proved the bias-correction*
*approach is a feasible method for a more robust calibration of the NWP-based ETo forecasting.*

**Response: We appreciate the reviewer's valuable suggestions. We remove redundant**
**sentences and combine the last two paragraphs in the Conclusion section:**

" This investigation clearly suggests the necessity of improving input variables as part of ETo forecast
calibration. With this extra step, the bias, correlation coefficient, and skills of the calibrated ETo forecasts
are all improved. Further investigation indicates that the improvements tend to be independent of the
calibration method applied to ETo forecasts. Forecasting the highly variable ETo is often challenging.
This investigation addresses a common challenge in NWP-based ETo forecasting and develops an
effective calibration strategy for adding extra skills to ETo forecasts. We anticipate that future NWP-
based ETo forecasting could benefit from adopting this strategy to produce more skillful calibrated ETo
forecasts. This strategy is also expected to be applicable to enhancing the forecasting of other integrated
variables that are calculated using multiple NWP/GCM variables as inputs."

**Responses to Reviewer #3**

**Point #1**

*Author(s): Qichun Yang et al.*

*MS No.: hess-2021-69*

*This paper focuses on the comparison of two calibration strategies to provide short-term reference crop evapotranspiration (ETo). ETo forecasting is still a relatively new area of research, in Australia and elsewhere, and has received more attention in the past few years. Skilful ETo forecasts in Australia would help support efficient water use and water management. Two strategies to calibrate ETo forecasts have emerged: i) the calibration of raw ETo forecasts and ii) bias-correcting input variables first before calibrating ETo forecasts. Little work to date compares the two approaches, it is unclear which method might be more advantageous or skilful. This paper therefore addresses a topical subject with a large audience interest.*

*I have some reservations regarding some methodological choices and justifications (purpose and inclusion of experiment 3 and 4), as well as a lack of interpretations of the results overall. I recommend revision to strengthen this paper.*

**Response: Thank you for the valuable suggestions and careful review. We revise this work carefully based on your constructive suggestions.**

**Point #2**

*The authors re-grid the weather forecast variables of ACCESS-G2 to match the timeframe and resolution of the gridded data AWAP. They perform four experiments: experiments 1) and 2) are based on the ETo anomaly and climatological mean, whereas experiment 3 and 4) use the ETo values directly. Furthermore, experiment 1) and 3) use raw inputs to calculate and calibrate ETo forecasts whereas experiments 2) and 4) first bias-correct inputs before ETo calibration. The SCC calibration method is used for ETo forecast while a quantile mapping method is used to bias-correct input forecasts. The authors evaluate the forecasts using three metrics for the theoretical assessment of bias, reliability and accuracy. Overall results suggest that the second strategy (bias-correction of inputs before ETo calibration) provides more skilful forecasts.*

**Response: We appreciate the reviewer's thorough review. The work has been substantially improved through addressing the valuable comments.**

 Point #3

 *Major comments:*

*Methodology:*

*P4 section 2.3: Why not compare the calibration method used SCC to other methods tested in the*
*literature which would enable to place this work in context to other studies on ETo forecasting?*

**Response: We appreciate the constructive comments. We understand that comparing the**
**performance of SCC with existing methods will help readers better understand the strengths**
**of our methodology in ETo forecasting. We did not compare the calibration based on SCC**
**model directly with other models in the original submission for a couple of reasons:**

**First, the primary objective of this investigation is to address a common challenge faced by**
**NWP-based ETo forecasting, rather than to develop a new calibration model. As a result, we**
**primarily focus on evaluating the necessity of correcting forecasts of input variables prior to**
**calibrating ETo forecasts. As we introduced in the main text, the developed calibration**
**strategy is expected to benefit ETo forecast calibrations broadly, rather than improving an**
**individual calibration model. As suggested by the model experiments (Calibrations 1-4), the**
**developed strategy could be applicable to other calibration models.**

**Second, we feel it is not necessary to compare the performance of SCC against calibration**
**models, which are widely used but less sophisticated models. Simple calibration models, such**
**as quantile mapping (QM), have been widely used in calibrating hydroclimate forecasts.**
**These models are often readily available, or could be easily coded and implemented.**
**However, the limitations of these models in forecast calibration have been reported (Zhao et**
**al., 2017). When we started this investigation, we used quantile mapping to calibrate ETo**
**forecasts (raw ETo forecasts constructed with raw forecasts of input variables). As**
**demonstrated in the following figure, the CRPS skill score of quantile mapped ETo forecasts is**
**not only lower than the SCC-calibrated forecasts for each corresponding lead time (Figure 7),**
**but also becomes negative (worse than climatological forecasts) in parts of Australia starting**
**from lead time 4. As a result, calibration of ETo forecasts with quantile mapping further**
**confirms the limitations of this model. Therefore, using such models as a reference to**
**evaluate the performance of SCC is not necessary since their limitations have been reported.**
**As a result, we decide not to include a comparison with quantile mapping in this manuscript.**

[Figure]

*CRPS skill score of calibrated ETo forecasts using Quantile Mapping*

Third, we have limited access to sophisticated calibration models. There is no global post-processing software library archiving these models. We found it was hard to access the source code of these models and to directly compare SCC with them. In addition, previous comparisons suggest that the performance of these models varied with study areas, NWP models, and choice of evaluation metrics (Wilks, 2018), and there is no conclusion regarding which group of post-processing models has the best performance. Our indirect comparison with other models confirms this conclusion. Details will be presented in the following paragraphs.

**Fourth, the short-achieved NWP forecasts (3-year) used in this study represent a challenge for conducting the calibration using other models. Many calibration models, particularly those based on models of the joint probability of forecasts and observations** (Krzysztofowicz and Herr, 2001; Wang and Robertson, 2011)**, require long hindcasts (20-30 years) to establish a joint distribution to link observations and forecasts. Applying such models to short-archived forecasts such as those used in this study will substantially undermine the statistical assumption of these models. In contrast, the SCC model has been developed specifically to address the challenge associated with short-archived forecasts. The advantages of SCC in calibrating short-archived forecasts have been explained in our recent publications** (Wang et al., 2019; Yang et al., 2021)**.**

**As a result, we decide not to compare SCC directly with other models. However, we totally agree with the reviewer that comparison of model performance with other models will help readers better understand the performance of our calibration. As a result, we extract our results at three Australia sites where ETo forecasts were also calibrated based on the Bayesian Joint Probability (BJP) model** (Zhao et al., 2019)**, and compare the results of the two investigations. In addition, we also compare our results with site-scale investigations in other regions of Australia. We also compare results of this study with investigations in the U.S. We add the following paragraph to discuss findings of our work relative to existing investigations to the Discussion section (4.1):**

"This investigation further highlights the importance of statistical calibration in NWP-based ETo forecasting (Medina and Tian, 2020). According to an investigation across 40 sites in Australia, raw ETo forecasts constructed with NWP outputs reasonably captured the magnitude and variability of ETo, but forecast skills better than climatology were only limited to the first 6 lead times (Perera et al., 2014). Our investigation suggests that statistical calibration could substantially improve forecast skills and successfully extend the skillful forecasts to lead time 9 across Australia. Findings of this investigation agree well with the site-scale short-term ETo forecasting based on GCM outputs (Zhao et al., 2019a) in the improvements of forecast skills through statistical calibration. Calibrated forecasts from Calibration 2 demonstrate similar skills as Zhao et al. (2019a) across three Australian sites. Thanks to the capability of SCC in calibrating short-archived forecasts (Wang et al., 2019), we achieve the improvements based on much shorter archived raw forecasts (3-year vs. 23-year) than Zhao et al. (2019a). Calibrated forecasts from Calibration 2 also demonstrate low biases (0.32-0.95%) comparable with calibrated ETo forecasts (0.49-0.63%) based on the Bayesian Model Averaging (BMA) model and weather forecasts from three NWP models in the U.S. during 2014-2016 (Medina and Tian, 2020)."

**In addition, we also highlight the importance of further testing the proposed calibration strategy (strategy ii) based on other calibration models. We add the following contents to section 4.2:**

**"Third, further investigations based on other calibration models are needed to validate findings
of this investigation. Our analyses based on two different methods (based on ETo anomalies vs.
based on original ETo) demonstrate similar improvements in calibrated ETo forecasts with the
adoption of bias-correction to input variables. Additional evaluations will be needed to verify
whether forecast skills will be improved using strategy ii but based on a different calibration
model."**

**Reference:**

Medina, H. and Tian, D.: Comparison of probabilistic post-processing approaches for improving
numerical weather prediction-based daily and weekly reference evapotranspiration forecasts,
Hydrol. Earth Syst. Sci., 24, 1011–1030, 2020.

Perera, K. C., Western, A. W., Nawarathna, B. and George, B.: Forecasting daily reference
evapotranspiration for Australia using numerical weather prediction outputs, Agric. For. Meteorol.,
194, 50–63, doi:10.1016/j.agrformet.2014.03.014, 2014.

Wilks, D.S., 2018. Chapter 3. Univariate Ensemble Forecasting, in: Vannitsem, S., Wilks, D.S., Messner,
J.W. (Eds.), Statistical Postprocessing of Ensemble Forecasts. pp. 49–89.
https://doi.org/https://doi.org/10.1016/C2016-0-03244-8

Krzysztofowicz, R., Herr, H.D., 2001. Hydrologic uncertainty processor for probabilistic river stage
forecasting: precipitation-dependent model. J. Hydrol. 249, 46–68.

Wang, Q.J., Robertson, D.E., 2011. Multisite probabilistic forecasting of seasonal flows for streams with
zero value occurrences. Water Resour. Res. 47, 1–19. https://doi.org/10.1029/2010WR009333

Wang, Q.J., Zhao, T., Yang, Q., Robertson, D., 2019. A Seasonally Coherent Calibration ( SCC ) Model for
Postprocessing Numerical Weather Predictions. Mon. Weather Rev. 147, 3633–3647.
https://doi.org/10.1175/MWR-D-19-0108.1

Yang, Q., Wang, Q.J., Hakala, K., 2021. Achieving effective calibration of precipitatioAn forecasts over a
continental scale. J. Hydrol. Reg. Stud. 35, 100818. https://doi.org/10.1016/j.ejrh.2021.100818

Zhao, T., Wang, Q.J., Schepen, A., 2019. A Bayesian modelling approach to forecasting short-term
reference crop evapotranspiration from GCM outputs. Agric. For. Meteorol. 269–270, 88–101.
https://doi.org/10.1016/j.agrformet.2019.02.003

Point #4
*Presentation of summary statistics. Why not use boxplots to present overall statistics and across lead*
*times (for example next to figure 4 and so on)? Reliability diagrams for particular ETo thresholds would*
*be helpful to communicate when the forecasts are reliable.*

**Response: Thank you for the constructive suggestions. We created boxplots for results
shown as maps (Figures 1 to 9 in the main text). For Figures 1 and 7, which already include
many subplots, we present the corresponding boxplots in the Supplementary Material. For
other map figures (Figures 2-6, and 8-9), which have extra zoom for adding new subplots, we**

**combine boxplots with the maps. We also update the main text accordingly. Please find the**

**boxplots as follows:**

[Figure]

**Figure 2 Boxplot summarizing improvements in *r* in raw ETo forecasts following bias-correction to**
**input variables**

[Figure]

**Figure 3 Boxplot summarizing bias in calibrated ETo forecasts**

[Figure]

**Figure 4 Boxplot summarizing differences in absolute bias between calibrated ETo forecasts from Calibration 2 with Calibration 1**

[Figure]

**Figure 5 Boxplot summarizing correlation coefficient between calibrated ETo forecasts from Calibration 2 and AWAP ETo data**

[Figure]

**Figure 6 Boxplot summarizing differences in the correlation coefficient (calibrated forecasts vs. AWAP ETo) between Calibrations 2 and 1**

[Figure]

**Figure 8 Boxplot summarizing differences in CRPS skill scores between the calibrated forecast from Calibration 2 with those from Calibration 1**

[Figure]

**Figure 9 Boxplot summarizing the alpha index in the calibrated ETo forecasts**

[Figure]

*Figure S12. Boxplot of biases in raw ETo forecasts constructed raw (blue) and bias-corrected*
*inputs (pink)*

[Figure]

*Figure S13. Boxplot of CRPS skill score in raw (pink) and calibrated ETo forecast (blue) from Calibration 2*

We also create reliability diagrams to summarize to evaluate the calibrated ensemble
forecasts from Calibration 2. The three thresholds used to generate the reliability diagram are
3 mm/day, 6mm/day, and 9 mm/day:

[Figure]

**Figure 10: Reliability diagrams of calibrated ETo forecasts during 4/2016-3/2019 with thresholds of**
**3, 6, and 9 mm day$^{-1}$.**

**We update the Method section to introduce how the reliability diagram is created and how to understand the diagram:**

*"We further evaluate the reliability of calibrated ETo forecasts from calibration 2 using the reliability diagram (Hartmann et al., 2002), which assesses how well the predicted probabilities of forecasts match observed frequencies. We convert the calibrated ensemble ETo forecasts to forecast probabilities exceeding three thresholds, including 3, 6, and 9 mm day-1. We pool forecasts of different grid cells, days, and lead times together in the calculation of forecast probability. In the reliability diagram, perfectly reliable forecasts would demonstrate a curve along the diagonal. A plotted curve above the diagonal indicates underestimations and vice versa."*

**We add the following sentence to section 3.5 (Reliability of calibrated ETo forecasts) to introduce the reliability diagram.**

"The reliability diagram further confirms the consistency between forecast probabilities and observed frequencies (Figure 10). The plotted curves based on three thresholds (3, 6, and 9 mm day-1) are mainly distributed along the 1:1 line, further indicating the high reliability of calibrated ETo forecasts."

**Point #5**

*Authors present experiments 1-4 in the method but then only present some results one experiment 3) and 4) in the last section of results (CRPSS in 3.5). No explanation are provided of why calibration 3) and 4) are only briefly introduced. Why is there a big gap with no results on calibration 3) and 4) on the bias and reliability results? Could the authors please expand on the purpose of including these at all in? At p17 l350-354, 'a further evaluation based on a different way of implementing the calibration demonstrate similar improvements in calibrated ETo forecasts with the adoption of bias-correction to input variables'. Is the purpose of including experiment 3) and 4) to test the generalisation of the method? If so, it needs to be clearly stated and justified earlier.*

**Response: Thank you for the valuable comments. The reviewer is correct that adding calibrations 3 and 4 is to further evaluate whether the developed calibration strategy could be generally applied to future NWP-based ETo forecasting, and will the strategy be independent of calibration models. We further clarify why we include Calibrations 3 and 4 in Method (section 2.3):**

"The comparison between Calibrations 1 and 2 is to investigate whether the bias-correction of input variables would further improve ETo forecasts when the calibration is conducted based on ETo anomalies and climatological mean. We also conduct additional calibrations which post-process ETo forecasts directly (Calibrations 3 and 4), to test whether the contribution of improving input variables to ETo forecast calibration, if there is any, will depend on how ETo forecasts are calibrated (based on anomalies vs. based on ETo). Calibrations 3 and 4 will help evaluate the general applicability of strategy ii to enhance NWP/GCM-based ETo forecasting. Key steps of the four calibrations could be found in the schematic diagram introducing how raw ETo forecasts are constructed and how calibrations are conducted (Figure S1). In the main text, we primarily analyze results from Calibrations 1 and 2.

Improvements with the adoption of bias-correction to input variables in Calibrations 3 and 4 are very similar to Calibrations 1 and 2 (see the Supplementary Material). To avoid redundancy, we mainly present results from Calibrations 3 and 4 in the Supplementary Material.**"**

**In the original submission, we did not present all results from Calibrations 3 and 4 because**

**these two calibrations were complementary for supporting findings from Calibrations 1 and**

**2. In addition, differences in bias, reliability, and correlation coefficient between Calibrations**

**3 and 4  are very similar to those between Calibrations 1 and 2. We thought it might be a bit**

**redundant and may confuse readers if we present all results from Calibrations 3 and 4 in the**

**main text. However, we agree with the reviewer that it is necessary to present results from**

**Calibrations 3 and 4 in case readers are interested in them. In the revised manuscript, we**

**present them in the supplementary material (See the figures below), in order not to distract**

**readers from understanding key objectives (e.g., the necessity of bias-correcting input**

**variables prior to ETo calibration) of this investigation. Specifically, in addition to the figure**

**showing improvements in CRPS skill score, we also add figures demonstrating differences in**

**absolute bias (Figure S15), correlation coefficients (Figure S16), and alpha index (Figure S18)**

**between Calibrations 3 and 4 in the Supplementary Material:**

[Figure]

*Figure S15. Differences in absolute bias between Calibrations 3 and 4*

[Figure]

*Figure S16.  Differences in correlation coefficient between Calibrations 3 and 4*

[Figure]

Figure S18.  Differences in alpha index between Calibrations 3 and 4

**We add one new subsection in Results to introduce results from Calibrations 3 and 4**

**3.7 Results from Calibrations 3 and 4**

"We also compare the bias, correlation coefficient, CRPS skill score, and reliability of calibrated forecasts from Calibrations 3 and 4, to evaluate whether we can obtain similar improvements through the bias-correction of input variables if we conduct the ETo forecast calibration in a different way (without using ETo climatological mean and anomalies). Results show that the adoption of bias-correction also leads to lower bias, higher correlation coefficient, and higher CRPS skill score in terms of magnitude, spatial patterns, and trend along the lead times, when ETo forecasts are calibrated directly (Figure S15-S17). In
addition, the alpha index was only slightly different between Calibrations 3 and 4 (Figure S18). This
additional comparison further confirms the general applicability of strategy ii for enhancing NWP-based
ETo forecasting."

**Point #6**

*Methodological choices for evaluation:*

*P7 l 180-185 : why choosing the absolute bias and over a relative measure e.g. percentage bias? This*
*choice makes it difficult to compare the magnitude of the errors in the results across different variables*
*and studies. For example, figure 1 shows a bias between -2 to 2mm/day which does not seem like much*
*compared to other input variables such as precipitation. Figure 3 with a range of -0.1 to 0.1 seems very*
*small. Conversely, percentages are used for the correlation coefficient in Figure 6 so why not use it for*
*the bias?*

**Response: We appreciate the reviewer's valuable comments. Bias shows differences of the**
**mean between forecasts and observations, and could be either positive (overestimation) or**
**negative (underestimation). Larger departures from the observed mean, no matter the bias is**
**positive or negative, suggest more significant inconsistencies with observations. Absolute**
**bias is a good indicator measuring the departure from the observed mean. As a result, using**
**absolute bias, we can compare results from two different calibrations, with smaller absolute**
**bias indicating closer to the observed mean, and thus suggesting better performance.**

**We agree with the reviewer that using percentages will make the results more comparable**
**with other variables, or with other studies. As a result, we change the unit of bias in figures 1,**
**S12, 3, 4 to percentage:**

[Figure]

**Figure 1: Bias in (three panels on the left) raw ETo forecasts constructed with raw forecasts of input variables and (three panels on the right) raw ETo forecasts constructed with bias-corrected input variables.**

[Figure]

*Figure S12. Boxplot of biases in raw ETo forecasts constructed raw (blue) and bias-corrected inputs (pink)*

[Figure]

**Figure 3: Bias in calibrated ETo forecasts of 9 lead times from Calibration 2, in which raw ETo forecasts are constructed with bias-corrected input variables. Maps on the left show the spatial patterns of bias, and the boxplot on the right summarizes results for all grid cells.**

[Figure]

**Figure 4: Differences in absolute bias between calibrated ETo forecasts from Calibration 2 with Calibration 1. Maps on the left show the spatial patterns of difference in absolute bias, and the boxplot on the right summarizes results for all grid cells.**

**Point #7**

*P8 l205-2015: why is climatology used as reference forecast for the skill score? In hydrological forecasting persistence is typically used for short lead times, whereas climatology would be used for longer lead times, see fore example (Pappenberger, Ramos et al. 2015). Could you please expand and justify the choice of reference forecast used and implication of interpretation of results?*

**Response: We really appreciate the reviewer's valuable suggestion and the introduction of this classic paper. We choose the climatology forecasts as the reference rather than using persistence for several reasons:**

**1, Climatology forecasts have been widely used as the reference in the calculation of CRPS skill score for short-term hydroclimate forecasts. Since climatology forecasts have similar errors across all lead times (Bennett et al., 2014), they have been used as the reference to compare forecast skills among different lead times (Academies, 2014; Zhao et al., 2019).**

**2, Persistence is also a good reference, but it's been mainly used for the first two lead times. As demonstrated in figure 5 of Bennett et al. (2014), errors in persistence could increase quickly with lead time. As a result, multiple studies suggested that persistence is good for skill discrimination for short lead times (Pappenberger et al., 2015; Thiemig et al., 2015).**

**Since we investigate 9 lead times in this study, errors in persistency are expected to be significant at long lead times. Using persistence as the reference may artificially exemplify forecast skills at long lead times. As a result, we think the use of climatology forecasts as the reference for the calculation of the CRPS skill score is acceptable.**

**We add the following sentences to section 2.4.3 (Skills of the raw and calibrated forecasts) to explain the use of climatology forecasts as the reference for the calculation of CRPS skill score**

"In the calculation of CRPS skill score, both climatology forecasts or the last observations (persistence) have been used as reference forecasts (Pappenberger et al., 2015; Thiemig et al., 2015). However, reference forecasts based on persistence are more suitable for evaluating the performance of forecasts shorter than two days. As a result, we choose climatology forecasts as the reference, since errors in climate forecasts are similar among all lead times and thus could be used to demonstrate the increasing errors in raw and calibrated forecasts as lead time advances."

**Reference:**

Academies, N.: The science of NOAA'S Operational Hydrologic Ensemble Forecast Service, Bull. Am. Meteorol. Soc., (January), 79–98, doi:10.1175/BAMS-D-12-00081.1, 2014.

Bennett, J. C., Robertson, D. E., Lal, D., Wang, Q. J., Enever, D., Hapuarachchi, P. and Tuteja, N. K.: A System for Continuous Hydrological Ensemble Forecasting (SCHEF) to lead times of 9 days, J. Hydrol., 519, 2832–2846, doi:10.1016/j.jhydrol.2014.08.010, 2014.

Pappenberger, F., Ramos, M. H., Cloke, H. L., Wetterhall, F., Alfieri, L., Bogner, K., Mueller, A.

and Salamon, P.: How do I know if my forecasts are better? Using benchmarks in hydrological ensemble prediction, J. Hydrol., 522, 697–713, doi:10.1016/j.jhydrol.2015.01.024, 2015.

Thiemig, V., Bisselink, B., Pappenberger, F. and Thielen, J.: A pan-African medium-range ensemble flood forecast system, Hydrol. Earth Syst. Sci., 19, 3365–3385, doi:10.5194/hess-19-

3365-2015, 2015.

Zhao, T., Wang, Q. J. and Schepen, A.: A Bayesian modelling approach to forecasting short-term reference crop evapotranspiration from GCM outputs, Agric. For. Meteorol., 269–270(January),

88–101, doi:10.1016/j.agrformet.2019.02.003, 2019.

**Point #8**

*P8 l214. Why is the definition of CRPSS using percentage? As far as I am aware, most studies do not*

*present the CRPSS in terms of percentage, could you please comment on the reason of this choice with*

*references that also use percentages and if there is any advantages?*

**Response: Thank you for the comments. We agree with the reviewer that many studies use**

**ratios when presenting the CRPS skill score. Meanwhile, we also notice that some studies (see**

**the reference list at the bottom of our response to this comment) use percentage as the unit**

**of CRPS skill score.**

**As shown in Figure 7, skills of calibrated forecasts decreased quickly with lead time. As a**

**result, the CRPS skill score approaches zero at lead time 9. One advantage of using the**

**percentage as the unit of CRPS skill score is that small decimals of low skills will be converted**

**to more readable percent.**

**We add the following sentence to explain why the percentage is used as the unit of CRPS skill**

**score:**

**"**We use percentage as the unit of CRPS skill score so low skill scores at long lead times will be converted from small decimals to more readable percent.**"**

**We believe the choice of percentage as the unit of CRPS skill score will not affect the**

**conclusions of this study. Here are some investigations using % as the unit of CRPS skill score:**

Brown, J. D. and Seo, D. J.: A nonparametric postprocessor for bias correction of hydrometeorological and hydrologic ensemble forecasts, J. Hydrometeorol., 11(3), 642–665, doi:10.1175/2009JHM1188.1,

2010.

Kumar, L. G. A., Smith, A. S. D., Gonzalez, G. B. P., Merryfield, V. K. W. and Newman, A. S. Á. M.: A

verification framework for interannual-to-decadal predictions experiments, Clim. Dyn., 40, 245–272, doi:10.1007/s00382-012-1481-2, 2013.

Munkhammar, J., van der Meer, D. and Widén, J.: Probabilistic forecasting of high-resolution clear-sky
index time-series using a Markov-chain mixture distribution model, Sol. Energy, 184(January), 688–695,
doi:10.1016/j.solener.2019.04.014, 2019.

Robertson, D. E. and Wang, Q. J.: Seasonal Forecasts of Unregulated Inflows into the Murray River ,
Australia, Water Resour. Manag., 27, 2747–2769, doi:10.1007/s11269-013-0313-4, 2013.

Schepen, A., Wang, Q. J. and Robertson, D. E.: Seasonal Forecasts of Australian Rainfall through
Calibration and Bridging of Coupled GCM Outputs, Mon. Weather Rev., 142, 1758–1770,
doi:10.1175/MWR-D-13-00248.1, 2014.

Point #9

Analysis and interpretation of results:

*P11 l259-261: why the higher difference in bias in approaches for the Nothern Territory? How does this*
*relate to the biases, errors and assumptions of the NWP? Is it correlated to the biases of specific input*
*variables? How is it correlated to the nonlinear relationship in calculatint ETo? Why are the biases most*
*pronounced for shorter lead times? Please comment.*

**Response: Thank you for the valuable comments. To answer these questions, we present**
**more results to explain how quantile mapping to input variables contributes to improving**
**calibrated ETo forecasts. Specifically, we (1) calculate the correlation coefficients ($r$) between**
**raw/bias-corrected forecasts of the five input variables and AWAP data to further analyze**
**how quantile mapping has improved input variables, in addition to correcting bias (shown in**
**figure 1); (2) investigate the improvements in correlation coefficients between raw ETo**
**forecasts following the bias-correction to input variables and AWAP ETo, to examine how**
**improvements in each variable are translated into the resultant raw ETo forecasts; (3) explain**
**how improvements in raw ETo forecasts through bias-correcting input variables lead to**
**improvements in calibrated ETo forecasts. Please find more details as follows:**

**1, In addition to correcting bias (Figures S2 to S6), quantile mapping also generally improves**
**the temporal patterns of raw forecasts of the input variables. Following figures shows $r$**
**between raw forecasts of the input variables and their corresponding AWAP data (three**
**columns on the left), and improvements in $r$ by quantile mapping (three columns on the**
**right):**

[Figure]

r between raw forecasts and observations

Improvements in r following bias correction

*Figure S7. Correlation coefficients (r) between raw Tmax forecasts and AWAP data (three panels on the left), and improvements in r (three panels on the right) through quantile mapping*

[Figure]

*Figure S8. Correlation coefficients (r) between raw Tmin forecasts and AWAP data (three panels on the left), and improvements in r*
*(three panels on the right) through quantile mapping*

[Figure]

r between raw forecasts and observations    Improvements in r following bias correction

*Figure S9. Correlation coefficients (r) between raw vapor pressure forecasts and AWAP data (three panels on the left), and*
*improvements in r (three panels on the right) through quantile mapping*

[Figure]

*Figure S10. Correlation coefficients (r) between raw solar radiation forecasts and AWAP data (three panels on the left), and*
*improvements in r (three panels on the right) through quantile mapping*

[Figure]

*Figure S11. Correlation coefficients (r) between raw wind speed forecasts and AWAP data (three panels on the left), and*
*improvements in r (three panels on the right) through quantile mapping*

**As shown in the above figures, *r* between raw forecasts of the input variables and AWAP data**
**varies with the input variables. The two temperature variables have higher *r* values than the**
**other three variables, and wind speed forecasts demonstrate the lowest correlation with**
**AWAP data. For all variables, the *r* decreases with lead time, indicating higher uncertainties in**
**raw forecasts at longer lead times.**

**Quantile mapping generally improves the correlation between forecasts of the input**
**variables and AWAP data. The above figures show that bias-corrected forecasts demonstrate**
**higher *r* for Tmax, solar radiation, and wind speed across most parts of Australia; for Tmin**
**and vapor pressure, changes in *r* are less significant, and both increases and slight decreases**
**in *r* are observed.**

**We add the above figures to the supplementary. We also add following descriptions to**
**section 3.1:**

"Raw forecasts of the input variables generally agree with the AWAP data in temporal patterns during the
study period, but the r varies with variables (Figures S7-S11). The two temperature variables (Tmax and
Tmin) have higher r (>0.9) than the other three variables, and wind speed forecasts demonstrate the
lowest correlations with AWAP data. For all variables, the r decreases with lead time, indicating higher
uncertainties at long lead times in raw forecasts."

"In addition, quantile mapping also improves the correlation between forecasts of input variables and
AWAP data (Figures S7-S11). The most significant improvements are found in wind speed forecasts, in
which the *r* is improved by up to 0.2 in central and southern parts of Australia. Forecasts of Tmax and
solar radiation also demonstrate higher *r* with the adoption of quantile mapping. Both increases and slight
decreases were found for vapor pressure and Tmin, indicating less significant improvements in temporal
patterns than other variables. "

**2, With the adoption of quantile mapping to raw forecasts of individual variables, raw ETo**
**forecasts (Calibrations 2 or 4) also show higher *r* with observations, than the raw ETo**
**forecasts constructed with the raw forecasts of input variables (Calibrations 1 or 3):**

[Figure]

Differences in correlation coefficient (%)

**Figure 2: The comparison between the correlation coefficient of AWAP ETo and raw ETo forecasts**
**constructed with the bias-corrected inputs vs. the correlation coefficient of AWAP ETo and raw ETo**
**forecasts constructed with the uncorrected inputs. The boxplot on the right summarizes results for all**
**grid cells.**

**As is shown in the above figure, the quantile mapping also improves the temporal patterns of raw ETo**
**forecasts, for the lead times. More significant improvements are found in northern Australia. Due to**
**the nonlinearity in the calculation of ETo using the input variables, spatial patterns of improvements**
**in *r* (Figure 2) do not resemble improvements of any individual input variables. Although both Tmax**
**and wind speed show more significant improvements in northern Australia, where the *r***
**improvements are greater than other regions (Figure 2), improvements in the two variables do not**
**lead to higher *r* in other parts of Australia. As a result, we believe that improvements in *r* of raw ETo**
**forecasts are contributed jointly by these input variables and their interactions.**

**We add the above figure (Figure 2) and the following contents to the manuscript:**

**"**The adoption of quantile mapping to input variables also improves the temporal patterns of raw ETo
forecasts (Figure 2). Compared with the raw ETo forecasts constructed with raw input variables, the raw
ETo forecasts based on bias-corrected inputs generally shows higher correlations with AWAP ETo,
particularly in northern Australia, where r is improved by more than 10%. However, due to the
nonlinearity in the calculation of ETo using the input variables, spatial patterns of improvements in r
(Figure 2) does not resemble improvements in any individual input variables (Figures S7 to S11). The
improvements in r of raw ETo forecasts seem to be contributed jointly by these input variables and their
interactions.**"**

**3, We add the following contents to section 3.3 to explain the spatial patterns of changes in *r***
**and absolute bias:**

**"**Larger reductions in absolute bias in northern Australia coincide with the improvements in the
correlation between raw ETo forecasts and AWAP ETo (Figure 2). However, unlike the improvements in

*r* for all lead times in raw ETo forecasts, the improvements in absolute bias are more pronounced at short
lead times (Days 1-3) than long lead times (Days 7-9). The uneven improvements across different lead
times may be caused by the significant intrinsic uncertainties in forecasts, which have hindered the
manifestation of improvements to raw ETo forecasts at long lead times in calibrated forecasts.**"**

**Based on the above analyses, we can then answer the questions the reviewer raised in this**
**comment.**

**More significant reductions in absolute bias in northern Australia show similar spatial**
**patterns with that of the improvements in *r* between raw ETo forecasts and AWAP ETo. As we**
**further explained in our response to your next comment (#10), deficiencies in NWP models in**
**simulating weather dynamics in tropical regions have been reported. Bias-correction**
**effectively corrects errors in these areas. However, improvements to raw ETo forecasts in r**
**with the application of quantile mapping could not be explained by any individual variable.**
**The nonlinearity in calculating ETo based on the individual variables may have combined**
**improvements in each variable and lead to more significant improvements in northern**
**Australia. Less significant improvements in calibrated ETo forecasts at longer lead times may**
**be caused by the more significant intrinsic uncertainties in forecasts than short lead times.**
**These uncertainties have inhibited the translation of improvements in raw ETo forecasts to**
**calibrated forecasts.**

Point #10
*P13 l282-285: Why lowest score of correlation coefficient in northern Territory? Is it linked to the NWP*
*(and if so how?) or is it linked to observations? E.g. differneces in observations compared to rest of*
*country?*

**Response:  Thank you for the comments. We believe the low correlation results from the**
**NWP forecasts rather than from observations for several reasons:**

**1, Evaluation of the observations (AWAP data) did not show larger errors in northern**
**Australia than other areas of Australia (Jones et al., 2009). As a result, we do not have**
**evidence that the quality of observations in this region is lower than in other regions**

**2, Deficiencies of NWP forecasts in tropical regions in Australia have been well documented.**
**Due to its highly dynamic nature, forecasts for tropical regions often demonstrate larger**
**uncertainties than other climate zones. In the evaluation of NWP forecasts in Australia,**
**tropical zones show lower skills than other regions (Ebert and Mcbride, 2000; Mcbride and**
**Ebert, 2000; Roux et al., 2010). According to Huang et a. (2018), ACCESS models have been**
**suffering from low skills in simulating the convective processes in tropical zones of Australia.**

**3, Raw ETo forecasts constructed with outputs of an early version of the ACCESS model in**
**another study showed higher RMSE in Northern Territory than other regions (Perera et al.,**

**2014), further confirms that lower correlation coefficient is mainly caused by the NWP**
**forecasts.**

**We add the following sentences to section 3.3:**

**"**Deficiencies in ACCESS models in simulating dynamics of tropical climate systems may have
resulted in the low *r* in northern Australia.**"**

**Reference:**

Ebert, E. E. and Mcbride, J. L.: Verification of precipitation in weather systems : determination
of systematic errors, J. Hydrol., 239, 179–202, 2000.

Huang, J., Rikus, L. J., Qin, Y. and Katzfey, J.: Assessing model performance of daily solar
irradiance forecasts over Australia, Sol. Energy, 176(November), 615–626,
doi:10.1016/j.solener.2018.10.080, 2018.

Jones, D. A., Wang, W. and Fawcett, R.: High-quality spatial climate data-sets for Australia, Aust.
Meteorol. Oceanogr. J., 58, 233–248, 2009.

Mcbride, J. L. and Ebert, E. E.: Verification of quantitative precipitation forecasts from
operational numerical weather prediction models over Australia, Weather Forecast., 15(1),
103–121, doi:10.1175/1520-0434(2000)015<0103:VOQPFF>2.0.CO;2, 2000.

Perera, K. C., Western, A. W., Nawarathna, B. and George, B.: Forecasting daily reference
evapotranspiration for Australia using numerical weather prediction outputs, Agric. For.
Meteorol., 194, 50–63, doi:10.1016/j.agrformet.2014.03.014, 2014.

Roux, B., Seed, A., Pagano, T. and Roux, B.: Improved use of precipitation forecasts in short-
term water forecasting – progress report, The Centre for Australian Weather and Climate
Research A partnership between CSIRO and the Bureau of Meteorology Improved., 2010.

## Point #11

*P14 l294-297: The geographical patterns of the correlation performance is very similar to the patterns of*
*the bias performance. Could you please comment why and if the reasons are the same? Are these related*
*to either the NWP or observations?*

**Response: Thank you for the valuable comments. We add the following figure to the**
**manuscript to demonstrate how bias-correction of input variables improves correlations**
**between raw EToforecasts and AWAP ETo:**

[Figure]

**Figure 2: The comparison between the correlation coefficient of AWAP ETo and raw ETo forecasts constructed with the bias-corrected inputs vs. the correlation coefficient of AWAP ETo and raw ETo forecasts constructed with the uncorrected inputs. The boxplot on the right summarizes results for all grid cells.**

**The above figure shows that when input variables are bias-corrected, the resultant raw ETo forecasts show higher correlation coefficients, than raw ETo forecasts constructed with raw inputs. Spatial patterns of the improvements in *r* in raw forecasts for short lead times are consistent with the improvements in *r* in calibrated forecasts (Figure 6). As a result, we believe this is how the new calibration strategy improves the calibration of ETo forecasts. Less significant improvements in ETo forecasts at longer lead times may be caused by the more significant intrinsic uncertainties in raw forecasts than short lead times. These uncertainties have inhibited the translation of improvements to raw ETo forecasts in calibrated forecasts. We have explained the connections between improvements in raw ETo forecasts and calibrated ETo forecasts in response to your comment #9.**

**As we introduced in the manuscript, when we calibrate the raw ETo forecasts (f(t)), we built a conditional distribution ($\tilde{o}(\mathrm{m}(t))$) for observations (o(t)), and 100 values will be drawn from this conditional distribution to generate the calibrated ensemble forecasts:**

$$\tilde{o}(\mathrm{m}(t)) \sim N\left(\mu_o(\mathrm{m}(t)) + r\frac{\sigma_o(\mathrm{m}(t))}{\sigma_f(\mathrm{m}(t))}(f(t) - \mu_f(\mathrm{m}(t))), (1-r^2)\sigma_o^2\right)$$

**in which where $\mathrm{m}(t)$ returns the month k (k=1 to 12) of daily forecasts or observations of day $t$; $\mu_f(\mathrm{m}(t))$ and $\sigma_f(\mathrm{m}(t))$ refer to the marginal distribution's mean and standard deviation of $f(t)$ in month m($t$), respectively; $\mu_o(m(t))$ and $\sigma_o(m(t))$ are the mean and standard deviation of the**

marginal distribution of $o(t)$ in month $\mathrm{m}(t)$; $r$ is the correlation between $f(t)$ and $o(t)$ in the
transformed space.

As a result, when the correlation is improved, it will help improve the estimation of the mean
and standard deviation of the above conditional distributions. As a result, bias in calibrated
forecasts will be further reduced. That is why improvements in bias demonstrate a similar
spatial pattern as those of the correlation coefficient.

To explain improvements in r in calibrated forecasts, we add the following sentence to
section 3.4:

"Spatial patterns of improvements in r in calibrated ETo forecasts (Figure 6) are consistent with
the improvements in r of raw ETo forecasts with the adoption of bias-correction (Figure 2),
particularly for the short lead times. The improvements in *r* of calibrated ETo forecasts (Figure
6) may also lead to more reasonable conditional distributions for a given raw forecast (equation
4). As a result, regions showing improvements in *r* in calibrated ETo forecasts (Figure 6) often
demonstrate reductions in absolute bias (Figure 4)."

## Point #12
*P16 l320-328. Please comment on why the accuracy has larger differences in terms of geographical*
*patterns than for the bias and PIT performance which had very strong localised performance.*

Response: Thank you for the comments. We believe there are four reasons for the differences
in spatial patterns of CRPS skill score (Figure 8) with changes in bias (Figure 4), correlation
coefficient (Figure 6), and alpha index (Figure S13):

1, The metrics measure different features of the quality of forecasts, and may have different
sensitivities to changes in calibrated forecasts. As a result, it is not unexpected that their
spatial patterns show differences. Bias measures average differences; correlation coefficient
shows consistency between observations and forecasts; the CRPS skill score measures the
performance of calibrated forecasts relative to the climatology forecast; the alpha index is an
indicator showing whether the distribution of calibrated forecasts is overconfident or
underconfident. As a result, improvements indicated by these metrics do not necessarily
show exactly the same spatial patterns.

2, The alpha index is less sensitive to changes in forecasts than other metrics. It is well known
that the quality of forecasts often declines with lead time, even for calibrated forecasts. This
tendency can be seen from the correlation coefficient (Figure 5) and CRPS skill score (Figure
7). However, the same trend is not shown in the alpha index (Figure 9). As demonstrated by
figure 9, the alpha index demonstrates similar magnitudes and spatial patterns among the 9
lead times. As was introduced in equations 13 and 14, PIT value and alpha index are mainly
used to measure the consistency between distributions of forecasts and observations.
Improvements achieved through the adoption of calibration strategy ii (e.g., Calibrations 2

**and 4) may not significantly change the statistical distributions of the calibrated forecasts, as**
**evidenced by the *t-test* (Table S2). As a result, differences in the alpha index (Figure S13)**
**between Calibrations 2 and 1 do not show spatial patterns resembling absolute bias (Figure**
**4), correlation coefficient (Figure 6), and CRPS skill score (Figure 8).**

**3, The spatial patterns of improvements in absolute bias, correlation coefficient, and CRPS**
**skill score are generally consistent. We calculate the spatial correlation for changes in CRPS**
**skill score vs. changes in absolute bias (Figure 8 vs. Figure 4), and the spatial correlation for**
**changes in CRPS skill score vs. changes in correlation coefficients (Figure 8 vs. Figure 6). As is**
**shown in the following figure, the metrics show high spatial correlation.**

[Figure]

**4, The upper and lower limits used for the maps may have affected our understanding of the**
**spatial patterns of the evaluation metrics. Following comparison shows that when using**
**narrower limits (-3% to 3%, rather than -5% to 5%) for the color bar of the maps showing**
**improvements in correlation coefficients (the subplot on the right), the spatial pattern looks**
**more consistent with the maps showing increases in CRPS skill score (Figure 8).  In the revised**
**manuscript, we use the plot with narrower color bar limits.**

[Figure]

**To explain spatial patterns of the evaluation metrics, we add a new subsection to the Results section (3.8 Summary of results):**

"Although the selected metrics measure different aspects of forecast quality, they generally agree with each other in demonstrating improvements in calibrated ETo forecasts with the adoption of the Strategy ii. As introduced in the Method section, bias measures average differences; correlation coefficient shows consistency between observations and forecasts in temporal variability; the CRPS skill score measures the performance of the calibrated forecasts relative to climatology forecast; the α index is an indicator showing whether the distribution of calibrated forecasts is overconfident or underconfident. As a result, these metrics may differ from each other in magnitude when used to evaluate different calibrations (Figures 4, 6, 8, and S14). However, improvements in bias, correlation, and skills with the adoption of bias-correction to input variables are generally consistent in spatial patterns. Compared with the other three metrics, the α index demonstrates less significant changes when input variables are bias-corrected first (Table S2 and Figure S14), mainly because this index is less sensitive to changes in calibrated forecasts than other metrics."

**Point #13**

*P16 l329: Results on calibration 2 and 4: what is the comparison between 2 and 4? Why are these only addressed in the evaluation of forecast accuracy section? Why is there no mention of these for the bias and reliability evaluation? I suggest changing the section order and moving this section first. Then, add a sentence in the bias and reliability section to explicitly communicate what results of experiment 3) and 4) are not presented and why.*

Response: Thank you for the valuable suggestions. We check the original submission and
believe your comments refer to Calibrations 3 and 4 here.

As we explain in our response to your comment #5, calibrations 3 and 4 are to further confirm
that whether our strategy is suitable for general application. We further explain the reason of
by adding the following sentences to clarify why Calibrations 3 and 4 are included in this
study in Method:

"The comparison between Calibrations 1 and 2 is to investigate whether the bias-correction of input
variables would further improve ETo forecasts when the calibration is conducted based on ETo anomalies
and climatological mean. We also conduct additional calibrations which post-process ETo forecasts
directly (Calibrations 3 and 4), to test whether the contribution of improving input variables to ETo
forecast calibration, if there is any, will depend on how ETo forecasts are calibrated (based on anomalies
vs. based on ETo). Calibrations 3 and 4 will help evaluate the general applicability of strategy ii to
enhance NWP/GCM-based ETo forecasting. Key steps of the four calibrations could be found in the
schematic diagram introducing how raw ETo forecasts are constructed and how calibrations are
conducted (Figure S1). In the main text, we primarily analyze results from Calibrations 1 and 2.
Improvements with the adoption of bias-correction to input variables in Calibrations 3 and 4 are very
similar to Calibrations 1 and 2 (see the Supplementary Material). To avoid redundancy, we mainly
present results from Calibrations 3 and 4 in the Supplementary Material."

As we introduced in our response to your comment point #5, we add more results (bias,
correlation, and alpha-index) from Calibrations 3 and 4 to the Supplementary Material. We
also add one new subsection (3.7) to briefly introduce the results shown in these figures
(Figures S15-S18).

**Point #14**

*Discussion:*

*There are little to no direct comparison of results and calibration work presented here to any previous*
*methods or studies (which were mentioned in the introduction). To address a research closure, please put*
*the work presented in this paper in context with other studies applying strategy 1 and strategy 2.*

Response: We appreciate the reviewer's valuable suggestion. We explain in detail why we do
not compare our calibration directly with calibrations using other models in our response to
your comment #3. However, we totally agree with the reviewer that it is necessary to
compare our results with previous investigations in ETo forecasting to help the audience
better understand the performance of our calibration. Therefore, we add the following
contents to the Discussion:

"This investigation further highlights the importance of statistical calibration in NWP-based ETo
forecasting (Medina and Tian, 2020). According to an investigation across 40 sites in Australia,
raw ETo forecasts constructed with NWP outputs reasonably captured the magnitude and
variability of ETo, but forecast skills better than climatology were only limited to the first 6 lead times (Perera et al., 2014). Our investigation suggests that statistical calibration could substantially improve forecast skills and successfully extend the skillful forecasts to lead time 9 across Australia. Findings of this investigation agree well with the site-scale short-term ETo forecasting based on GCM outputs (Zhao et al., 2019a) in the improvements of forecast skills through statistical calibration. Calibrated forecasts from Calibration 2 demonstrate similar skills as Zhao et al. (2019a) across three Australian sites. Thanks to the capability of SCC in calibrating short-archived forecasts (Wang et al., 2019), we achieve the improvements based on much shorter archived raw forecasts (3-year vs. 23-year) than Zhao et al. (2019a). Calibrated forecasts from Calibration 2 also demonstrate low biases (0.32-0.95%) comparable with calibrated ETo forecasts (0.49-0.63%) based on the Bayesian Model Averaging (BMA) model and weather forecasts from three NWP models in the U.S. during 2014-2016 (Medina and Tian, 2020)."

**In addition, we also highlight the importance of testing the proposed calibration strategy (strategy ii) in the future in section 4.2:**

"Third, further investigations based on other calibration models are needed to validate findings of this investigation. Our analyses based on two different methods (based on ETo anomalies vs. based on original ETo) demonstrate similar improvements in calibrated ETo forecasts with the adoption of bias-correction to input variables. Additional evaluations will be needed to verify whether forecast skills will be improved using strategy ii but based on a different calibration model."

**Point #15**

*It is unclear whether authors recommend the use of experiment 2) or 4), when and why. In that sense, I question again the inclusion of these experiments without further elaborating and discussing these results.*

**Response: Thank you for the valuable suggestion. As we explain in our response to your comments #5 and #13, the objective of this study is to evaluate the necessity of correcting the input variables prior to ETo forecast calibration. We also further explain that including Calibrations 3 and 4 was to further evaluate whether the strategy could be generally applied to other calibration models. In addition, we add results from Calibrations 3 and 4 and discussed implications from these two calibrations (section 3.7):**

"We also compare the bias, correlation coefficient, CRPS skill score, and reliability of calibrated forecasts from Calibrations 3 and 4, to evaluate whether we can obtain similar improvements through the bias-correction of input variables if we conduct the ETo forecast calibration in a different way (without using ETo climatological mean and anomalies). Results show that the adoption of bias-correction also leads to lower bias, higher correlation coefficient, and higher CRPS skill score in terms of magnitude, spatial patterns, and trend along the lead times, when ETo forecasts are calibrated directly (Figure S15-S17). In addition, the alpha index was only slightly different between Calibrations 3 and 4 (Figure S18). This additional comparison further confirms the general applicability of strategy ii for enhancing NWP-based ETo forecasting."

**Point #16**

Structure:

*The introduction is well structured and appropriately present previous work studies and existing strategies.*

**Response: We appreciate your constructive comments.**

**Point #17**

*The title is a bit lengthy, authors could consider shortening it.*

**Response: We change the title from:**

**"**Bias-correcting input variables prior to combined calibration leads to more skillful forecasts of reference crop evapotranspiration**"**

**to:**

"Bias-correcting input variables enhances forecasting of reference crop evapotranspiration."

**Point #18**

*As noted above, I suggest authors consider the order of results presented in the context of results from experiment 3) and 4).*

**Response: As we explained in our response to your comments #5, #13, and #15, we add a new subsection (3.7) to present results from calibrations 3 and 4 and discuss the implications of these two Calibrations.**

**Point #19**

*Minor comments:*

*P4 l106: I suggest adding a diagram clearly explaining steps and differences of procedure between the calibration experiments.*

**Response: We appreciate the valuable suggestions and create a diagram to show the key steps of the four calibrations**

[Figure]

*Figure S1. Schematic of the four calibrations*

**Point #20**

*P3 l68: '…pressing need to investigate.' Please expand why it is pressing?*

**Response: Thank you for the comments. ETo forecasts have been increasingly used in the planning of farming activities (e.g., amount and timing of irrigation) in Australia. We improve this sentence as follows:**

"Since NWP/GCM-based ETo forecasting is increasingly conducted to support water resource management, there is a need to investigate the necessity of correcting raw forecasts of the input variables in ETo forecast calibration."

**Point #21**

*P3 l74: Calibrate should be calibrate with small cap letter.*

**Response: Thank you for the careful review. We correct this typo.**

 ## Point #22

 *P3 l80-84: There are many efforts to develop downscaling methods, please comment on what was been*
 *done here to downscale ACCESS-G2 to the AWAP grid. Why not scaling AWAP to the match the forecast*
 *grid?*

 **Response: Thank you for the valuable suggestions. In the revised manuscript, we further**
 **introduce that we used bilinear interpolation to remap ACCESS-G2 forecasts. Meanwhile, we**
 **agree with the reviewer that sophisticated methods have been developed to downscale**
 **coarse resolution forecasts to match observations.**

 **In this study, the purpose of the regridding is to connect forecasts with the corresponding**
 **observations so we can calibrate the forecasts, rather than trying to reconstruct the spatial**
 **patterns of forecasts at a finer scale.**

 **We conducted a literature review on the remapping methods used in forecasts post-**
 **processing. It is common that raw forecasts and references data have different spatial**
 **resolutions. We found that bilinear interpolation of forecasts from a coarser resolution to a**
 **finer resolution has been widely used in forecast post-processing and verification. For**
 **example, Hamill et al. (2015) used bilinear interpolation to downscale the resolution of**
 **Global Ensemble Forecast System (GEFS) forecasts from 1° to 1/8° to match observations**
 **before post-processing with an analogy-based model. Yuan et al. (2014) used bilinear**
 **interpolation to remap the Global Ensemble Forecast System (GEFS, with resolutions of**
 **~0.469° and ~0.625°) to match the North-American Land Data Assimilation System (NLDAS,**
 **with the resolution of 1/8°), before the forecasts were post-processed with a quantile**
 **mapping method. Zeng and Yuan (2018) used bilinear interpolation to remap sub-seasonal to**
 **seasonal forecasts from ECMWF (0.25°X0.25°to 0.5°X0.5° for different lead times), NCEP**
 **(1°X1°), China Meteorological Administration (CMA, 1°X1°), Hydrometeorological Centre of**
 **Russia (HMCR, 1.1°X1.4°), and Australian Bureau of Meteorology (BoM, 2°X2°) to a common**
 **resolution of 0.7°, in order to match the reanalysis data. James et al. (2017) regridded the**
 **wind forecasts with bilinear interpolation from the 3-km High-Resolution Rapid Refresh**
 **(HRRR) NWP model to an observation tower in Colorado to evaluate forecast quality. Bowler**
 **et al. (2008) interpolated the ECMWF forecasts with a grid spacing of 1.5° bilinearly to the site**
 **scale for forecast verification. Yuan and Wood (2012) used bilinear interpolation to match**
 **forecasts from the Euro- Mediterranean Centre for Climate Change (CMCC-INGV), the**
 **European Centre for Medium-Range Weather Forecasts (ECMWF), the Leibniz Institute of**
 **Marine Sciences at Kiel University (IFM-GEOMAR), Météo France, and UK Met Office (UKMO),**
 **which have a spatial resolution of 2.5° to match the observation of 1°.**

 **As a result, previous investigations suggested that downscaling with a sophisticated method**
 **could potentially be useful, but that is not necessarily essential in forecast post-processing,**
 **and bilinear interpolation is acceptable.**

**However, we agree with the reviewer that whether a better remapping method will further improve the forecast calibration should be investigated in the future. Therefore, we add the following sentence to section 4.2 (Implications for forecasting of integrated variables and future work):**

"More sophisticated remapping methods should be evaluated to understand the impacts of forecast regridding on statistical calibration."

**Reference:**

Bowler, N.E., Arribas, A., Mylne, K.R., Robertson, K.B., Beare, S.E., 2008. The MOGREPS short-range ensemble prediction system. Q. J. R. Meteorol. Soc. 722, 703–722. https://doi.org/10.1002/qj

Hamill, T., Scheuerer, M., Bates, G., 2015. Analog Probabilistic Precipitation Forecasts Using GEFS Reforecasts and Climatology-Calibrated Precipitation Analyses. Mon. Weather Rev. 143, 3300–3309. https://doi.org/10.1175/MWR-D-15-0004.1

James, E.P., Benjamin, S.G., Marquis, M., 2017. A unfied high-resolution wind and solar dataset from a rapidly updating numerical weather prediction model. Renew. Energy 102, 390–405. https://doi.org/10.1016/j.renene.2016.10.059

Monteiro, J.A.F., Strauch, M., Srinivasan, R., Abbaspour, K., Gucker, B., 2016. Accuracy of grid precipitation data for Brazil : application in river discharge modelling of the Tocantins catchment. Hydrol. Process. 30, 1419–1430. https://doi.org/10.1002/hyp.10708

Yuan, X., Wood, E.F., 2012. On the clustering of climate models in ensemble seasonal forecasting. Geophys. Res. Lett. 39, 1–7. https://doi.org/10.1029/2012GL052735

Yuan, X., Wood, E.F., Liang, M., 2014. Integrating weather and climate prediction: Toward seamless hydrologic forecasting. Geophys. Res. Lett. 5891–5896. https://doi.org/10.1002/2014GL061076.Received

Zeng, D., Yuan, X., 2018. Multiscale Land – Atmosphere Coupling and Its Application in Assessing Subseasonal Forecasts over East Asia. J. Hydrometeology 19, 745–760. https://doi.org/10.1175/JHM-D-17-0215.1

**Point #23**

*P4 l100: please add a comment that SCC model will be described in section 2.3.2*

**Response: We added the following sentence to this section:**

"The calibration model used in this study is the Seasonally Coherent Calibration (SCC) model, which is introduced in detail in section 2.3.2."

**Point #24**

*P5 l134 climatological means or mean? Please rephrase and clarify this sentence.*

**Response: Thank you, and we change it to 'climatological mean'.**

**Point #25**

*P6 l165: Why are only 100 members drawn, is there any difference with a varying number of ensemble members for forecast reliability?*

**Response: Thank you for the comments. We use 100 members because the computation cost is more affordable than using a larger ensemble size.**

**In order to evaluate how different ensemble sizes would affect the reliability and skills of forecasts, we choose 22 sites randomly across Australia and compare the alpha index and CRPS skill score across these sites using 100, 500, and 1000 ensemble members. The following map shows the locations of the 22 sites.**

[Figure]

**The following figure shows the alpha index is almost identical across the selected sites for the three ensemble sizes:**

[Figure]

**Comparison of CRPS skill score shows that different ensemble sizes have negligible impacts on the score:**

[Figure]

**As a result, we conclude that the ensemble size used in this study is reasonable.**

**Point #26**

*Is there a need or a reason to verify accumulated Eto forecast values across lead times (as is often the case for streamflow forecasting)? Please comment.*

**Response: Thank you for the comments. For short-term weather forecasts, which are issued on a daily basis, users are often interested in the short-lead-time forecasts (e.g., lead times 1 to 3). Accumulated forecasts across all lead times will not provide the information that users are particularly interested.**

**In addition, the evaluation by lead time shows that improvements with the adoption of the new calibration strategy (Calibrations 2 and 4) decrease with lead time, but still show better performance than the calibrations (Calibrations 1 and 3) based on raw forecasts of input variables, event at lead time 9. As a result, we are confident that evaluation based on accumulated ETo will not change the conclusion of this study.**

**Point #27**

*P8 l225: 'wind speed is higher than 1m/s than the reference in Australia'. Could you please translate that in terms of percentage so that this statement can be more easily compared to other locations.*

**Response: We add more quantitative information in the evaluation of raw forecasts of input variables and use percentage to measure the changes:**

"The daily minimum temperature (Tmin) is underpredicted by more than 1.5 °C in western and central parts of Australia by the raw forecasts, but is overpredicted by ca. 1 °C in eastern and southern Australia. Vapor pressure is underpredicted in western and central regions by ca.14%, but is overpredicted by ca. 6% in coastal areas of southeastern Australia by the raw forecasts. Raw solar radiation forecasts are about 5% higher than AWAP data across Australia. Forecasted wind speed is higher than the reference data by more than 1 m s-1 (or by ca. 63%) in most parts of Australia. For each input variable, spatial patterns of biases in raw forecasts are consistent across the 9 lead times, demonstrating systematic errors in the raw NWP forecasts."

**Point #28**

*P18 l380' NWP outputs have been increasingly used for ETo forecasting.' For which applications? Please finish the sentence.*

**Response: We modify this sentence as follows:**

" NWP outputs have been increasingly used for ETo forecasting to support water resource management."

**Point #29**

*P18 l385 Addition 'of' in … skill 'of' the calibrated ETo forecasts.*

**Response: We add the missing 'of' to this sentence:**

**"**With this extra step, the bias, correlation coefficient, and skills of the calibrated ETo forecasts are all improved.**"**

**Point #30**

*References:*

*Pappenberger, F., M. H. Ramos, H. L. Cloke, F. Wetterhall, L. Alfieri, K. Bogner, A. Mueller and P. Salamon (2015). "How do I know if my forecasts are better? Using benchmarks in hydrological ensemble prediction." Journal of Hydrology 522: 697-713.*

**Response:  We cite this paper in the revised manuscript in introducing the CRPS skill score.**